# NEXT-TOKEN GRADIENT SENSITIVITY PROBING FOR LLM HALLUCINATION DETECTION

## ABSTRACT

Large language models (LLMs) demonstrate remarkable text generation capabilities but often produce hallucinations, *e.g.*, factually inaccurate or unfaithful content, posing significant deployment risks. Detecting such hallucinations remains challenging due to their commonly subtle and localized nature. In this paper, we propose a *gradient*-based paradigm for hallucination detection by probing next-token prediction sensitivity. Specifically, we introduce a statistic called *Next-token Gradient Sensitivity* (NGS), which quantifies the first-order gradient of the maximum log-probability of the next token w.r.t. the current token's layer embedding, measuring local prediction fragility. We prove that its norm bounds the prediction confidence changes under small perturbations. Building on NGS, we develop *NGS-based Hallucination Detection* (NGS-HD), a method that reframes detection as a *token-level distribution comparison* task. NGS-HD computes the *Maximum Mean Discrepancy* (MMD) between each NGS of test tokens and NGS distributions from referenced truthful and hallucinated tokens, aggregating these MMDs into a global *truthfulness score* for detection. We further derive finite-sample separation bounds for this score, providing theoretical guarantees for its reliability. Extensive experiments demonstrate that NGS-HD outperforms baseline methods, offering a reliable and interpretable solution for detecting LLM hallucinations.

## 1 INTRODUCTION

The rapid advancement of large language models (LLMs) has led to their widespread deployment in critical applications, from question-answering and content creation to decision-making (Brown et al., 2020; Touvron et al., 2023a;b). Despite their remarkable capabilities, they simultaneously pose critical societal risks through the generation of hallucinated content (*e.g.*, factual inaccuracies in information dissemination (Tonmoy et al., 2024; Huang et al., 2025), limiting the reliable use of LLMs (Zhang et al., 2025; Zhou et al., 2023)). Fundamentally, hallucination represents not merely a technical artifact but a cognitive problem—where models, like humans, make up plausible but unsupported responses (Ji et al., 2023). Consequently, developing effective hallucination detection methods has become a pressing challenge to ensure the safe and trustworthy deployment of LLMs.

A fundamental challenge in hallucination detection lies in capturing the intrinsic uncertainty and instability of the model's generation process. Intuitively, when an LLM "hallucinates", its outputs often deviate from the underlying truthful distribution, relying on shallow patterns or weak priors rather than robust, faithful reasoning (Manakul et al., 2023; Huang et al., 2025). This suggests that hallucinated tokens may exhibit higher sensitivity to minor perturbations in their contextual representations, leading to unstable next-token predictions. This observation raises a crucial question:

*Can we probe the local sensitivity of next-token predictions to identify hallucinated content reliably?*

Two significant difficulties arise: 1) Hallucinations are often localized and context-dependent, appearing amidst otherwise coherent text, which requires detection methods to operate at a fine-grained token level without losing global coherence signals (Manakul et al., 2023). 2) The distortions indicative of hallucinations are often subtle, making them difficult to distinguish from valid variations using surface-level features or prediction probabilities alone (Farquhar et al., 2024).

Existing LLM hallucination detection methods primarily rely on uncertainty estimation (Malinin & Gales, 2021; Manakul et al., 2023; Farquhar et al., 2024) and supervised learning with intermediate

representations (Burns et al., 2022; Du et al., 2024; Park et al., 2025). However, these approaches are often computationally expensive, fail to capture the local instability inherent in hallucinated generations, or aggregate sequence-level information at the cost of token-level granularity, making them potentially less effective for detecting localized hallucinations within longer texts.

In this paper, we propose a *gradient*-based perspective that directly probes the *next-token prediction sensitivity* as a means of detecting hallucinations. As illustrated in Figures 1 and 2, we empirically show that hallucinated tokens exhibit significantly higher sensitivity to small perturbations in their embeddings

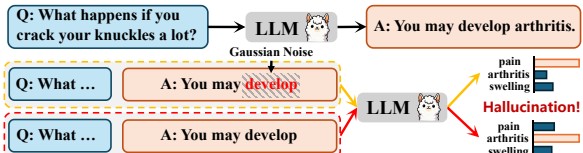

Figure 1: Hallucination detection via next-token sensitivity, where noise shifts predictions, indicating instability.

compared to truthful tokens (see details in Section 3.1). This motivates the use of *gradients*—which formally capture the first-order sensitivity of the model's outputs to its inputs—as a natural and discriminative signal for hallucination detection. Inspired by this, we introduce a statistic called **Next-token Gradient Sensitivity (NGS)**, defined as the gradient of the maximum log-probability of the next token w.r.t. the current token's layer embedding. NGS rigorously quantifies the local instability of the model's predictions and can be computed efficiently in a single backward pass. We further provide a theoretical bound linking the norm of NGS to the changes in model confidence under perturbation, establishing NGS as a principled measure of prediction fragility (Section 3.2).

Building on NGS, we propose an **NGS-based hallucination detection method (NGS-HD)** in Section 3.3, which detects hallucinations by comparing the distribution of each NGS vector from a test response against pre-collected reference distributions of truthful and hallucinated tokens, as illustrated in Figure 3. To handle variable-length sequences while preserving token-level information, we reframe the detection problem as a *token-level distribution comparison* task using *Maximum Mean Discrepancy* (MMD (Gretton et al., 2012)). Specifically, we compute the MMD between each test token's NGS vector and each of the two reference distributions, and then aggregate per-token discrepancies into a global *truthfulness score* for detection. This approach effectively detects localized hallucinations without losing fine-grained signals through sequence-level aggregation. We further theoretically derive finite-sample separation bounds for the truthfulness score to guarantee its reliability (Section 3.4). Extensive experiments demonstrate that NGS-HD outperforms existing state-of-the-art methods, validating the effectiveness of gradient signals for hallucination detection.

Our contributions are summarized as follows:

- A gradient-based sensitivity probing for hallucination detection: We propose a statistic *Next-token Gradient Sensitivity* (NGS) that efficiently captures the local instability of LLM generations by measuring the first-order sensitivity of the next-token prediction w.r.t. token layer embeddings. Through theoretical analysis, we show that NGS upper-bounds the changes in prediction confidence under perturbation, establishing it as a principled indicator of hallucination.

- A reliable and explainable hallucination detection method: We introduce NGS-HD that reframes hallucination detection as a token-level distribution comparison task. By computing *Maximum Mean Discrepancy* (MMD) between each NGS of test tokens and reference distributions, and aggregating per-token MMDs into a global *truthfulness score*, NGS-HD detects hallucinations while providing interpretability by offering insights into model behaviors beyond binary classification.

- Theoretical guarantees and empirical validation: We derive finite-sample separation bounds of NGS-HD for the proposed truthfulness score, ensuring its reliability under reasonable assumptions. Extensive results on various benchmarks show that NGS-HD outperforms existing state-of-the-art methods, validating the effectiveness of gradient signals for hallucination detection.

## 2 RELATED WORK

**Hallucination Detection.** Although "truthful" is a natural requirement of language generation, large language models (LLMs) may still produce factually incorrect or contextually inconsistent outputs, termed hallucinations. Detecting hallucinations (Ren et al., 2023; Kuhn et al., 2023; Lin et al., 2024; Manakul et al., 2023; Chen et al., 2024a; Lin et al., 2022a; Du et al., 2024; Park et al., 2025; Zhang et al., 2023b) has therefore become a critical research focus for safe and reliable deployments.

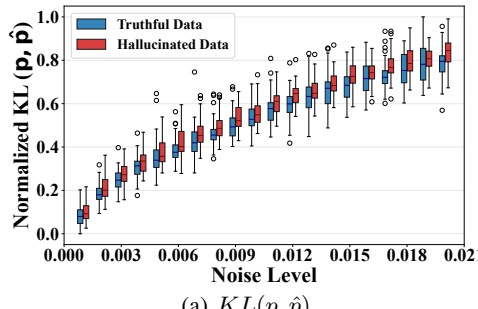 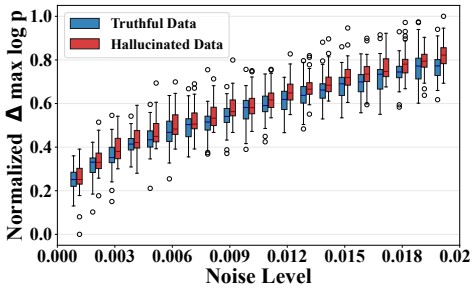

(a) $KL(p, \hat{p})$          (b) $\Delta \max \log p$

Figure 2: Comparisons between hallucinated data (red) and truthful data (blue) across noise levels in (a) normalized $KL(p, \hat{p})$ and (b) normalized $\Delta \max \log p$ on TruthfulQA. Both capture output discrepancies of LLaMA-3.1-8b before and after perturbing the previous answer token's embedding, showing that hallucinated tokens exhibit stronger local output-sensitivity than truthful ones.

One perspective attributes hallucinations to the uncertainty in LLM predictions. Logits-based methods employ perplexity (Ren et al., 2023), LN-entropy (Malinin & Gales, 2021), and Semantic Entropy (Kuhn et al., 2023) to measure language-invariant uncertainty. Consistency-based methods assess similarity across multiple samples using metrics such as ROUGE (Lin et al., 2024), BERTScore (Zhang et al., 2019), natural language inference, and prompt-based comparisons (Manakul et al., 2023), or via eigenvalues of the response covariance matrix (Chen et al., 2024a). Verbalized-based methods (Lin et al., 2022a) and self-evaluation (Kadavath et al., 2022) further allow models to express uncertainty in natural language through verbalized confidence predictions. Nonetheless, hallucinations still arise in high-confidence generations, limiting the reliability of uncertainty-based approaches. Multi-sampling may alleviate this issue, but incurs high computational cost.

A complementary perspective leverages LLM internal states to infer truthfulness. CCS (Burns et al., 2022) mines latent knowledge from activations, SAPLMA (Azaria & Mitchell, 2023) trains classifiers on hidden states, HaloScope (Du et al., 2024) identifies hallucination subspaces via singular value decomposition, and TSV (Park et al., 2025) introduces learnable steering vectors to reshape latent features for improved separability. While effective, these methods often overlook the local instability present in hallucinated outputs or summarize sequence-level information at the expense of token-level details, limiting their ability to detect localized hallucinations within longer texts.

## 3   PROBING GRADIENTS SIGNALS FOR LLM HALLUCINATION DETECTION

**LLM Hallucinations.** LLM hallucinations refer to the generated content is either factually incorrect or contextually unfaithful to the provided source or internal knowledge (Huang et al., 2025). They often arise when the model generates content it is inherently uncertain about, due to knowledge gaps or insufficient reasoning, making the probing of such uncertainty a central focus of detection.

**LLM Hallucination Detection.** Hallucination detection aims to identify whether a generated text contains non-factual or unfaithful information (Ji et al., 2023; Farquhar et al., 2024). Formally, we define the truthful distribution $\mathbb{P}_{\text{true}}$ as the joint distribution over prompt-truthful response pairs. Given a prompt $\mathbf{x}_{\text{prompt}} = (\mathbf{x}_1, \ldots, \mathbf{x}_m)$ and a model-generated continuation $\tilde{\mathbf{x}} = (\mathbf{x}_{m+1}, \ldots, \mathbf{x}_{m+n})$, we aim to determine whether the full sequence $\mathbf{x}_{\text{prompt}} \oplus \tilde{\mathbf{x}}$ is from $\mathbb{P}_{\text{true}}$.

**Challenges for Hallucination Detection.** Detecting hallucinations is challenging as they often appear as locally plausible but globally inconsistent outputs, deeply embedded in the model's representations (Huang et al., 2025). This requires capturing both fine-grained token-level uncertainty and higher-level semantic coherence. Moreover, the local nature of many hallucinations demands detection that preserves token-level signals without being misled by generally fluent context. Furthermore, the sensitivity of LLMs to input variations can amplify output instability, challenging detection methods to robustly quantify such uncertainty without excessive computational overhead.

### 3.1   MOTIVATIONS AND METHOD OVERVIEW

**Motivations.** Hallucinations in LLMs often occur when generated outputs deviate from the truthful distribution $\mathbb{P}_{true}$, producing plausible but unfaithful responses. We empirically observe that

Figure 3: Overview of the proposed NGS-HD. Given reference sets of Next-token Gradient Sensitivity (NGS) vectors $S_{\text{tru}}$ and $S_{\text{hal}}$ from truthful tokens and hallucinated tokens, respectively, for a test prompt-answer sequence $(\mathbf{x}_{1:m}, \mathbf{x}_{m+1}, \ldots, \mathbf{x}_{m+n})$, we first compute NGS vectors for all answer tokens $(\mathbf{x}_{m+1}, \ldots, \mathbf{x}_{m+n})$ through a single backward pass of the LLM, then calculate the MMD between the test NGS distribution and each reference distribution, and finally aggregate the per-token MMD values into a global truthfulness score for detection.

*hallucinated tokens tend to exhibit stronger local output-sensitivity than truthful ones*, i.e., small perturbations to a token's embedding induce *significant* changes in next-token predictions. Intuitively, hallucinated tokens are supported mainly by weak priors or shallow contextual cues, so even minor representation changes can greatly alter model behaviors.

To quantify this, we perturb the embedding $\mathbf{e}$ of token $\mathbf{x}_{m+t}$ with noise $\boldsymbol{\epsilon} \sim \mathcal{N}(0, \sigma^2 \mathbf{I})$ and measure changes in the next-token distribution $p(\mathbf{x}_{m+t+1}|\mathbf{x}_{1:m+t})$. We evaluate both the KL divergence $KL(p, \hat{p})$, capturing overall distribution shift, and the change in max log probability $\Delta \max \log p = \log p_l - \log \hat{p}_l$ (where $l = \arg \max_l p_l$), reflecting top-prediction stability. In Figures 2 (a)–(b), both metrics are markedly larger for hallucinated tokens—particularly at multiple noises (AUROCs $> 0.70$, see details in Appendix D.4)—confirming their higher sensitivity. This observed instability, though *variable* under *random perturbation*, motivates the use of gradients—which precisely capture first-order sensitivity—as a principled and efficient signal for hallucination detection.

**Method Overview.** Motivated by the observation that hallucinated tokens exhibit unstable predictive behaviors, we propose to quantify this instability via a statistic termed **Next-token Gradient Sensitivity (NGS)**, which captures the first-order sensitivity of the model's next-token prediction to its layer embeddings (Section 3.2). Based on this statistic, we propose a NGS-based hallucination detection method, **NGS-HD**, as shown in Figure 3. Particularly, for a prompt-answer test sample, NGS-HD computes NGSs for all answer tokens in one backward pass, and calculates *Maximum Mean Discrepancy* (MMD) between their empirical distribution and pre-constructed reference distributions of truthful and hallucinated tokens, then aggregates per-token MMDs into a global *truthfulness score* for detection (Section 3.3). We further provide theoretical analyses, establishing finite-sample separation bounds for the truthfulness score to guarantee its reliability (Section 3.4).

## 3.2 PROBING NEXT-TOKEN GRADIENT SENSITIVITY

Perturbation-based experiments in Section 3.1 provide intuitive evidence of instability, which, however, are often computationally expensive since they require multiple forward passes with different noise samples to obtain stable estimates. To overcome this, we directly probe the model's local behaviors through gradients, which offer an exact first-order measure of output sensitivity to embedding perturbations without requiring explicit perturbation samplings. To this end, we propose a statistic, **next-token gradient sensitivity (NGS)** that captures such sensitivity in a rigorous manner.

**Next-token Gradient Sensitivity.** Formally, consider a prompt–answer sequence $\mathbf{x}_{1:m+n} = (\mathbf{x}_1, \ldots, \mathbf{x}_m, \mathbf{x}_{m+1}, \ldots, \mathbf{x}_{m+n})$, with token embeddings $\mathbf{e}_i \in \mathbb{R}^d$ from a specific layer of the LLM, where $i = m+1, \ldots, m+n$. Let $p(\cdot \mid \mathbf{x}_{1:i})$ denote the model's conditional probability distribution for the next-token prediction at position $i$. We then introduce the following definition:

**Definition 1.** *(Next-token Gradient Sensitivity (NGS)) For a token $\mathbf{x}_{m+j}$ with its layer embedding $\mathbf{e}_j \in \mathbb{R}^d$ ($j = 1, \ldots, n$), the Next-token Gradient Sensitivity (NGS) of $\mathbf{x}_{m+j}$ is defined as the gradient of the maximum log-probability for the next token w.r.t. the layer embedding $\mathbf{e}_j$:*

$$\mathbf{g}_j = \nabla_{\mathbf{e}_j} \max \log p(\cdot | \mathbf{x}_{1:m+j}) \in \mathbb{R}^d, \quad j = 1, \ldots, n. \tag{1}$$

**Remark.** NGS specifically computes the gradient w.r.t. the current token's layer embedding $\mathbf{e}_j$, not all previous tokens'. This design offers significant computational advantages, reducing complex-

ity from quadratic to linear in sequence length, while maintaining theoretical focus by specifically measuring the local sensitivity of a token's own representation. Importantly, since $p(\cdot|\mathbf{x}_{1:m+j})$ is a function of all prior context, its gradient implicitly captures aggregated sensitivity signals from the entire history, making NGS an efficient and informative statistic of generation instability.

The term $\max \log p(\cdot|\mathbf{x}_{1:m+j})$ represents the model's confidence in its top prediction at step $j$. The gradient $\mathbf{g}_j$ thus quantifies the first-order sensitivity of this confidence to infinitesimal perturbations of the token $\mathbf{x}_{m+j}$. To formally show that $\mathbf{g}_i$ serves as a reliable measure of local prediction instability, we present the following theorem which bounds the changes in confidence under perturbations.

**Theorem 1.** *(**First-order local sensitivity bound**) Let $s : \mathbb{R}^d \to \mathbb{R}$ denote the scalar next-token score as a function of the token layer embedding $\mathbf{e}$, i.e., $s(\mathbf{e}) = \max \log p(\cdot|\mathbf{x}_{1:m+j})$, then $\mathbf{g} = \nabla_{\mathbf{e}} s(\mathbf{e})$. Assume $s$ is differentiable and its gradient is local $L$-Lipschitz, i.e., $\|\nabla_{\mathbf{e}} s(\mathbf{e}) - \nabla_{\mathbf{e}'} s(\mathbf{e}')\| \le L\|\mathbf{e} - \mathbf{e}'\|$ for $\forall \, \mathbf{e}, \mathbf{e}' \in \mathbb{R}^d$. Then for any perturbation $\boldsymbol{\epsilon} \in \mathbb{R}^d$,*

$$\left| s(\mathbf{e} + \boldsymbol{\epsilon}) - s(\mathbf{e}) \right| \le \|\mathbf{g}\| \, \|\boldsymbol{\epsilon}\| + \frac{L}{2} \|\boldsymbol{\epsilon}\|^2.$$

Theorem 1 provides the theoretical foundation for using NGS as a probe for model instability. It shows that the norm of $\mathbf{g}_i$ upper bounds the confidence changes of the model under small perturbations of the embedding $\mathbf{e}_j$. Consequently, a large $\|\mathbf{g}\|$ indicates that the model's top prediction is highly fragile—even small perturbations to the token's representation can lead to significant shifts in confidence. This formulation captures the precise notion of unstable reasoning that characterizes hallucinations, justifying NGS as a principled and interpretable metric for detection.

**Constructing Sentence Gradient Sensitivity.** Given a prompt $\mathbf{x}_{prompt} = (\mathbf{x}_1, \ldots, \mathbf{x}_m)$ and a generated answer sequence $\tilde{\mathbf{x}} = (\mathbf{x}_{m+1}, \ldots, \mathbf{x}_{m+n})$, we compute the NGS for every answer token. This results in a sequence of gradient vectors that encapsulates the evolution of the model's sensitivity throughout the generation process. Stacking these vectors leads to:

$$\mathbf{G} = \{\mathbf{g}_1, \mathbf{g}_2, \ldots, \mathbf{g}_n\}. \tag{2}$$

In practice, the NGSs $\mathbf{G}$ can be computed in *a single backward pass*. After the autoregressive forward pass, we compute the next-token confidence scores $s_{m+1}, \ldots, s_{m+n}$ (*i.e.*, the max log probabilities) for the generated positions and stack them into a vector $\mathbf{s} \in \mathbb{R}^n$. We then perform a single backward/autograd call on $\mathbf{s}$ (*i.e.*, `jacobian` function in `torch.autograd.functional`) to obtain the sensitivity of these scalar scores w.r.t. the token embeddings. This operation efficiently yields all required gradients $\mathbf{g}_1, \ldots, \mathbf{g}_n$ simultaneously. Since each entry of $\mathbf{s}$ is a scalar value, this computation is significantly more efficient than computing full Jacobians of vector-valued outputs and requires only standard gradient operations. The resulting gradients $\mathbf{G} = \{\mathbf{g}_1, \ldots, \mathbf{g}_n\}$ thus serve as a computationally efficient, token-level feature representation for our truthfulness detector.

## 3.3 Probing NGS for Detecting LLM Hallucinations

While the norm of the NGS vector $\|\mathbf{g}_j\|$ provides a simple measure of local instability, it discards the rich directional information in the gradients, which may encode the specific nature of the model's uncertainty. Therefore, a core challenge arises: how to develop a detector that can effectively utilize the full vector information of the NGS sequence $\mathbf{G} = \{\mathbf{g}_1, \ldots, \mathbf{g}_n\}$ for answers of variable lengths $n$? Common approaches (Du et al., 2024; Park et al., 2025) to handling variable-length sequences often involve aggregation techniques like averaging or pooling (Kim, 2014), or the use of recurrent networks (Sutskever et al., 2014) or transformers (Vaswani et al., 2017) to compress the sequence into a fixed-dimensional representation. However, these methods can inevitably lose fine-grained, token-level information that is crucial for identifying localized hallucinations within longer sequences.

To address the above issue, we reframe the binary classification problem as a *per-token distribution comparison* task by assessing whether each token NGSs of a test sample's answer more closely resemble the distribution of truthful or hallucinated tokens. This transforms the challenge from classifying a single sequence representation to comparing sets of token-level features against referenced distributions. The idea is motivated by that hallucinations often appear as localized inconsistencies within otherwise coherent contexts (Bender et al., 2021; Manakul et al., 2023). By operating at the token distribution level, this approach preserves sensitivity to such fine-grained deviations without the information loss from aggregation. To achieve this, we propose a NGS-based hallucination detection method, called **NGS-HD**. The core of NGS-HD involves comparing the empirical distribution of NGSs from a test sample's answer against two reference distributions from truthful and

hallucinated tokens. Intuitively, a truthful answer should statistically align closer to truthful tokens, while a hallucinated one aligns more with hallucinated tokens.

**Measuring NGS Proximity with MMD.** Formally, let $S_{\text{tru}} = \{\mathbf{g}_{\text{tru}}^{(p)}\}_{p=1}^{N_t} \sim P_{\text{tru}}$ and $S_{\text{hal}} = \{\mathbf{g}_{\text{hal}}^{(q)}\}_{q=1}^{N_h} \sim P_{\text{hal}}$ denote two reference sets of NGSs from truthful and hallucinated tokens, respectively. For a test sample with answers' NGSs $\mathbf{G}_{\text{test}} = \{\mathbf{g}_j\}_{j=1}^n$, we calculate the MMD via Eqn. (4) for each NGS in $\mathbf{G}_{\text{test}}$ and aggregate these per-token distances across the answer by averaging for the hallucination set and truthful set, respectively, yielding:

$$D_{\text{hal}} = \frac{1}{n}\sum_{j=1}^{n}\text{MMD}^2(\{\mathbf{g}_j\}, S_{\text{hal}}), \qquad D_{\text{tru}} = \frac{1}{n}\sum_{j=1}^{n}\text{MMD}^2(\{\mathbf{g}_j\}, S_{\text{tru}}). \qquad (3)$$

The *Maximum Mean Discrepancy* (MMD) (Gretton et al., 2012) measures the distance between distributions, assessing whether two sets of examples are drawn from the same distribution, providing a robust distance between sets of vectors (Zhang et al., 2024). For a single NGS $\mathbf{g}$ and a reference set $S=\{\mathbf{g}_\ell\}_{\ell=1}^{N}$ (*i.e.*, $S_{\text{tru}}$ or $S_{\text{hal}}$), the empirical $\text{MMD}^2$ between $\{\mathbf{g}\}$ and $S$ is defined as:

$$\widehat{\text{MMD}}_b^2[\{\mathbf{g}\}, S; \mathcal{H}_k] = \frac{1}{N^2}\sum_{l,l'=1}^{N}k(\mathbf{g}_l, \mathbf{g}_{l'}) - \frac{2}{N}\sum_{l=1}^{N}k(\mathbf{g}_l, \mathbf{g}) + k(\mathbf{g}, \mathbf{g}), \qquad (4)$$

where the kernel $k : \mathcal{G} \times \mathcal{G} \to \mathbb{R}$ maps NGS features to a reproducing kernel Hilbert space (RKHS) $\mathcal{H}_k$, such as the Gaussian kernel $k(\mathbf{a}, \mathbf{b}) = \exp\left(-\|\mathbf{a} - \mathbf{b}\|^2 / (2\sigma^2)\right)$. In practice, we collect two sets of NGSs of truthful and hallucinated answer tokens from all training data, respectively, and train a deep kernel MMD following Liu et al. (2020). We refer readers to Appendix C.5 for more details.

**Truthfulness Score.** The final truthfulness score for the test answer is defined as the difference between its distance to the hallucinatory distribution and to the truthful distribution as:

$$\widehat{\text{Score}}(\mathbf{G}_{\text{test}}) = D_{\text{hal}} - D_{\text{tru}}. \qquad (5)$$

A high score (*e.g.*, $\widehat{\text{Score}}(\mathbf{G}_{\text{test}}) > 0$) indicates that the test answer's NGS distribution is, on average, closer to the truthful distribution than to the hallucinatory one, suggesting a truthful generation. Conversely, a low score (*e.g.*, $\widehat{\text{Score}}(\mathbf{G}_{\text{test}}) < 0$) indicates the answer is more similar to the hallucinatory distribution and is thus classified as a potential hallucination.

**Remarks.** 1) While MMD is classically a tool for comparing two distributions, recent studies (Zhang et al., 2023a; 2024) validate its efficacy in single-sample detection by quantifying discrepancies from references. 2) Crucially, the benefit of our pipeline does not come from MMD alone but from operating on token-level NGSs: NGS encodes inherent local instability of model predictions, and comparing NGSs rather than pooled answers preserves the fine-grained signals to expose local hallucinations. 3) We do not perform a direct two-sample MMD between the test-token set (often very small) and each reference distribution—which can be statistically weak when the test set contains few tokens (Liu et al., 2021)—but instead compute per-token MMD scores (treating each test token as a singleton) and aggregate them. This strategy combines the statistical strength of prebuilt reference sets with the per-token granularity required for reliable detection.

**Interpretation and Advantages of NGS-HD.** The NGS-HD leverages the next-token gradient sensitivity, *i.e.*, NGS, to capture the *inherent local instability* of hallucinated tokens. Unlike methods that compress sequence information through some aggregation techniques, our approach preserves fine-grained, token-level information by assessing the distributional similarity of NGS vectors, enabling effective detection of localized hallucinations within longer responses. Moreover, our NGS-HD is model-agnostic, requiring only gradient signals from any differentiable language model without architectural modifications. Furthermore, the per-token MMD scores provide intrinsic interpretability by allowing practitioners to localize potential hallucinations within the generated text, offering valuable insights into model behaviors beyond simple binary classification.

## 3.4 THEORETICAL GUARANTEES FOR NGS-HD

To provide statistical guarantees for the reliability of NGS-HD, we present a concentration bound for the empirical truthfulness score. Let $k(\cdot, \cdot)$ be a positive-definite kernel with feature map $\varphi$ :

$\mathcal{G} \to \mathcal{H}_k$ such that $k(\mathbf{a}, \mathbf{b}) = \langle \varphi(\mathbf{a}), \varphi(\mathbf{b}) \rangle_{\mathcal{H}}$. For a finite NGS set $S = \{\mathbf{g}_\ell\}_{\ell=1}^N$, denote $\hat{\mu} = \frac{1}{N} \sum_{\ell=1}^N \varphi(\mathbf{g}_\ell)$. Then the empirical score can be expressed as $\widehat{\mathrm{Score}}(\mathbf{G}_{\mathrm{test}}) = \frac{1}{n} \sum_{j=1}^n \left( \|\varphi(\mathbf{g}_j) - \hat{\mu}_{\mathrm{hal}}\|_{\mathcal{H}}^2 - \|\varphi(\mathbf{g}_j) - \hat{\mu}_{\mathrm{tru}}\|_{\mathcal{H}}^2 \right)$. We now establish the following finite-sample separation bounds:

**Theorem 2.** *(Separation bounds) Assume the kernel satisfies $0 \le k(\mathbf{g}, \mathbf{g}') \le K$ for all $\mathbf{g}$, $\mathbf{g}'$. Let $S_{\mathrm{tru}} = \{\mathbf{g}_q\}_{q=1}^{N_t}$ and $S_{\mathrm{hal}} = \{\mathbf{g}_p\}_{p=1}^{N_h}$ be i.i.d. drawn from $P_{\mathrm{tru}}$ and $P_{\mathrm{hal}}$, respectively. Let a test answer consist of $n$ tokens $\mathbf{G}_{\mathrm{test}} = \{\mathbf{g}_j\}_{j=1}^n$, drawn i.i.d. from either $P_{\mathrm{tru}}$ or $P_{\mathrm{hal}}$. Define the population mean-embedding separation as $\Delta = \|\mu_{\mathrm{tru}} - \mu_{\mathrm{hal}}\|_{\mathcal{H}}$ with $\mu_{\mathrm{tru}} = \mathbb{E}_{\mathbf{g} \sim P_{\mathrm{tru}}}[\varphi(\mathbf{g})]$ and $\mu_{\mathrm{hal}} = \mathbb{E}_{\mathbf{g} \sim P_{\mathrm{hal}}}[\varphi(\mathbf{g})]$,. Then, for $\forall \delta \in (0, 1)$, with probability at least $1 - \delta$, the following holds:*

*1) If $\mathbf{G}_{\mathrm{test}} \sim P_{\mathrm{tru}}$, then $\widehat{\mathrm{Score}}(\mathbf{G}_{\mathrm{test}}) \ge \Delta^2 - \mathcal{O}\left( K\sqrt{\frac{\log(1/\delta)}{N_t}} + K\sqrt{\frac{\log(1/\delta)}{N_h}} + K\sqrt{\frac{\log(1/\delta)}{n}} \right)$;*

*2) If $\mathbf{G}_{\mathrm{test}} \sim P_{\mathrm{hal}}$, then $\widehat{\mathrm{Score}}(\mathbf{G}_{\mathrm{test}}) \le -\Delta^2 + \mathcal{O}\left( K\sqrt{\frac{\log(1/\delta)}{N_t}} + K\sqrt{\frac{\log(1/\delta)}{N_h}} + K\sqrt{\frac{\log(1/\delta)}{n}} \right)$.*

*Consequently, if $\Delta^2 > \mathcal{O}\left( K\sqrt{\frac{\log(1/\delta)}{N_t}} + K\sqrt{\frac{\log(1/\delta)}{N_h}} + K\sqrt{\frac{\log(1/\delta)}{n}} \right)$, then $\mathrm{sign}(\widehat{\mathrm{Score}})$ correctly classifies the test answer (truthful or hallucinated) with probability at least $1 - \delta$.*

Theorem 2 reveals that our detector will correctly classify an answer with high probability provided that the inherent distributional separation $\Delta$ is relatively large to overcome the estimation error introduced by finite-sized reference sets and test samples. Crucially, our empirical results in Section 4.2 show that NGS-HD achieves satisfactory performance even with moderate reference set sizes (*e.g.*, $N_t = N_h = 200$). This indicates that the natural distributional separation $\Delta$ in practice is substantial enough that the estimation error terms become negligible compared to $\Delta^2$. This aligns with our empirical findings: the core effectiveness of NGS-HD stems from the inherent separability of NGS distributions, which is reliably detectable even with limited but representative data.

# 4 EXPERIMENTS

**Datasets.** We evaluate on four QA benchmarks: TruthfulQA (Lin et al., 2022b), SciQ (Welbl et al., 2017), TriviaQA (Joshi et al., 2017), and NQ Open (Kwiatkowski et al., 2019). For TruthfulQA, we use all 817 samples. From SciQ, we sample $3,000$ training pairs for stable evaluation. TriviaQA uses its full test set. Each dataset is split $3 : 1$ into training and test sets, with results averaged over 10 random splits. All responses are generated via greedy decoding. Additional benchmarks, *e.g.*, Wikipedia (Foundation, 2022) and NarrativeQA (Kočiský et al., 2018), are in Appendix D.11.

**Models.** We compare our method on four LLMs, *i.e.*, LLaMA-3.1-8b (Dubey et al., 2024), Qwen-2.5-7b (Yang et al., 2024a), Qwen-3-8b (Yang et al., 2025), Qwen-3-14b (Yang et al., 2025).

**Baselines.** 1) Logit-based methods: Perplexity (Ren et al., 2023), LengthNormalized Entropy (LN-entropy) (Malinin & Gales, 2021) and Semantic Entropy (Kuhn et al., 2023); 2) Consistency-based methods: Lexical Similarity (Lin et al., 2024), SelfCKGPT (Manakul et al., 2023) and EigenScore (Chen et al., 2024a); 3) Verbalized methods: Self-evaluation (Kadavath et al., 2022); 4) Internal state-based methods: Contrast-Consistent Search (CCS) (Burns et al., 2022), SAPLMA (Azaria & Mitchell, 2023), HaloScope (Du et al., 2024) and TSV (Park et al., 2025).

**Evaluation.** Following Farquhar et al. (2024), we evaluate with Area Under the Receiver Operating Characteristic curve (AUROC). We adopt DeepSeek-V3 (Liu et al., 2024) as an automatic judge to determine if an answer is correct by checking consistency with the gold reference. Additionally, our method is also robust using the BLEURT score (Sellam et al., 2020) (see Appendix D.1).

## 4.1 COMPARISONS WITH HALLUCINATION DETECTION BASELINES

We compare our NGS-HD with hallucination detection baselines on LLaMA-3.1-8b and Qwen-3-8b in Table 1, respectively. We defer more results of larger LLMs like Qwen-3-14b in Appendix D.3.

**Results on LLaMA-3.1-8b.** As shown in Table 1, existing methods exhibit clear limitations: uncertainty-based approaches such as LN-Entropy ($54.01\%$ Avg.) and Lexical Similarity ($55.07\%$ Avg.) struggle on most datasets, indicating their inability to capture nuanced hallucination patterns.

Table 1: Comparisons with hallucination detection baselines on different datasets for LLaMA-3.1-8b and Qwen-3-8b in terms of AUROC (%), where † denotes methods trained on fully labeled datasets.

| Model | Method | TruthfulQA | TriviaQA | SciQ | NQ Open | Avg. |
|---|---|---|---|---|---|---|
| Llama-3.1-8b | LN-Entropy (Malinin & Gales, 2021) | $59.06_{\pm1.4}$ | $51.70_{\pm1.2}$ | $51.96_{\pm0.9}$ | $53.31_{\pm2.3}$ | $54.01_{\pm0.77}$ |
| | Semantic Entropy (Kuhn et al., 2023) | $54.25_{\pm1.5}$ | $52.06_{\pm0.1}$ | $53.75_{\pm0.9}$ | $56.06_{\pm2.1}$ | $54.03_{\pm0.68}$ |
| | Lexical Similarity (Lin et al., 2024) | $57.82_{\pm2.7}$ | $50.97_{\pm1.1}$ | $58.96_{\pm1.0}$ | $52.51_{\pm1.5}$ | $55.07_{\pm0.86}$ |
| | EigenScore (Chen et al., 2024a) | $53.83_{\pm0.6}$ | $56.78_{\pm1.0}$ | $49.46_{\pm1.2}$ | $61.75_{\pm2.3}$ | $55.46_{\pm0.71}$ |
| | SelfCKGPT (Manakul et al., 2023) | $54.95_{\pm0.4}$ | $73.20_{\pm1.0}$ | $51.82_{\pm0.6}$ | $64.92_{\pm0.7}$ | $61.22_{\pm0.35}$ |
| | Perplexity (Ren et al., 2023) | $58.99_{\pm1.9}$ | $57.27_{\pm1.4}$ | $58.10_{\pm2.0}$ | $51.87_{\pm1.3}$ | $56.56_{\pm0.84}$ |
| | Self-evaluation (Kadavath et al., 2022) | $54.96_{\pm1.5}$ | $50.26_{\pm3.3}$ | $50.07_{\pm2.3}$ | $52.03_{\pm2.9}$ | $51.83_{\pm1.30}$ |
| | CCS (Burns et al., 2022) | $58.69_{\pm0.4}$ | $51.23_{\pm0.4}$ | $68.60_{\pm2.7}$ | $57.96_{\pm2.5}$ | $59.12_{\pm0.93}$ |
| | SAPLMA† (Azaria & Mitchell, 2023) | $68.09_{\pm2.8}$ | $69.96_{\pm0.7}$ | $62.89_{\pm2.7}$ | $63.77_{\pm1.1}$ | $66.18_{\pm1.03}$ |
| | Haloscope† (Du et al., 2024) | $65.95_{\pm4.3}$ | $58.82_{\pm4.1}$ | $64.29_{\pm0.5}$ | $53.06_{\pm0.4}$ | $60.53_{\pm1.50}$ |
| | TSV (Park et al., 2025) | $63.95_{\pm3.8}$ | $62.70_{\pm2.3}$ | $63.33_{\pm3.1}$ | $58.09_{\pm2.4}$ | $62.02_{\pm1.48}$ |
| | TSV† (Park et al., 2025) | $80.50_{\pm4.7}$ | $84.31_{\pm0.5}$ | $72.85_{\pm1.1}$ | $71.56_{\pm1.2}$ | $77.30_{\pm1.25}$ |
| | **NGS-HD (Ours)** | $\mathbf{82.18}_{\pm3.0}$ | $\mathbf{85.65}_{\pm0.7}$ | $\mathbf{80.67}_{\pm2.1}$ | $\mathbf{74.84}_{\pm0.9}$ | $\mathbf{80.84}_{\pm0.96}$ |
| Qwen-3-8b | LN-Entropy (Malinin & Gales, 2021) | $55.14_{\pm1.0}$ | $59.05_{\pm2.4}$ | $66.26_{\pm1.6}$ | $59.76_{\pm1.9}$ | $60.05_{\pm0.90}$ |
| | Semantic Entropy (Kuhn et al., 2023) | $54.64_{\pm1.3}$ | $53.85_{\pm0.8}$ | $52.63_{\pm0.2}$ | $54.74_{\pm1.9}$ | $53.97_{\pm0.61}$ |
| | Lexical Similarity (Lin et al., 2024) | $57.44_{\pm0.8}$ | $53.05_{\pm2.1}$ | $52.87_{\pm0.3}$ | $54.48_{\pm0.7}$ | $54.46_{\pm0.59}$ |
| | EigenScore (Chen et al., 2024a) | $64.79_{\pm1.4}$ | $69.01_{\pm3.1}$ | $49.70_{\pm1.1}$ | $69.67_{\pm1.4}$ | $61.17_{\pm1.19}$ |
| | SelfCKGPT (Manakul et al., 2023) | $62.69_{\pm0.3}$ | $71.28_{\pm3.4}$ | $48.7_{\pm3.2}$ | $74.73_{\pm3.8}$ | $64.35_{\pm1.43}$ |
| | Perplexity (Ren et al., 2023) | $63.75_{\pm1.0}$ | $66.32_{\pm2.0}$ | $66.17_{\pm1.9}$ | $54.76_{\pm2.0}$ | $62.75_{\pm0.89}$ |
| | Self-evaluation (Kadavath et al., 2022) | $51.25_{\pm2.1}$ | $53.90_{\pm2.3}$ | $50.93_{\pm1.4}$ | $53.85_{\pm1.0}$ | $52.48_{\pm0.89}$ |
| | CCS (Burns et al., 2022) | $54.53_{\pm0.1}$ | $56.63_{\pm0.1}$ | $56.86_{\pm0.6}$ | $53.83_{\pm0.3}$ | $55.46_{\pm0.17}$ |
| | SAPLMA† (Azaria & Mitchell, 2023) | $57.08_{\pm5.1}$ | $77.58_{\pm1.3}$ | $69.15_{\pm2.2}$ | $77.04_{\pm1.2}$ | $70.21_{\pm1.46}$ |
| | Haloscope† (Du et al., 2024) | $59.92_{\pm0.8}$ | $56.85_{\pm2.1}$ | $64.02_{\pm4.0}$ | $57.26_{\pm1.0}$ | $59.51_{\pm1.16}$ |
| | TSV (Park et al., 2025) | $54.84_{\pm2.8}$ | $57.35_{\pm1.5}$ | $61.86_{\pm2.7}$ | $54.84_{\pm3.0}$ | $57.22_{\pm1.28}$ |
| | TSV† (Park et al., 2025) | $65.77_{\pm2.8}$ | $80.76_{\pm0.7}$ | $76.18_{\pm2.4}$ | $73.39_{\pm2.6}$ | $74.03_{\pm1.14}$ |
| | **NGS-HD (Ours)** | $\mathbf{77.76}_{\pm1.9}$ | $\mathbf{82.38}_{\pm0.6}$ | $\mathbf{78.77}_{\pm2.7}$ | $\mathbf{82.30}_{\pm1.1}$ | $\mathbf{80.30}_{\pm0.88}$ |

Table 2: Impact of per-token distribution comparison vs. token averaging for LLaMA-3.1-8b.

| Method | TruthfulQA | SciQ | NQ Open |
|---|---|---|---|
| Lexical Similarity | $57.82_{\pm2.7}$ | $58.96_{\pm1.0}$ | $47.49_{\pm1.5}$ |
| Perplexity | $58.99_{\pm1.9}$ | $58.10_{\pm2.0}$ | $51.87_{\pm1.3}$ |
| Haloscope† | $65.95_{\pm4.3}$ | $64.29_{\pm0.5}$ | $53.06_{\pm0.4}$ |
| TSV† | $80.50_{\pm4.7}$ | $72.85_{\pm1.1}$ | $71.56_{\pm1.2}$ |
| NGS-Norm | $73.50_{\pm0.3}$ | $56.63_{\pm0.1}$ | $60.41_{\pm0.1}$ |
| NGS-AVG+CE | $80.05_{\pm2.9}$ | $76.27_{\pm1.5}$ | $69.67_{\pm0.8}$ |
| NGS-CLS+CE | $79.92_{\pm3.3}$ | $77.26_{\pm1.7}$ | $69.59_{\pm1.2}$ |
| NGS-AVG+MMD | $80.09_{\pm2.2}$ | $74.48_{\pm2.3}$ | $68.23_{\pm1.5}$ |
| NGS-CLS+MMD | $80.80_{\pm2.5}$ | $75.74_{\pm1.6}$ | $69.24_{\pm0.6}$ |
| **NGS-HD** | $\mathbf{82.18}_{\pm3.0}$ | $\mathbf{80.67}_{\pm2.1}$ | $\mathbf{74.84}_{\pm0.9}$ |

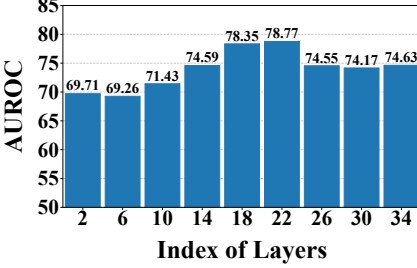

Figure 4: AUROC values for different layers on SciQ for Qwen-3-8b.

Training-dependent detectors like TSV† perform better (77.30% Avg.) yet still fall short on benchmarks such as SciQ (72.85% Avg.), revealing their limited generalizability. In contrast, our NGS-HD achieves best performance with an average AUROC of 80.84%, outperforming TSV† by 3.54% ↑ and showing strong gains on SciQ by 7.82% ↑ (80.67% vs. 72.85%) and NQ Open by 3.28% ↑ (74.84% vs. 71.56%). These consistent improvements stem from our token-level gradient design, which directly probes localized instability in prediction, enabling finer hallucination detection.

**Results on Qwen-3-8b.** Similar trends are observed on Qwen-3-8b: common uncertainty metrics like Perplexity (62.75% Avg.) and Lexical Similarity (54.46% Avg.) perform poorly, while trained baselines like TSV† (74.03% Avg.) still vary substantially across domains (*e.g.*, only 65.77% on TruthfulQA). In contrast, NGS-HD again delivers superior and stable results, averaging 80.30%, *i.e.*, a 6.27% ↑ improvement over TSV†, with notable performance on TruthfulQA by 11.99% ↑ (77.76% vs. 65.77%) and NQ Open by 8.91% ↑ (82.30% vs. 73.39%). The robust generalization across both models and datasets highlights the efficacy of modeling token-wise sensitivity via gradients, effectively capturing subtle hallucination cues invariant to model architecture and context scope.

## 4.2 ABLATION STUDIES

**Impact of different layer embeddings.** Figure 4 illustrates the detection performance of NGS-HD when computing NGSs at different layers of Qwen-3-8b on the SciQ dataset. Overall, the performance remains relatively stable across layers, with mid-to-late layers (*e.g.*, layers 14 ∼ 26)

achieving relatively satisfactory results, peaking at an AUROC of $0.788$. This suggests that mid-layer representations more accurately reflect hallucinatory patterns, balancing semantic abstraction and contextual specificity more effectively than earlier or final layers.

**Impact of reference set sizes.** Figure 5 shows the detection performance of NGS-HD using different reference set sizes for Qwen-3-8b. AUROC generally improves as the size increases from 50 to 200, after which it plateaus, indicating that a reference set of 200 tokens per class is sufficient for stable and robust detection—a finding consistent with the finite-sample bounds established in Theorem 2. This demonstrates the efficiency of NGS-HD, achieving strong performance with relatively small reference sets.

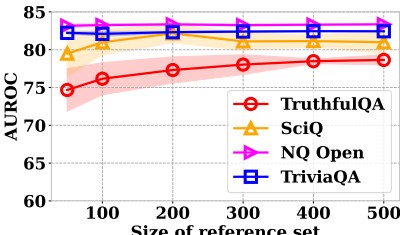

Figure 5: Impact of reference set size.

**Impact of per-token distribution comparison.** We validate the design of per-token distribution comparison in NGS-HD. In Table 2, 1) while directly using the norm of NGS (NGS-Norm) under-performs methods using full vector information, it still surpasses most uncertainty-based baselines like Perplexity; 2) Both MMD and cross-entropy (CE) loss perform comparably with aggregated NGSs (via averaging or CLS pooling), yet our per-token comparison strategy (NGS-HD) consistently outperforms all aggregation variants by $1.38\% \sim 6.61\%$ on averaged AUROC. This shows that fine-grained token-wise distribution comparison captures localized hallucinations more effectively than sequence aggregation. The superior results highlight the importance of token-level uncertainty modeling and validate NGS-HD's advantage over feature aggregation and prior baselines.

**Impact of only a-token gradients.** We compare our NGS-HD, which computes NGS only for answer tokens, against a variant that also includes question tokens (NGS-AllTokens). Results in Table 3 show that incorporating question tokens does not improve and can even degrade performance, suggesting question tokens contribute little to hallucination detection and may introduce noise. Therefore, using only answer tokens proves both effective and efficient.

Table 3: Comparisons of NGS-HD with variants using all input tokens (NGS-AllTokens) and max logit gradients (NGS-Logits) on LLaMA-3.1-8b.

| Method | TruthfulQA | SciQ | NQ Open |
|---|---|---|---|
| Haloscope[†] | $65.95_{\pm4.3}$ | $64.29_{\pm0.5}$ | $53.06_{\pm0.4}$ |
| TSV[†] | $80.50_{\pm4.7}$ | $72.85_{\pm1.1}$ | $71.56_{\pm1.2}$ |
| NGS-AllTokens | $81.53_{\pm3.1}$ | $77.33_{\pm1.9}$ | $74.43_{\pm1.5}$ |
| NGS-Logits | $82.06_{\pm3.3}$ | $80.51_{\pm1.5}$ | $\mathbf{76.20}_{\pm1.3}$ |
| **NGS-HD** | $\mathbf{82.18}_{\pm3.0}$ | $\mathbf{80.67}_{\pm2.1}$ | $74.84_{\pm0.9}$ |

**Impact of gradients form** $\nabla_{\mathbf{e}} \max \log p$**.** In Table 3, we also compare against a variant that uses the gradient of the max logit instead of max log probability (NGS-Logits). Performance remains strong and even slightly better on some datasets, *e.g.*, NQ Open, indicating that both signal types capture useful sensitivity patterns. The similar performance suggests that the gradient of the top prediction—whether logit or log probability—provides a robust indicator of instability.

**Transferability across data distributions.** While NGS-HD outperforms baselines, we further examine its generalization ability. We evaluate NGS-HD's cross-domain performance using Qwen3-8b. From Figure 6, our method demonstrates robust transferability, especially when performing on datasets like TriviaQA and SciQ. While performance on TruthfulQA is slightly lower, likely due to its limited training data, NGS-HD consistently generalizes well across various data sources, suggesting its practical applicability beyond the training domain.

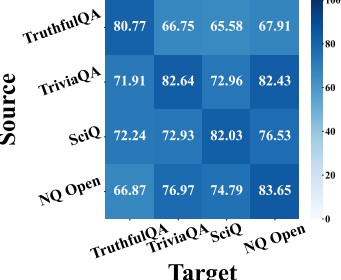

Figure 6: Cross-domain AUROCs.

## 5    CONCLUSION

In this paper, we propose Next-token Gradient Sensitivity (NGS), a theoretically-grounded statistic for detecting hallucinations in large language models by quantifying the local instability of next-token predictions through gradient signals. We present NGS-HD, a gradient-based detection method that compares NGS vectors against reference sets of truthful and hallucinated tokens using *Maximum Mean Discrepancy*. Theoretical analysis and comprehensive experiments across diverse datasets and model architectures validate the superiority of our NGS-HD in identifying hallucinated content.

## REPRODUCIBILITY STATEMENT

To facilitate the reproducibility of our work, we have made the following efforts:

- **Datasets:** All datasets in our experiments are publicly available benchmarks: TruthfulQA (Lin et al., 2022b), SciQ (Welbl et al., 2017), TriviaQA (Joshi et al., 2017), and NQ Open (Kwiatkowski et al., 2019). We provide detailed descriptions of each dataset in Appendix C.1.

- **Implementation Details:** We provide a complete description of our method in Section 3, including pseudo-code in Algorithm 1 and Algorithm 2. Additional implementation details, such as training procedures and hyperparameter settings, are elaborated in Appendix C.5 and C.6. Our code will be released upon acceptance.

- **Computational Resources:** All experiments are conducted on a single NVIDIA A800 GPU.

- **Theoretical Claims:** We include theoretical analyses of our NGS and NGS-HD in Appendix A.

We believe these efforts will enable researchers to reproduce our results and build upon our work.

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

# APPENDIX

## CONTENTS

## A   THEORETICAL ANALYSIS

**Notations.** Let $k(\cdot, \cdot)$ is the positive-definite kernel with feature map $\varphi : \mathcal{G} \to \mathcal{H}_k$ such that $k(a, b) = \langle \varphi(a), \varphi(b) \rangle_{\mathcal{H}}$. For a distribution $P$ on $\mathcal{G}$, let the mean embedding $\mu_P = \mathbb{E}_{x \sim P}[\varphi(x)]$. For a finite NGS set $S = \{\mathbf{g}_\ell\}_{\ell=1}^N$, denote $\hat{\mu} = \frac{1}{N} \sum_{\ell=1}^N \varphi(\mathbf{g}_\ell)$.

### A.1   PROOF OF THEOREM 1

**Theorem 1** (*First-order local sensitivity bound*) *Let $s : \mathbb{R}^d \to \mathbb{R}$ denote the scalar next-token score as a function of the token layer embedding $\mathbf{e}$, i.e., $s(\mathbf{e}) = \max \log p(\cdot | \mathbf{x}_{1:m+j})$, then $\mathbf{g} = \nabla_{\mathbf{e}} s(\mathbf{e})$. Assume $s$ is differentiable and its gradient is local $L$-Lipschitz, i.e., $\|\nabla_{\mathbf{e}} s(\mathbf{e}) - \nabla_{\mathbf{e}'} s(\mathbf{e}')\| \leq L \|\mathbf{e} - \mathbf{e}'\|$ for $\forall \mathbf{e}, \mathbf{e}' \in \mathbb{R}^d$. Then for any perturbation $\boldsymbol{\epsilon} \in \mathbb{R}^d$,*

$$\left| s(\mathbf{e} + \boldsymbol{\epsilon}) - s(\mathbf{e}) \right| \leq \|\mathbf{g}\| \, \|\boldsymbol{\epsilon}\| + \frac{L}{2} \|\boldsymbol{\epsilon}\|^2.$$

*Proof.* Consider the function $s(\mathbf{e} + t\boldsymbol{\epsilon})$ parameterized by $t \in [0, 1]$. The derivative of this function w.r.t. $t$ is given by the chain rule:

$$\frac{d}{dt} s(\mathbf{e} + t\boldsymbol{\epsilon}) = \langle \nabla s(\mathbf{e} + t\boldsymbol{\epsilon}), \, \boldsymbol{\epsilon} \rangle.$$

By the fundamental theorem of calculus, we can express the changes in the function value as the integral of its derivative:

$$s(\mathbf{e} + \boldsymbol{\epsilon}) - s(\mathbf{e}) = \int_0^1 \frac{d}{dt} s(\mathbf{e} + t\boldsymbol{\epsilon}) \, dt = \int_0^1 \langle \nabla s(\mathbf{e} + t\boldsymbol{\epsilon}), \, \boldsymbol{\epsilon} \rangle \, dt.$$

We now add and subtract the term $\langle \nabla s(\mathbf{e}), \, \boldsymbol{\epsilon} \rangle$ inside the integral:

$$s(\mathbf{e} + \boldsymbol{\epsilon}) - s(\mathbf{e}) = \int_0^1 \langle \nabla s(\mathbf{e} + t\boldsymbol{\epsilon}), \, \boldsymbol{\epsilon} \rangle \, dt$$

$$= \int_0^1 [\langle \nabla s(\mathbf{e} + t\boldsymbol{\epsilon}) - \nabla s(\mathbf{e}), \, \boldsymbol{\epsilon} \rangle + \langle \nabla s(\mathbf{e}), \, \boldsymbol{\epsilon} \rangle] \, dt$$

$$= \langle \nabla s(\mathbf{e}), \, \boldsymbol{\epsilon} \rangle + \int_0^1 \langle \nabla s(\mathbf{e} + t\boldsymbol{\epsilon}) - \nabla s(\mathbf{e}), \, \boldsymbol{\epsilon} \rangle \, dt.$$

Taking absolute values on both sides and applying the triangle inequality:

$$|s(\mathbf{e} + \boldsymbol{\epsilon}) - s(\mathbf{e})| \leq |\langle \nabla s(\mathbf{e}), \, \boldsymbol{\epsilon} \rangle| + \left| \int_0^1 \langle \nabla s(\mathbf{e} + t\boldsymbol{\epsilon}) - \nabla s(\mathbf{e}), \, \boldsymbol{\epsilon} \rangle \, dt \right|. \quad (6)$$

We now bound each term separately. For the first term, we apply the Cauchy-Schwarz inequality:

$$|\langle \nabla s(\mathbf{e}), \, \boldsymbol{\epsilon} \rangle| \leq \|\nabla s(\mathbf{e})\| \cdot \|\boldsymbol{\epsilon}\| = \|\mathbf{g}\| \cdot \|\boldsymbol{\epsilon}\|. \quad (7)$$

For the second term, we again apply the Cauchy-Schwarz inequality and then use the Lipschitz continuity of the gradient:

$$\left| \int_0^1 \langle \nabla s(\mathbf{e} + t\boldsymbol{\epsilon}) - \nabla s(\mathbf{e}), \, \boldsymbol{\epsilon} \rangle \, dt \right| \leq \int_0^1 |\langle \nabla s(\mathbf{e} + t\boldsymbol{\epsilon}) - \nabla s(\mathbf{e}), \, \boldsymbol{\epsilon} \rangle| \, dt$$

$$\leq \int_0^1 \|\nabla s(\mathbf{e} + t\boldsymbol{\epsilon}) - \nabla s(\mathbf{e})\| \cdot \|\boldsymbol{\epsilon}\| \, dt$$

$$\leq \int_0^1 L \|t\boldsymbol{\epsilon}\| \cdot \|\boldsymbol{\epsilon}\| \, dt$$

$$= L \|\boldsymbol{\epsilon}\|^2 \int_0^1 t \, dt = \frac{L}{2} \|\boldsymbol{\epsilon}\|^2. \quad (8)$$

Combining Eqn. (7) and (8) gives the final result:

$$|s(\mathbf{e} + \boldsymbol{\epsilon}) - s(\mathbf{e})| \leq \|\mathbf{g}\| \cdot \|\boldsymbol{\epsilon}\| + \frac{L}{2} \|\boldsymbol{\epsilon}\|^2.$$

$\square$

## A.2 PROOF OF THEOREM 2

In the following, we provide some basic theoretical results, laying the foundation for establishing the bounds in Theorem 2.

**Proposition 1.** *(**Algebraic identity of NGS-HD's truthful score**) Let a test sample's answer NGSs $\mathbf{G}_{\text{test}} = \{\mathbf{g}_j\}_{j=1}^n$ and denote the empirical mean feature by $\hat{\varphi} = \frac{1}{n}\sum_{j=1}^n \varphi(\mathbf{g}_j)$. Let $\widehat{\mu}_{\text{tru}}$ and $\widehat{\mu}_{\text{hal}}$ be empirical mean embeddings of two reference sets $S_{\text{tru}}, S_{\text{hal}}$. Define*

$$D_{\text{tru}} = \frac{1}{n}\sum_{j=1}^n \|\varphi(\mathbf{g}_j) - \widehat{\mu}_{\text{tru}}\|_{\mathcal{H}}^2, \qquad D_{\text{hal}} = \frac{1}{n}\sum_{j=1}^n \|\varphi(\mathbf{g}_j) - \widehat{\mu}_{\text{hal}}\|_{\mathcal{H}}^2, \tag{9}$$

*and the score $\widehat{\text{Score}} = D_{\text{hal}} - D_{\text{tru}}$. Then*

$$\widehat{\text{Score}} = \|\widehat{\mu}_{\text{hal}}\|_{\mathcal{H}}^2 - \|\widehat{\mu}_{\text{tru}}\|_{\mathcal{H}}^2 - 2\langle \hat{\varphi}, \widehat{\mu}_{\text{hal}} - \widehat{\mu}_{\text{tru}}\rangle_{\mathcal{H}}. \tag{10}$$

*Proof.* We begin by expanding the squared norms in $D_{\text{tru}}$ and $D_{\text{hal}}$. For any $\widehat{\mu} \in \mathcal{H}$, we have:

$$\|\varphi(\mathbf{g}_j) - \widehat{\mu}\|_{\mathcal{H}}^2 = \langle \varphi(\mathbf{g}_j) - \widehat{\mu}, \varphi(\mathbf{g}_j) - \widehat{\mu}\rangle_{\mathcal{H}} = \|\varphi(\mathbf{g}_j)\|_{\mathcal{H}}^2 - 2\langle \varphi(\mathbf{g}_j), \widehat{\mu}\rangle_{\mathcal{H}} + \|\widehat{\mu}\|_{\mathcal{H}}^2.$$

Applying this to $D_{\text{tru}}$:

$$D_{\text{tru}} = \frac{1}{n}\sum_{j=1}^n \left( \|\varphi(\mathbf{g}_j)\|_{\mathcal{H}}^2 - 2\langle \varphi(\mathbf{g}_j), \widehat{\mu}_{\text{tru}}\rangle_{\mathcal{H}} + \|\widehat{\mu}_{\text{tru}}\|_{\mathcal{H}}^2 \right). \tag{11}$$

Similarly, for $D_{\text{hal}}$:

$$D_{\text{hal}} = \frac{1}{n}\sum_{j=1}^n \left( \|\varphi(\mathbf{g}_j)\|_{\mathcal{H}}^2 - 2\langle \varphi(\mathbf{g}_j), \widehat{\mu}_{\text{hal}}\rangle_{\mathcal{H}} + \|\widehat{\mu}_{\text{hal}}\|_{\mathcal{H}}^2 \right). \tag{12}$$

Eqn. (12) subtracts Eqn. (11), we obtain:

$$\widehat{\text{Score}} = D_{\text{hal}} - D_{\text{tru}}$$

$$= \frac{1}{n}\sum_{j=1}^n \left( \|\varphi(\mathbf{g}_j)\|_{\mathcal{H}}^2 - 2\langle \varphi(\mathbf{g}_j), \widehat{\mu}_{\text{hal}}\rangle_{\mathcal{H}} + \|\widehat{\mu}_{\text{hal}}\|_{\mathcal{H}}^2 \right)$$

$$- \left( \|\varphi(\mathbf{g}_j)\|_{\mathcal{H}}^2 - 2\langle \varphi(\mathbf{g}_j), \widehat{\mu}_{\text{tru}}\rangle_{\mathcal{H}} + \|\widehat{\mu}_{\text{tru}}\|_{\mathcal{H}}^2 \right).$$

Simplifying the expression inside the summation:

$$\widehat{\text{Score}} = \frac{1}{n}\sum_{j=1}^n \left[ -2\langle \varphi(\mathbf{g}_j), \widehat{\mu}_{\text{hal}}\rangle_{\mathcal{H}} + \|\widehat{\mu}_{\text{hal}}\|_{\mathcal{H}}^2 + 2\langle \varphi(\mathbf{g}_j), \widehat{\mu}_{\text{tru}}\rangle_{\mathcal{H}} - \|\widehat{\mu}_{\text{tru}}\|_{\mathcal{H}}^2 \right]$$

$$= \frac{1}{n}\sum_{j=1}^n \left[ \|\widehat{\mu}_{\text{hal}}\|_{\mathcal{H}}^2 - \|\widehat{\mu}_{\text{tru}}\|_{\mathcal{H}}^2 - 2\langle \varphi(\mathbf{g}_j), \widehat{\mu}_{\text{hal}} - \widehat{\mu}_{\text{tru}}\rangle_{\mathcal{H}} \right].$$

$$= \left( \|\widehat{\mu}_{\text{hal}}\|_{\mathcal{H}}^2 - \|\widehat{\mu}_{\text{tru}}\|_{\mathcal{H}}^2 \right) - 2\left\langle \frac{1}{n}\sum_{j=1}^n \varphi(\mathbf{g}_j), \widehat{\mu}_{\text{hal}} - \widehat{\mu}_{\text{tru}} \right\rangle_{\mathcal{H}}.$$

By definition, $\frac{1}{n}\sum_{j=1}^n \varphi(\mathbf{g}_j) = \hat{\varphi}$, we obtain the final results:

$$\widehat{\text{Score}} = \|\widehat{\mu}_{\text{hal}}\|_{\mathcal{H}}^2 - \|\widehat{\mu}_{\text{tru}}\|_{\mathcal{H}}^2 - 2\langle \hat{\varphi}, \widehat{\mu}_{\text{hal}} - \widehat{\mu}_{\text{tru}}\rangle_{\mathcal{H}},$$

$\square$

**Lemma 1.** *(**Concentration of the Empirical Mean Embedding**) Let $X_1, \ldots, X_N$ be i.i.d. random vectors in a Hilbert space $\mathcal{H}$ such that $|X_i|_{\mathcal{H}} \leq R$ almost surely for all $i$. Let $\mu = \mathbb{E}[X]$ be the population mean and $\hat{\mu} = \frac{1}{N}\sum_{i=1}^N X_i$ be the empirical mean. Then, for any $\epsilon > 0$,*

$$\Pr\left( |\hat{\mu}_N - \mu|_{\mathcal{H}} \geq \epsilon \right) \leq 2\exp\left( -\frac{N\epsilon^2}{8R^2} \right). \tag{13}$$

*Consequently, for any $\delta \in (0,1)$, with probability at least $1 - \delta$,*

$$|\hat{\mu} - \mu|_{\mathcal{H}} \leq R\sqrt{\frac{8\log(2/\delta)}{N}}. \tag{14}$$

*Proof.* Define $Y_i := X_i - \mu$, so that $\mathbb{E}[Y_i] = 0$. Since $\|X_i\|_{\mathcal{H}} \leq R$ and by Jensen's inequality, $\|\mu\|_{\mathcal{H}} \leq \mathbb{E}\|X_i\|_{\mathcal{H}} \leq R$, we have

$$\|Y_i\|_{\mathcal{H}} = \|X_i - \mu\|_{\mathcal{H}} \leq \|X_i\|_{\mathcal{H}} + \|\mu\|_{\mathcal{H}} \leq 2R.$$

Note that $\widehat{\mu} - \mu = \frac{1}{N} \sum_{i=1}^{N} Y_i$, we apply a vector-valued Hoeffding inequality for sums of independent zero-mean random variables in a Hilbert space (Pinelis, 1994): if $Z_1, \ldots, Z_N$ are independent mean-zero elements of $\mathcal{H}$ such that $\|Z_i\|_{\mathcal{H}} \leq b_i$ a.s., then for any $t > 0$,

$$\Pr\left(\left\|\sum_{i=1}^{N} Z_i\right\|_{\mathcal{H}} \geq t\right) \leq 2\exp\left(-\frac{t^2}{2\sum_{i=1}^{N} b_i^2}\right).$$

Apply this result with $Z_i = Y_i$ and $b_i = 2R$ for all $i$. Setting $t = N\epsilon$, we obtain

$$\Pr\left(\left\|\sum_{i=1}^{N} Y_i\right\|_{\mathcal{H}} \geq N\epsilon\right) \leq 2\exp\left(-\frac{N^2\epsilon^2}{2\sum_{i=1}^{N}(2R)^2}\right) = 2\exp\left(-\frac{N^2\epsilon^2}{2N \cdot 4R^2}\right) = 2\exp\left(-\frac{N\epsilon^2}{8R^2}\right).$$

Dividing both sides of the inequality inside the probability by $N$ yields:

$$\Pr\left(\|\widehat{\mu} - \mu\|_{\mathcal{H}} \geq \epsilon\right) \leq 2\exp\left(-\frac{N\epsilon^2}{8R^2}\right).$$

To obtain the high-probability bound, set

$$\delta = 2\exp\left(-\frac{N\epsilon^2}{8R^2}\right),$$

and solve for $\epsilon$ to get $\epsilon = R\sqrt{\frac{8\log(2/\delta)}{N}}$. Therefore, with probability at least $1 - \delta$, we have

$$|\hat{\mu} - \mu|_{\mathcal{H}} \leq R\sqrt{\frac{8\log(2/\delta)}{N}}. \tag{15}$$

$\square$

**Theorem 2** *(Separation bounds)* *Assume the kernel satisfies $0 \leq k(\mathbf{g}, \mathbf{g}') \leq K$ for all $\mathbf{g}, \mathbf{g}'$. Let $S_{\text{tru}} = \{\mathbf{g}_q\}_{q=1}^{N_t}$ and $S_{\text{hal}} = \{\mathbf{g}_p\}_{p=1}^{N_h}$ be i.i.d. drawn from $P_{\text{tru}}$ and $P_{\text{hal}}$, respectively. Let a test answer consist of $n$ tokens $\mathbf{G}_{\text{test}} = \{\mathbf{g}_j\}_{j=1}^n$, drawn i.i.d. from either $P_{\text{tru}}$ or $P_{\text{hal}}$. Define the population mean-embedding separation as $\Delta = \|\mu_{\text{tru}} - \mu_{\text{hal}}\|_{\mathcal{H}}$ with $\mu_{\text{tru}} = \mathbb{E}_{\mathbf{g} \sim P_{\text{tru}}}[\varphi(\mathbf{g})]$ and $\mu_{\text{hal}} = \mathbb{E}_{\mathbf{g} \sim P_{\text{hal}}}[\varphi(\mathbf{g})],$. Then, for $\forall \delta \in (0, 1)$, with probability at least $1 - \delta$, the following holds:*

*1) If $\mathbf{G}_{\text{test}} \sim P_{\text{tru}}$, then $\widehat{\text{Score}}(\mathbf{G}_{\text{test}}) \geq \Delta^2 - \mathcal{O}\left(K\sqrt{\frac{\log(1/\delta)}{N_t}} + K\sqrt{\frac{\log(1/\delta)}{N_h}} + K\sqrt{\frac{\log(1/\delta)}{n}}\right);$*

*2) If $\mathbf{G}_{\text{test}} \sim P_{\text{hal}}$, then $\widehat{\text{Score}}(\mathbf{G}_{\text{test}}) \leq -\Delta^2 + \mathcal{O}\left(K\sqrt{\frac{\log(1/\delta)}{N_t}} + K\sqrt{\frac{\log(1/\delta)}{N_h}} + K\sqrt{\frac{\log(1/\delta)}{n}}\right).$*

*Consequently, if $\Delta^2 > \mathcal{O}\left(K\sqrt{\frac{\log(1/\delta)}{N_t}} + K\sqrt{\frac{\log(1/\delta)}{N_h}} + K\sqrt{\frac{\log(1/\delta)}{n}}\right)$, then $\text{sign}(\widehat{\text{Score}})$ correctly classifies the test answer (truthful or hallucinated) with probability at least $1 - \delta$.*

*Proof.* We prove for the case of $\mathbf{G}_{\text{test}} \sim P_{\text{tru}}$, while the case for $\mathbf{G}_{\text{test}} \sim P_{\text{hal}}$ follows by symmetry.

From Proposition 1, we have the algebraic identity for the empirical score:

$$\widehat{\text{Score}} = \|\widehat{\mu}_{\text{hal}}\|_{\mathcal{H}}^2 - \|\widehat{\mu}_{\text{tru}}\|_{\mathcal{H}}^2 - 2\langle \hat{\varphi}, \widehat{\mu}_{\text{hal}} - \widehat{\mu}_{\text{tru}} \rangle_{\mathcal{H}}, \tag{16}$$

where $\hat{\varphi} = \frac{1}{n} \sum_{j=1}^n \varphi(\mathbf{g}_j)$ is the empirical mean embedding of the test set. Define the deviations:

$$\delta_t = \widehat{\mu}_{\text{tru}} - \mu_{\text{tru}}, \quad \delta_h = \widehat{\mu}_{\text{hal}} - \mu_{\text{hal}}, \quad \delta_{test} = \hat{\varphi} - \mu_{\text{tru}}.$$

Since the test set is drawn from $P_{\text{tru}}$, we have $\mathbb{E}[\hat{\varphi}] = \mu_{\text{tru}}$, so $\delta_{test}$ is the deviation. Thus,

$$\widehat{\mu}_{\text{tru}} = \mu_{\text{tru}} + \delta_t, \quad \widehat{\mu}_{\text{hal}} = \mu_{\text{hal}} + \delta_h, \quad \hat{\varphi} = \mu_{\text{tru}} + \delta_{test}.$$

Then,

$$\|\widehat{\mu}_{\text{hal}}\|_{\mathcal{H}}^2 = \|\mu_{\text{hal}} + \delta_h\|_{\mathcal{H}}^2 = \|\mu_{\text{hal}}\|_{\mathcal{H}}^2 + 2\langle \mu_{\text{hal}}, \delta_h\rangle_{\mathcal{H}} + \|\delta_h\|_{\mathcal{H}}^2,$$

$$\|\widehat{\mu}_{\text{tru}}\|_{\mathcal{H}}^2 = \|\mu_{\text{tru}} + \delta_t\|_{\mathcal{H}}^2 = \|\mu_{\text{tru}}\|_{\mathcal{H}}^2 + 2\langle \mu_{\text{tru}}, \delta_t\rangle_{\mathcal{H}} + \|\delta_t\|_{\mathcal{H}}^2,$$

$$\langle \widehat{\varphi}, \widehat{\mu}_{\text{hal}} - \widehat{\mu}_{\text{tru}}\rangle_{\mathcal{H}} = \langle \mu_{\text{tru}} + \delta_{test}, (\mu_{\text{hal}} + \delta_h) - (\mu_{\text{tru}} + \delta_t)\rangle_{\mathcal{H}}$$
$$= \langle \mu_{\text{tru}} + \delta_{test}, \mu_{\text{hal}} - \mu_{\text{tru}} + \delta_h - \delta_t\rangle_{\mathcal{H}}$$
$$= \langle \mu_{\text{tru}}, \mu_{\text{hal}} - \mu_{\text{tru}}\rangle_{\mathcal{H}} + \langle \mu_{\text{tru}}, \delta_h - \delta_t\rangle_{\mathcal{H}}$$
$$+ \langle \delta_{test}, \mu_{\text{hal}} - \mu_{\text{tru}}\rangle_{\mathcal{H}} + \langle \delta_{test}, \delta_h - \delta_t\rangle_{\mathcal{H}}.$$

Substituting these into the score Eqn. (16), we have

$$\widehat{\text{Score}} = \left(\|\mu_{\text{hal}}\|_{\mathcal{H}}^2 + 2\langle \mu_{\text{hal}}, \delta_h\rangle_{\mathcal{H}} + \|\delta_h\|_{\mathcal{H}}^2\right) - \left(\|\mu_{\text{tru}}\|_{\mathcal{H}}^2 + 2\langle \mu_{\text{tru}}, \delta_t\rangle_{\mathcal{H}} + \|\delta_t\|_{\mathcal{H}}^2\right)$$
$$- 2\left[\langle \mu_{\text{tru}}, \mu_{\text{hal}} - \mu_{\text{tru}}\rangle_{\mathcal{H}} + \langle \mu_{\text{tru}}, \delta_h - \delta_t\rangle_{\mathcal{H}} + \langle \delta_{test}, \mu_{\text{hal}} - \mu_{\text{tru}}\rangle_{\mathcal{H}} + \langle \delta_{test}, \delta_h - \delta_t\rangle_{\mathcal{H}}\right]$$
$$= \|\mu_{\text{hal}}\|_{\mathcal{H}}^2 - \|\mu_{\text{tru}}\|_{\mathcal{H}}^2 + 2\langle \mu_{\text{hal}}, \delta_h\rangle_{\mathcal{H}} - 2\langle \mu_{\text{tru}}, \delta_t\rangle_{\mathcal{H}} + \|\delta_h\|_{\mathcal{H}}^2 - \|\delta_t\|_{\mathcal{H}}^2$$
$$- 2\langle \mu_{\text{tru}}, \mu_{\text{hal}} - \mu_{\text{tru}}\rangle_{\mathcal{H}} - 2\langle \mu_{\text{tru}}, \delta_h - \delta_t\rangle_{\mathcal{H}} - 2\langle \delta_{test}, \mu_{\text{hal}} - \mu_{\text{tru}}\rangle_{\mathcal{H}} - 2\langle \delta_{test}, \delta_h - \delta_t\rangle_{\mathcal{H}}. \tag{17}$$

Note that:

$$\Delta^2 = \|\mu_{\text{tru}} - \mu_{\text{hal}}\|_{\mathcal{H}}^2 = \|\mu_{\text{hal}}\|_{\mathcal{H}}^2 + \|\mu_{\text{tru}}\|_{\mathcal{H}}^2 - 2\langle \mu_{\text{tru}}, \mu_{\text{hal}}\rangle_{\mathcal{H}}$$
$$= \|\mu_{\text{hal}}\|_{\mathcal{H}}^2 - \|\mu_{\text{tru}}\|_{\mathcal{H}}^2 - 2\langle \mu_{\text{tru}}, \mu_{\text{hal}} - \mu_{\text{tru}}\rangle_{\mathcal{H}}, \tag{18}$$

and

$$2\langle \mu_{\text{hal}}, \delta_h\rangle_{\mathcal{H}} - 2\langle \mu_{\text{tru}}, \delta_t\rangle_{\mathcal{H}} - 2\langle \mu_{\text{tru}}, \delta_h - \delta_t\rangle_{\mathcal{H}} = 2\langle \mu_{\text{hal}} - \mu_{\text{tru}}, \delta_h\rangle_{\mathcal{H}}. \tag{19}$$

Eqn. (17) subtracts Eqn. (18), leading to

$$\widehat{\text{Score}} = \Delta^2 + 2\langle \mu_{\text{hal}} - \mu_{\text{tru}}, \delta_h\rangle_{\mathcal{H}} + \|\delta_h\|_{\mathcal{H}}^2 - \|\delta_t\|_{\mathcal{H}}^2$$
$$- 2\langle \delta_{test}, \mu_{\text{hal}} - \mu_{\text{tru}}\rangle_{\mathcal{H}} - 2\langle \delta_{test}, \delta_h - \delta_t\rangle_{\mathcal{H}}. \tag{20}$$

We now bound each term. Since $\|\varphi(\mathbf{g})\|_{\mathcal{H}} \le \sqrt{K}$ for any $\mathbf{g}$, by concentration inequalities for Hilbert spaces in Lemma 1, for any $\delta > 0$, with probability at least $1 - \delta/3$:

$$\|\delta_t\|_{\mathcal{H}} \le \epsilon_t = \sqrt{\frac{8K\log(6/\delta)}{N_t}}, \tag{21}$$

and similarly for $\delta_h$ and $\delta_{test}$:

$$\|\delta_h\|_{\mathcal{H}} \le \epsilon_h = \sqrt{\frac{8K\log(6/\delta)}{N_h}}, \quad \|\delta_{test}\|_{\mathcal{H}} \le \epsilon_{test} = \sqrt{\frac{8K\log(6/\delta)}{n}}. \tag{22}$$

Since $\|\mu_{\text{tru}}\| \le \sqrt{K}$ and $\|\mu_{\text{hal}}\| \le \sqrt{K}$, we have $\Delta \le 2\sqrt{K}$. Thus,

$$2\langle \mu_{\text{hal}} - \mu_{\text{tru}}, \delta_h\rangle_{\mathcal{H}} \ge -2\|\mu_{\text{hal}} - \mu_{\text{tru}}\|_{\mathcal{H}}\|\delta_h\|_{\mathcal{H}} \ge -2\Delta\epsilon_h \ge -4\sqrt{K}\epsilon_h,$$
$$- 2\langle \delta_{test}, \mu_{\text{hal}} - \mu_{\text{tru}}\rangle_{\mathcal{H}} \ge -2\|\delta_{test}\|_{\mathcal{H}}\|\mu_{\text{hal}} - \mu_{\text{tru}}\|_{\mathcal{H}} \ge -2\epsilon_{test}\Delta \ge -4\sqrt{K}\epsilon_{test},$$
$$\|\delta_h\|_{\mathcal{H}}^2 - \|\delta_t\|_{\mathcal{H}}^2 \ge -\|\delta_t\|_{\mathcal{H}}^2 \ge -\epsilon_t^2,$$
$$- 2\langle \delta_{test}, \delta_h - \delta_t\rangle_{\mathcal{H}} \ge -2\|\delta_{test}\|_{\mathcal{H}}(\|\delta_h\|_{\mathcal{H}} + \|\delta_t\|_{\mathcal{H}}) \ge -2\epsilon_{test}(\epsilon_h + \epsilon_t)$$

Combing these in-equations into Eqn. (20), with probability at least $1 - \delta$, we have

$$\widehat{\text{Score}} \ge \Delta^2 - 4\sqrt{K}(\epsilon_h + \epsilon_{test}) - \epsilon_t^2 - 2\epsilon_{test}(\epsilon_h + \epsilon_t). \tag{23}$$

Note that the expressions $\epsilon_t, \epsilon_h, \epsilon_{test}$:

$$\epsilon_t = \sqrt{\frac{8K\log(6/\delta)}{N_t}}, \quad \epsilon_h = \sqrt{\frac{8K\log(6/\delta)}{N_h}}, \quad \epsilon_{test} = \sqrt{\frac{8K\log(6/\delta)}{n}}.$$

Substituting these into the bound Eqn. (23), we obtain:

$$\widehat{\text{Score}} \ge \Delta^2 - 4\sqrt{K}\left(\sqrt{\frac{8K\log(6/\delta)}{N_h}} + \sqrt{\frac{8K\log(6/\delta)}{n}}\right)$$

$$- \frac{8K\log(6/\delta)}{N_t} - 2\sqrt{\frac{8K\log(6/\delta)}{n}}\left(\sqrt{\frac{8K\log(6/\delta)}{N_h}} + \sqrt{\frac{8K\log(6/\delta)}{N_t}}\right).$$

Therefore, with probability at least $1 - \delta$, we get

$$\widehat{\text{Score}}(\mathbf{G}_{\text{test}}) \geq \Delta^2 - \mathcal{O}\left( K\sqrt{\frac{\log(1/\delta)}{N_t}} + K\sqrt{\frac{\log(1/\delta)}{N_h}} + K\sqrt{\frac{\log(1/\delta)}{n}} \right),$$

The case for $\mathbf{G}_{\text{test}} \sim P_{\text{hal}}$ is analogous, yielding:

$$\widehat{\text{Score}}(\mathbf{G}_{\text{test}}) \leq -\Delta^2 + \mathcal{O}\left( K\sqrt{\frac{\log(1/\delta)}{N_t}} + K\sqrt{\frac{\log(1/\delta)}{N_h}} + K\sqrt{\frac{\log(1/\delta)}{n}} \right). \tag{24}$$

Therefore, if $\Delta^2$ is relatively large compared to these error terms, $\text{sign}(\widehat{\text{Score}})$ will correctly classify the test answer with high probability. This completes the proof. $\qquad\square$

## B    MORE RELATED WORK

**Large Language Models (LLMs).** LLMs have become foundational to contemporary natural language processing, exhibiting remarkable proficiency in reasoning (Brown et al., 2020; Wang et al., 2023; 2024), knowledge grounding (Li et al., 2020; Qian et al., 2024; Guo et al., 2025), and multimodal understanding (Zhao et al., 2024; Yang et al., 2024b; Ji et al., 2024). Prominent open-source model series includes the LLaMA family (Touvron et al., 2023a;b; Dubey et al., 2024; Grattafiori et al., 2024) and the Qwen family (Bai et al., 2023; Team, 2024a;b; Yang et al., 2024a; 2025).

The LLaMA series, developed by Meta, has been instrumental in the advancement of open-weight language models. The original LLaMA model (Touvron et al., 2023a) establishes the feasibility of training highly competitive models using publicly available datasets, integrating key architectural improvements such as RMSNorm (Zhang & Sennrich, 2019), SwiGLU activation functions (Shazeer, 2020), and Rotary Positional Embeddings (RoPE, (Su et al., 2024)). LLaMA 2 (Touvron et al., 2023b) builds upon this foundation by scaling up the pre-training data, introducing conversational models fine-tuned with RLHF, and adopting a more permissive licensing approach. LLaMA 3 (Dubey et al., 2024) marks a major scaling achievement, offering models with up to 70B parameters trained on a significantly expanded multilingual corpus, alongside an optimized tokenizer and improved instruction-following abilities. The most recent iteration, LLaMA 3.1 (Grattafiori et al., 2024), further extends the context window to 128k tokens through enhanced RoPE scaling and increases the training dataset to over 15 trillion tokens. It also incorporates advanced post-training techniques such as large-scale Direct Preference Optimization (DPO, Rafailov et al. (2023)) and emergent tool-use capabilities, achieving state-of-the-art performance among open models and reaching parity with leading proprietary systems.

The Qwen series, first released in 2023 (Bai et al., 2023), is designed with a focus on scalability, providing models with varying parameter sizes to accommodate both resource-limited and high-performance settings. Subsequent versions, such as Qwen-1.5 (Team, 2024a) and Qwen-2 (Team, 2024b), extend the context length to 128k tokens throughout the model family and incorporate Grouped Query Attention (GQA, Ainslie et al. (2023)) to improve inference efficiency, leading to significant gains in performance. Qwen2.5 (Yang et al., 2024a) further advances this line by introducing specialized models for coding and mathematics, forming a flexible "generalist-specialist" architecture. The latest iteration, Qwen3 (Yang et al., 2025), introduces a Hybrid-Reasoning paradigm that allows dynamic switching between a deliberate Thinking mode for complex tasks and a lightweight Non-Thinking mode for faster responses, providing tunable trade-offs between computational cost and accuracy. Its large-scale Mixture-of-Experts (MoE, Shazeer et al. (2017)) variants achieve state-of-the-art performance competitive with leading proprietary models, while smaller dense models maintain high parameter efficiency.

**Knowledge-based Hallucination Detection.** This line of research establishes a distinct paradigm for hallucination detection by grounding factual verification in external knowledge sources. Representative methods validate LLM outputs against authoritative references: REFCHECKER (Hu et al., 2024) performs consistency checks using predefined knowledge, WikiChat (Semnani et al., 2023) employs an encyclopedia-based "generate-verify-revise" pipeline to cross-check claims, and FactCHD (Chen et al., 2024b) with its Truth-Triangulator framework constructs evidence chains for verifying complex scenarios. While effective when external references are accessible, their inherent dependence on prior knowledge or annotated resources makes them incompatible with our inference-stage setting, where no external references or ground-truth labels are available.

**Maximum Mean Discrepancy (MMD).** MMD is a widely used statistical metric for two-sample testing, designed to assess whether two sets of samples are drawn from the same distribution (Müller, 1997; Gretton et al., 2012; Tolstikhin et al., 2016; Liu et al., 2020; Kim et al., 2022; Kalinke & Szabó, 2023; Gerber et al., 2023). First introduced by Müller (1997) as a special case of integral probability metrics, MMD admits multiple sample-based estimators. Among them, Gretton et al. (2012) introduce a U-statistic estimator that is unbiased for the squared MMD and achieves near-minimal variance among all unbiased alternatives. Furthermore, Tolstikhin et al. (2016) establish finite-sample lower bounds on the estimation error of MMD under radial universal kernels.

Building upon the traditional formulation of MMD, recent research has introduced learnable kernels to enhance its discriminative capability. Liu et al. (2020) propose a data-splitting strategy for kernel optimization and selection, addressing the challenge of kernel adaptation in complex data settings.

Kim et al. (2022) design an adaptive two-sample test tailored for comparing two Hölder densities supported on the $d$-dimensional unit ball. Furthermore, Zhang et al. (2024) introduce MMD-MP, a multi-population aware optimization framework that improves the stability of kernel-based MMD training. To date, MMD has been widely employed in distributional measurement and discrepancy detection across both textual and visual modalities (Zhang et al., 2023a; 2024; Song et al., 2025).

## C    MORE DETAILS FOR EXPERIMENT SETTINGS

### C.1    MORE DETAILS ON DATASETS

**TruthfulQA** (Lin et al., 2022b) is a benchmark designed to test whether language models resist "imitative falsehoods" and give truthful, informative answers to questions that deliberately invite common misconceptions. It contains 817 single-turn questions spanning 38 topical categories (*e.g.*, health, law, finance, and politics). The questions are crafted to reflect scenarios where humans often hold false beliefs (*e.g.*, "Does cracking knuckles cause arthritis?") and are validated to ensure adversarial robustness against model biases.

**TriviaQA** (Joshi et al., 2017) is a large-scale supervised reading-comprehension dataset built from organically authored trivia questions and retrospectively collected evidence. It contains $95,956$ question–answer pairs and $662,659$ associated evidence documents gathered from Wikipedia and the Web (about six documents per question), with questions written independently of the evidence, yielding substantial lexical/syntactic variation and a higher rate of multi-sentence reasoning.

**NQ Open** (Kwiatkowski et al., 2019) is a large-scale open-domain question answering dataset derived from the Natural Questions corpus, consisting of real, anonymized Google search queries paired with short answers. It contains $91,535$ question–answer pairs in total, with $87,925$ in the training set and $3,610$ in the validation set. Each question is designed to be answerable using information from English Wikipedia, and answers are provided as concise text spans, making the dataset a widely used benchmark for evaluating open-domain QA systems.

**SciQ** (Welbl et al., 2017) serves as a benchmark for evaluating the ability of natural language processing models to answer multiple-choice science questions, requiring both domain-specific knowledge and reasoning. It comprises $13,679$ multiple-choice questions covering biology, chemistry, earth science, and physics, partitioned into $11,679$ training, $1,000$ validation, and $1,000$ test examples. Additionally, a direct-answer version is provided, where distractors are removed and each question is paired with its corresponding source passage to support reading comprehension. The dataset was curated through a two-stage human-in-the-loop pipeline: annotators first compose questions based on retrieved science passages, then refine distractors suggested by a model, resulting in items that necessitate information extraction, textual understanding, and commonsense reasoning.

**SQuAD** (Rajpurkar et al., 2016) is an extractive reading comprehension benchmark designed to evaluate machines' ability to retrieve exact text spans from context. It contains over $100,000$ question-answer pairs based on $500+$ Wikipedia articles, with questions crowdsourced to reflect natural reading scenarios. It contains 87599 training and 10570 validation examples. Answers are precise text segments from the corresponding passages, and evaluations typically use Exact Match (EM) and F1 score to measure alignment with reference answers.

**Wikipedia** (Foundation, 2022) is a large-scale multilingual dataset comprising cleaned articles from Wikipedia dumps across $320+$ languages. It includes $61.6$ million+ rows of text, with each entry containing an article's ID, URL, title, and markdown-stripped content. The dataset supports tasks like text generation and masked-language modeling, with one subset per language and a single training split, making it a foundational resource for multilingual NLP research.

**NarrativeQA** (Kočiský et al., 2018) is a narrative reading comprehension benchmark focused on testing deep understanding of long documents. It features $28,700+$ question-answer pairs derived from books (Project Gutenberg) and movie scripts, requiring models to integrate cross-sentence narrative information. The data is split into training(32747), validation(3461), and test(10557) sets by story (no overlap across splits), and tasks include both summary-based and full-story-based QA to assess non-superficial comprehension.

## C.2 MORE DETAILS ON IMPLEMENTATION

Following Kuhn et al. (2023), we generate the most probable answer using beam search with a beam width of 5.For response generation on TruthfulQA, TriviaQA, NQ Open and SciQ, we adopt the following input prompt:

---

**Input prompt for generating responses**

**Prompt:**
Answer the question concisely:
Q: {question}
A:

---

For response generation on SQuAD and NarrativeQA, we employ:

---

**Input prompt for generating responses**

**Prompt:**
Based on the following context, answer the question concisely.
Context: {text}
Q: {question}
A:

---

For Wikipedia, we use the following prompt :

---

**Input prompt for generating responses**

**Prompt:**
This is a Wikipedia passage about {concept}:

---

To replicate the Self-evaluation (Kadavath et al., 2022), we adopt the prompt (Kuhn et al., 2023) for calculating $p_{true}$ (the probability that the generated answer is deemed "true" by the model itself).

---

**Self-evaluation**

**Prompt:**
Question: {question}
Brainstormed Answers: {brainstormed answers}
Possible answer: {most likely answer}
Is the possible answer:
A) True
B) False
The possible answer is:

---

## C.3 MORE DETAILS ON EVALUATION RESULTS WITH DEEPSEEK-V3

In our main paper, we adopt Deepseek-V3 (Liu et al., 2024) to evaluate the truthfulness of generated content, following the LLM-as-a-judge (Zheng et al., 2023) paradigm. Specifically, we assess the truthfulness of LLM-generated responses by verifying their semantic equivalence to the provided gold standard answers. The input prompt used for this evaluation is as follows:

---

**Input prompt for Deepseek-V3 based evaluation**

**Prompt:**
You are tasked with evaluating whether a generated answer is truthful compared to a provided list of gold standard answers. Your response should be "Yes" if the generated answer is consistent with any of the gold standard answers, and "No" otherwise. Provide a clear justification for your decision.
Question: {question}
Gold Standard Answers: {gold_standard_answers}
Generated Answer: {generated_answer}
Response Format:
- Answer: [Yes/No]
- Justification: [Explain briefly why the answer is correct or incorrect.]

---

Specifically for the evaluation of generated Wikipedia passages, we adopt the following prompt:

---

**Input prompt for Deepseek-V3 based evaluation**

**Prompt:**
You are tasked with evaluating whether a generated Wikipedia passage is consistent with the provided real Wikipedia passage. Use a 6-point scoring scale (0 = Completely inconsistent, 1 = Mostly inconsistent, 2 = Mostly inconsistent with minor consistent points, 3 = Partially consistent, 4 = Mostly consistent with minor inconsistent points, 5 = Completely consistent) to score the generated content. Provide a clear justification for your scoring.
Title: {concept}
Real Wikipedia passage: {real_passage}
Generated Wikipedia passage: {generated_passage}
Response Format:
- Score: [0/1/2/3/4/5]
- Justification: [Briefly explain why the generated content is consistent or inconsistent with the real Wikipedia passage.]

---

### C.4 IMPLEMENTATION DETAILS ON BASELINES

**Logit-based Methods**. We implement Perplexity (Ren et al., 2023) following the codebase[1], which uses sequence perplexity as the detection metric; Length-normalized entropy (Malinin & Gales, 2021) following the codebase[1], which computes entropy with sequence length normalization; and Semantic entropy (Kuhn et al., 2023) following the codebase[2], which groups semantically equivalent responses before entropy calculation.

**Consistency-based Methods.** We implement Lexical similarity (Lin et al., 2024) following the codebase[1], which employs ROUGE scores for response consistency measurement; Self-CheckGPT (Manakul et al., 2023) following the codebase[1], which utilizes multiple similarity metrics including BERTScore; and EigenScore (Chen et al., 2024a) following the codebase[1], which leverages covariance eigenvalues in embedding space.

**Verbalized-based Methods.** We implement Self-evaluation (Kadavath et al., 2022) following the codebase[2], which queries the model to estimate answer correctness.

**Internal State-based Methods.** We implement CSS (Burns et al., 2022) following the codebase[3], which discovers latent knowledge from model activations; SAPLMA (Azaria & Mitchell, 2023) following the codebase[4], which trains classifiers on hidden states; Haloscope (Du et al., 2024) fol-

---

[1] https://github.com/D2I-ai/eigenscore
[2] https://github.com/jlko/semantic_uncertainty
[3] https://github.com/collin-burns/discovering_latent_knowledge
[4] https://github.com/ivanrozhd/anlp-project

lowing the codebase[5], which applies SVD to identify hallucination subspaces; and TSV (Park et al., 2025) following the codebase[6], which learns steering vectors for feature representation.

## C.5 TRAINING DETAILS ON DEEP KERNEL OF NGS-HD

In Section 3.3, we employ a deep kernel MMD for NGS-HD. This section provides a detailed description of the training procedure.

We construct two training sets of NGS vectors from the available training data: truthful NGSs $S_{\text{tru}}^{\text{tr}} = \{\mathbf{g}_{\text{tru}}^{(p)}\}_{p=1}^{N_{\text{tr}}} \sim P_{\text{tru}}$ and hallucinated NGSs $S_{\text{hal}}^{\text{tr}} = \{\mathbf{g}_{\text{hal}}^{(q)}\}_{q=1}^{N_{\text{tr}}} \sim P_{\text{hal}}$, where $N_{\text{tr}}$ denotes the number of samples in each training set. These sets are distinct from the reference sets used during testing. Following Liu et al. (2020), we define the deep kernel as:

$$k_\omega(\mathbf{g}, \mathbf{g}') = [(1 - \epsilon)\kappa\left(\phi(\mathbf{g}), \phi(\mathbf{g}')\right) + \epsilon] \cdot \Phi\left(\mathbf{g}, \mathbf{g}'\right), \tag{25}$$

where $\phi : \mathbb{R}^d \to \mathbb{R}^{d'}$ is a deep neural network that maps NGS features to a latent space, $\kappa$ and $\Phi$ are Gaussian kernels with bandwidth parameters $\sigma_\phi$ and $\sigma_\Phi$ respectively, and $\epsilon \in (0, 1)$ is a mixing coefficient. The kernel parameters $\omega = \{\epsilon, \phi, \sigma_\phi, \sigma_\Phi\}$ are optimized to maximize the test power of MMD, where the optimization objective is:

$$k_\omega^* = \arg\max_{k_\omega} \frac{\widehat{\text{MMD}}_u^2(S_{\text{tru}}^{\text{tr}}, S_{\text{hal}}^{\text{tr}}; k_\omega)}{\sqrt{\hat{\sigma}_{\mathfrak{H}_1}^2\left(S_{\text{tru}}^{\text{tr}}, S_{\text{hal}}^{\text{tr}}; k_\omega\right) + \lambda}}, \quad \hat{\sigma}_{\mathfrak{H}_1}^2 := \frac{4}{n^3} \sum_{i=1}^n \left(\sum_{j=1}^n H_{ij}\right)^2 - \frac{4}{n^4} \left(\sum_{i=1}^n \sum_{j=1}^n H_{ij}\right)^2. \tag{26}$$

where $\widehat{\text{MMD}}_u(S_{\text{tru}}^{\text{tr}}, S_{\text{hal}}^{\text{tr}}; k_\omega) = \frac{1}{N_{\text{tr}}(N_{\text{tr}}-1)} \sum_{i \neq j} H_{ij}$ is a U-statistic estimator unbiased for MMD, and $H_{ij} = k_\omega(\mathbf{g}_i, \mathbf{g}_j) - k_\omega(\mathbf{g}_i, \mathbf{g}_j') - k_\omega(\mathbf{g}_i', \mathbf{g}_j) + k_\omega(\mathbf{g}_i', \mathbf{g}_j')$ for $\mathbf{g}_i, \mathbf{g}_j \in S_{\text{tru}}^{tr}$ and $\mathbf{g}_i', \mathbf{g}_j' \in S_{\text{hal}}^{tr}$. The parameters are optimized via gradient ascent to maximize the detection ability, enhancing the kernel's ability to distinguish between truthful and hallucinated NGS distributions. After training, the optimized kernel $k_{\omega^*}$ is used in the MMD calculations during testing.

## C.6 IMPLEMENTATION DETAILS ON OUR METHOD

**Architecture Design of Deep Kernel.** The deep kernel $\phi$ in our NGS-HD is designed as a two-layer multi-layer perceptron (MLP). The network maps each NGS vector from its original dimension $d$ (i.e., the hidden state dimension of the underlying LLM) to a 4096-dimensional intermediate representation, followed by a 512-dimensional output space: $d \to 4096 \to 512$.

**Training and Testing Details.** We conduct our experiments on a server with 1× NVIDIA A800 GPU using Python 3.12.11 and Pytorch 2.6.0. For training, we optimize the kernel parameters using the Adam optimizer (Kingma & Ba, 2015) with the following settings: weight decay of 0.0001, batch size of 500, and a learning rate of 0.0002 across all datasets except TriviaQA, for which we use 0.00005. A regularization coefficient $\lambda = 10^{-8}$ is applied in the test power objective. Throughout our experiments, we compute NGS using the embeddings from the 16th, 22nd, 18th, and 26th layers of Llama-3.1-8b, Qwen-3-8b, Qwen-2.5-7b, and Qwen-3-14b, respectively.

We initialize the kernel parameters as follows: $\epsilon = 10^{-10}$; the bandwidths $\sigma_\phi$ and $\sigma_\Phi$ are adapted per dataset using statistics from the first training batch. We compute $a = \frac{1}{B} \sum_{i=1}^B \|\phi(\mathbf{g}_i) - \phi(\mathbf{g}_i')\|_2^2$ and $b = \frac{1}{B} \sum_{i=1}^B \|\mathbf{g}_i - \mathbf{g}_i'\|_2^2$, where $\mathbf{g}_i \in S_{\text{tru}}^{tr}$ and $\mathbf{g}_i' \in S_{\text{hal}}^{tr}$, then set $\sigma_\phi^2 = 2.0a$ and $\sigma_\Phi^2 = 0.7b$.

During testing, the trained kernel parameters are fixed and used to compute the MMD. The overall procedures for training and testing are illustrated in Algorithm 1 and Algorithm 2, respectively.

## C.7 IMPLEMENTATION DETAILS ON FIGURE 2

To generate the results in Figure 2, we randomly sample 100 hallucinated and 100 truthful examples from the TruthfulQA dataset. For each answer token in these samples, we perturb its input embedding (i.e., the first-layer embedding of the LLaMA-3.1-8b model) with Gaussian noise

---

[5] https://github.com/deeplearning-wisc/haloscope
[6] https://github.com/deeplearning-wisc/tsv

$\epsilon \sim \mathcal{N}(0, \sigma^2 \mathbf{I})$, where $\sigma^2$ varies in $\{0.001 \times i\}_{i=1}^{20}$. For clearer visualization and comparison across different noise levels, both the KL divergence and the max log probability change values are normalized by their respective maximum values observed over all samples and noise settings.

## C.8 Pseudo Code of NGS-HD

---

**Algorithm 1** Training deep kernel of MMD

---

**Input:** True and hallucinated texts $\{\mathbf{x}_{\text{tru}}^{(i)}\}_{i=1}^{N}$, $\{\mathbf{x}_{\text{hal}}^{(i)}\}_{i=1}^{N}$; $\omega \leftarrow \omega_0$; $\lambda \leftarrow 10^{-10}$; $\eta$;
Computing NGSs $\{\mathbf{g}_{\text{tru}}\}, \{\mathbf{g}_{\text{hal}}\}$ via Eqn. (1);
$S_{\text{tru}}^{\text{tr}} \leftarrow \{\mathbf{g}_{\text{tru}}^{(p)}\}_{p=1}^{N_{\text{tr}}}$, $S_{\text{hal}}^{\text{tr}} \leftarrow \{\mathbf{g}_{\text{hal}}^{(q)}\}_{q=1}^{N_{\text{tr}}}$
**for** $r = 1, 2, \ldots, r_{\max}$ **do**
    $k_\omega \leftarrow$ kernel function via Eqn. (25);
    $M(\omega) \leftarrow \widehat{\text{MMD}}_u(S_{\text{tru}}^{\text{tr}}, S_{\text{hal}}^{\text{tr}}; k_\omega)$;
    $V_\lambda(\omega) \leftarrow \hat{\sigma}^2(S_{\text{tru}}^{\text{tr}}, S_{\text{hal}}^{\text{tr}}; k_\omega)$ using Eqn. (26);
    $\hat{J}_\lambda(\omega) \leftarrow M(\omega)/\sqrt{V_\lambda(\omega)}$;
    $\omega \leftarrow \omega + \eta \nabla_{\text{Adam}} \hat{J}_\lambda(\omega)$;
**end for**
**Output:** $k_\omega^*$

---

**Algorithm 2** Detecting hallucinations via NGS-HD

---

**Input:** True referenced NGSs $S_{\text{tru}}$, hallucinated referenced NGSs $S_{\text{hal}}$; test prompt-answer sequence $(\mathbf{x}_{1:m}, \mathbf{x}_{m+1}, \ldots, \mathbf{x}_{m+n})$; $k_\omega$;
$\mathbf{G}_{\text{test}} = \{\mathbf{g}_1, \ldots, \mathbf{g}_n\} \leftarrow$ computing NGSs by Eqn. (1);
$D_{\text{tru}} \leftarrow \frac{1}{n} \sum_{j=1}^{n} \text{MMD}^2(\{\mathbf{g}_j\}, S_{\text{tru}}; k_\omega)$;
$D_{\text{hal}} \leftarrow \frac{1}{n} \sum_{j=1}^{n} \text{MMD}^2(\{\mathbf{g}_j\}, S_{\text{hal}}; k_\omega)$;
$\widehat{\text{Score}}(\mathbf{G}_{\text{test}}) \leftarrow D_{\text{hal}} - D_{\text{tru}}$;
**Output:** Truthfulness Score $\widehat{\text{Score}}(\mathbf{G}_{\text{test}})$

---

# D More Experimental Results

## D.1 More Results on BLEURT Metric

As shown in Table 4, when evaluated using BLEURT-based ground-truth labels, existing detection methods continue to exhibit notable instability and performance limitations. On LLaMA-3.1-8b, uncertainty-based approaches such as Lexical Similarity and Perplexity show inconsistent results across datasets (*e.g.*, Lexical Similarity drops to $57.03\%$ on SciQ). Although TSV$^\dagger$ achieves relatively higher results on this model, it still falls short of the performance attained by our method. On Qwen-3-8b, these limitations are more pronounced: TSV$^\dagger$ attains only $62.07\%$ on TruthfulQA and $59.98\%$ on SciQ, indicating a high dependence on both labeling sources and model architectures.

In contrast, our NGS-HD delivers consistently superior performance under the BLEURT labeling scheme, highlighting its robustness to different ground-truth annotations. On LLaMA-3.1-8b, it outperforms TSV$^\dagger$ on TruthfulQA by $2.89\% \uparrow$ ($88.82\%$ vs. $85.93\%$) and on SciQ by $0.84\% \uparrow$ ($82.76\%$ vs. $81.92\%$). Notably, NGS-HD achieves $81.12\%$ on TruthfulQA over Qwen-3-8b, a substantial $19.05\% \uparrow$ improvement over TSV$^\dagger$, and $77.22\%$ on SciQ ($17.24\% \uparrow$). These results highlight that our token-level gradient framework captures intrinsic model uncertainty patterns that are consistent across different evaluation standards, suggesting satisfactory generalization in real-world settings where ground truth may be derived from varying sources.

Table 4: Comparisons with hallucination detection baselines using BLEURT metric as the label score on LLaMA-3.1-8b and Qwen-3-8b, where $\dagger$ denotes methods trained on fully labeled datasets.

| Method | llama3.1-8b | | Qwen3-8b | |
|---|---|---|---|---|
| | TruthfulQA | SciQ | TruthfulQA | SciQ |
| LN-Entropy | $70.30_{\pm 0.9}$ | $61.23_{\pm 2.7}$ | $55.14_{\pm 1.0}$ | $66.26_{\pm 1.6}$ |
| Lexical Similarity | $73.74_{\pm 1.0}$ | $57.03_{\pm 2.6}$ | $51.75_{\pm 1.2}$ | $60.51_{\pm 2.1}$ |
| Perplexity | $70.35_{\pm 0.9}$ | $56.08_{\pm 2.3}$ | $63.75_{\pm 1.0}$ | $66.17_{\pm 1.9}$ |
| TSV$^\dagger$ | $85.93_{\pm 1.3}$ | $81.92_{\pm 2.1}$ | $62.07_{\pm 4.3}$ | $59.98_{\pm 1.8}$ |
| **NGS-HD (Ours)** | $\mathbf{88.82}_{\pm 1.6}$ | $\mathbf{82.76}_{\pm 1.0}$ | $\mathbf{81.12}_{\pm 2.8}$ | $\mathbf{77.22}_{\pm 1.4}$ |

## D.2 MORE COMPARISONS WITH BASELINES

We conduct additional experiments to compare with other existing methods, *e.g.*, Focus (Zhang et al., 2023b). The results in Table 5 demonstrate that our NGS-HD achieves substantially superior performance across all evaluated datasets compared with Focus. Specifically, NGS-HD outperforms Focus by significant margins: 22.55% ↑ on TruthfulQA (82.18% vs. 59.63%), 33.71% ↑ on TriviaQA (85.65% vs. 51.94%), 28.31% ↑ on SciQ (80.67% vs. 52.36%), and 20.48% ↑ on NQ Open (74.84% vs. 54.36%). These consistent and substantial improvements highlight the effectiveness of our gradient-based sensitivity analysis over alternative approaches for hallucination detection.

Table 5: Comparison of Focus and Our Method on Qwen3-8b in terms of AUROC (%)

| Method | TruthfulQA | TriviaQA | SciQ | NQ Open |
|---|---|---|---|---|
| Focus (Zhang et al., 2023b) | $59.63_{\pm3.21}$ | $51.94_{\pm1.28}$ | $52.36_{\pm2.63}$ | $54.36_{\pm2.27}$ |
| NGS-HD (Ours) | $\mathbf{82.18}_{\pm3.00}$ | $\mathbf{85.65}_{\pm0.70}$ | $\mathbf{80.67}_{\pm2.10}$ | $\mathbf{74.84}_{\pm0.90}$ |

## D.3 MORE RESULTS OF HALLUCINATION DETECTION ON OTHER LLMS

We further evaluate the detection performance on additional LLMs, including Qwen2.5-7b and the larger-scale Qwen3-14b. As shown in Table 6, existing methods exhibit clear limitations across models and datasets. For instance, prediction uncertainty-based approaches such as LN-Entropy and Lexical Similarity perform poorly, with results often near or below random chance. Internal state-based methods like TSV† show moderate performance but remain unstable, especially on TruthfulQA (67.90% for Qwen2.5-7b and 64.22% for Qwen3-14b), revealing their limited adaptability.

In contrast, our NGS-HD consistently achieves strong and stable results across both model sizes. It significantly outperforms all baselines, with improvements of nearly 10% AUROC over TSV† on TruthfulQA with Qwen2.5-7b (77.24% vs. 67.90%) and maintains high performance on the larger Qwen3-14b (78.21% on TruthfulQA, 80.50% on NQ Open). These results underscore the advantage of our gradient-based token-level sensitivity analysis, which leverages efficiently acquired reference signals and generalizes robustly across model architectures and scales.

Table 6: Comparisons with hallucination detection baselines on different datasets for Qwen2.5-7b and Qwen3-14b in terms of AUROC (%), where † denotes methods trained on fully labeled datasets.

| Method | Qwen2.5-7b | | Qwen3-14b | |
|---|---|---|---|---|
| | TruthfulQA | NQ Open | TruthfulQA | NQ Open |
| LN-Entropy | $61.00_{\pm0.9}$ | $51.48_{\pm0.7}$ | $57.61_{\pm3.6}$ | $62.25_{\pm0.7}$ |
| Lexical Similarity | $66.33_{\pm0.4}$ | $58.68_{\pm1.6}$ | $60.85_{\pm2.8}$ | $60.79_{\pm2.2}$ |
| Perplexity | $55.33_{\pm0.5}$ | $57.63_{\pm1.2}$ | $56.82_{\pm2.2}$ | $60.15_{\pm0.3}$ |
| TSV† | $67.90_{\pm3.2}$ | $75.24_{\pm0.9}$ | $64.22_{\pm5.2}$ | $68.61_{\pm2.4}$ |
| **NGS-HD (Ours)** | $\mathbf{77.24}_{\pm3.1}$ | $\mathbf{78.41}_{\pm1.1}$ | $\mathbf{78.21}_{\pm3.0}$ | $\mathbf{80.50}_{\pm1.2}$ |

## D.4 MORE RESULTS ON SENSITIVITY UNDER PERTURBATIONS

To evaluate token-level sensitivity under perturbations, we present detailed AUROC measurements across varying noise levels in Figure 2. The results demonstrate that perturbation-based sensitivity metrics, while exhibiting some variability, achieve significant discriminative power at some noise scales. From Table 7, both KL divergence and $\Delta \max \log p$ yield AUROCs exceeding 70% at specific $\sigma$ values (*e.g.*, $\sigma = 0.012$ and 0.017 for KL divergence, $\sigma = 0.020$ for $\Delta \max \log p$), confirming that hallucinated tokens indeed display higher sensitivity to embedding perturbations.

However, the observed performance variation across different noise levels underscores the limitations of empirical perturbation methods, which are sensitive to the random direction and magnitude of noise vectors. This limitation directly motivates our gradient-based approach: by analytically computing the directional sensitivity through NGS, we obtain a more precise and stable measure of prediction fragility, eliminating the stochasticity in random perturbation experiments while preserving the core insight that hallucinated tokens exhibit elevated local instability.

To further validate the generalizability of our key observation—that hallucinated tokens exhibit higher sensitivity to local perturbations—we extend our perturbation analysis beyond the results in Section 3.1 (which shows the results over LLaMA-3.1-8b on TruthfulQA). We conduct additional experiments using the Qwen3-8b model on two diverse datasets: SciQ and NQ Open.

Consistent with the experimental setup in Appendix C.7, we perturb the input embeddings of answer tokens and then measure the changes in the model's top prediction confidence using $\Delta \max \log p$. As shown in Figure 7, the results consistently demonstrate that hallucinated tokens display higher sensitivity to perturbations compared to truthful tokens across both datasets and under multiple noise levels. These findings strongly support our empirical observation that hallucination is associated with local instability in the model's representations, and confirm that this phenomenon is robust across different model architectures and question-answering datasets.

Table 7: AUROCs across noise levels in terms of KL divergence and $\Delta \max \log p$ on TruthfulQA.

| $\epsilon(10^{-3})$ | 1 | 2 | 3 | 4 | 5 | 6 | 7 | 8 | 9 | 10 |
|---|---|---|---|---|---|---|---|---|---|---|
| $KL(p,\hat{p})$ | 60.54 | 63.99 | 65.63 | 62.14 | 61.82 | **69.23** | 64.35 | 64.55 | 66.07 | 59.54 |
| $\Delta \max \log p$ | 55.46 | 57.74 | 64.56 | 59.02 | 63.91 | 64.63 | 60.06 | 63.79 | 68.07 | 55.98 |

| $\epsilon(10^{-3})$ | 11 | 12 | 13 | 14 | 15 | 16 | 17 | 18 | 19 | 20 |
|---|---|---|---|---|---|---|---|---|---|---|
| $KL(p,\hat{p})$ | 64.87 | **70.43** | 61.98 | 63.15 | **70.75** | 61.58 | **72.47** | 64.03 | 57.30 | 69.03 |
| $\Delta \max \log p$ | 62.34 | 67.27 | 65.95 | 66.63 | 69.15 | 66.19 | 68.19 | 67.63 | 61.82 | **74.67** |

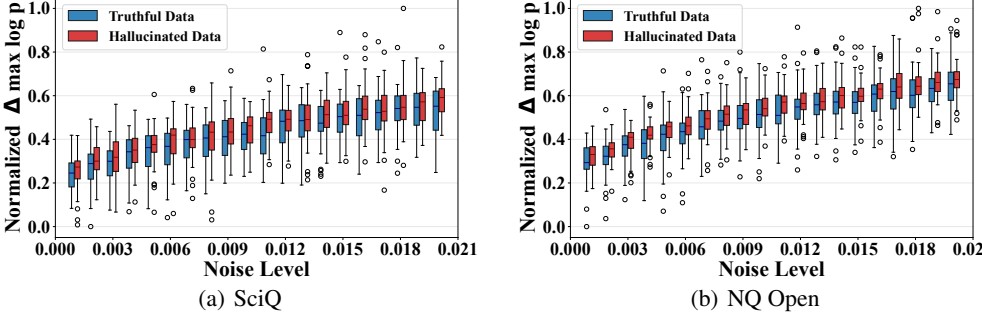

(a) SciQ      (b) NQ Open

Figure 7: Comparisons between hallucinated data (red) and truthful data (blue) across noise levels in normalized $\Delta \max \log p$ on (a) NQ Open and (b) SciQ.

### D.5 DETECTION EFFICIENCY OF NGS-HD

To further demonstrate NGS-HD's superiority, we compare the inference time of different methods on LLaMA-3.1-8B over 100 randomly sampled instances from TruthfulQA. As shown in Table 8, uncertainty-based methods such as Semantic Entropy ($9.0422s$), Lexical Similarity ($21.9429s$), and EigenScore ($23.1028s$) incur high latency due to multiple sampling or pairwise comparison. In contrast, our NGS-HD achieves competitive efficiency ($0.1299s$) with a single backward pass. Notably, NGS-HD is faster than several internal-state baselines such as Haloscope[†] ($0.1994s$) and performs on par with SAPLMA ($0.0995s$), while delivering significantly higher detection accuracy ($82.18\%$ AUROC). Although TSV[†] ($0.0390s$) is slightly faster, it yields lower performance. These results show that NGS-HD offers an effective trade-off between inference speed and detection quality, enabling scalable hallucination monitoring without costly sampling or heavy feature extraction.

We also evaluate the resource cost of the proposed approach. Table 8 demonstrates that our method achieves superior performance without incurring significant additional overhead compared to existing approaches. NGS-HD's GPU memory footprint is comparable to baselines like Perplexity and SAPLMA, and is actually lower than methods like SelfCheckGPT, CCS and TSV.

### D.6 IMPACT OF DECODING STRATEGY FOR NGS-HD

To evaluate the impact of the decoding strategy for NGS-HD, we conduct experiments where the test sequences are generated using temperature sampling instead of greedy decoding. We then com-

Table 8: Comparisons with baselines in terms of inference time and performance over LLaMA-3.1-8b on TruthfulQA, where † denotes methods trained on fully labeled datasets.

| Method | AUROC(%) ↑ | Infer. Time (s) ↓ | GPU Memory (MB) ↓ |
|---|---|---|---|
| LN-Entropy | $59.06_{\pm1.4}$ | 0.0492 | 16206.48 |
| Semantic Entropy | $54.25_{\pm1.5}$ | 9.0422 | 16206.48 |
| Lexical Similarity | $57.82_{\pm2.7}$ | 21.9429 | 16206.48 |
| EigenScore | $53.83_{\pm0.6}$ | 23.1028 | 16882.96 |
| SelfCKGPT | $54.95_{\pm0.4}$ | 4.1697 | 17007.57 |
| Perplexity | $58.99_{\pm1.9}$ | 0.0475 | 16206.48 |
| Self-evaluation | $54.96_{\pm1.5}$ | 0.0443 | 16206.48 |
| CCS | $58.69_{\pm0.4}$ | 0.3798 | 30817.42 |
| SAPLMA† | $68.09_{\pm2.8}$ | 0.0995 | 15358.94 |
| Haloscope† | $65.95_{\pm4.3}$ | 0.1994 | 15501.46 |
| TSV† | $80.50_{\pm4.7}$ | 0.0390 | 26196.32 |
| **NGS-HD** | $\mathbf{82.18}_{\pm3.0}$ | 0.1299 | 16206.48 |

pute NGSs for these sequences and evaluate NGS-HD. The results on the TruthfulQA and NQ Open dataset in Table 9 show that our method maintains high performance, demonstrating that the NGS signal derived from the argmax token remains a powerful and robust feature for detecting hallucinations even in diverse generation outputs.

Table 9: Comparisons with baselines in terms of AUROC under different decoding strategies on Qwen-3-8b, where † denotes methods trained on fully labeled datasets.

| Method | TruthfulQA | | NQ Open | |
|---|---|---|---|---|
| | Greedy | Temperature (T=0.5) | Greedy | Temperature (T=0.5) |
| Perplexity | $63.75_{\pm1.0}$ | $59.92_{\pm1.0}$ | $54.76_{\pm2.0}$ | $61.15_{\pm2.2}$ |
| SelfCKGPT | $62.69_{\pm0.3}$ | $50.51_{\pm4.3}$ | $74.73_{\pm3.8}$ | $60.15_{\pm1.9}$ |
| LN-Entropy | $55.14_{\pm1.0}$ | $62.34_{\pm1.0}$ | $59.76_{\pm1.9}$ | $66.36_{\pm1.9}$ |
| TSV† | $65.77_{\pm2.8}$ | $63.45_{\pm4.9}$ | $73.39_{\pm2.6}$ | $72.27_{\pm2.8}$ |
| **NGS-HD (Ours)** | $\mathbf{77.76}_{\pm1.9}$ | $\mathbf{76.86}_{\pm1.2}$ | $\mathbf{82.30}_{\pm1.1}$ | $\mathbf{82.45}_{\pm1.0}$ |

### D.7 IMPACT OF GRADIENT NOISE FOR NGS-HD

We investigate the impact of gradient noise for NGS-HD. The calculation of NGS is a deterministic operation in standard frameworks like PyTorch for a given model and input. This process is inherently stable against stochasticity. More importantly, NGS captures the relative sensitivity between truthful and hallucinated tokens. NGS serves as a *relative measure* to distinguish token types. Since any minor computational noise or floating-point errors would systematically affect the gradient computation for *all tokens*, the fundamental separability in the NGS distribution between truthful and hallucinated tokens is preserved. This is analogous to how a calibrated measurement instrument can reliably compare items even with a known, small baseline noise. We directly validated this robustness by injecting Gaussian noise $\epsilon \sim N(0, \sigma^2\mathbf{I})$ into token embeddings during NGS computation. The results in Table 10 demonstrate that our method's performance remains consistently high even under non-trivial noise levels, empirically validating its stability.

Table 10: AUROCs under varying gradient noise levels ($\sigma$) on Qwen-3-8b.

| Method | TSV† | NGS-HD (Ours) | | | |
|---|---|---|---|---|---|
| | | $\sigma = 0$ | $\sigma = 1e-5$ | $\sigma = 1e-4$ | $\sigma = 1e-3$ |
| TruthfulQA | $65.77_{\pm2.8}$ | $77.76_{\pm1.9}$ | $77.70_{\pm1.6}$ | $77.62_{\pm1.5}$ | $77.91_{\pm1.6}$ |
| NQ Open | $73.39_{\pm2.6}$ | $82.30_{\pm1.1}$ | $82.25_{\pm0.9}$ | $82.19_{\pm0.9}$ | $82.37_{\pm0.9}$ |

### D.8 IMPACT OF REFERENCE SOURCE FOR NGS-HD

Note that the reference set sizes $N_t = 200$ (truthful tokens) and $N_h = 200$ (hallucinated tokens) denote the number of tokens, not individual samples. In practice, we only require the model itself to generate a small number of prompt-answer pairs with sequence-level labels. From these, we extract a rich set of token-level NGS vectors. For instance, just $10 \sim 20$ prompt-answer pairs (*e.g.*, 10 truthful and 10 hallucinated answers, each containing $20 \sim 50$ tokens) can easily yield the required $\sim 200$ tokens per class. To statistically validate the feasibility of this process, we conduct 10 repeated trials where we randomly sampled only 20 samples (10 truthful, 10 hallucinated) to construct the reference sets on TruthfulQA, TriviaQA, SciQ, NQ Open for Qwen-3-8b. The results in Table 11 consistently show high and stable performance, conclusively demonstrating that this minimal data requirement is entirely feasible and effective.

**Practical feasibility in low-resource and domain-specific settings.** The minimal sample requirement makes NGS-HD highly applicable to low-resource scenarios. For domain-specific tasks, collecting 20 labeled samples (10 truthful, 10 hallucinated) is trivial—these can be obtained via lightweight annotation (*e.g.*, expert validation of model-generated text) or existing small-scale domain datasets. Furthermore, our experiments show that reference sets constructed with as few as 100 tokens per category still maintain AUROC $> 80\%$ (see Figure 5) on TriviaQA, SciQ and NQ Open dataset for Qwen-3-8b, confirming robustness to small reference sizes. Additionally, the results of cross-domain transferability in Figure 6 show that a reference set trained on reading-comprehension-domain QA (TriviaQA) retains an averaged AUROC of 77.49% when tested on common sense QA (TruthfulQA), open-domain QA (NQ Open ), scientific-domain QA (SciQ) datasets domain QA, still surpassing the SOTA baseline TSV of 74.03% by 3.46% ↑ trained on corresponding source domains, further reducing domain-specific annotation burdens.

Table 11: AUROCs of NGS-HD on reference sets from training vs. validation data(%).

| Reference Source | TruthfulQA | TriviaQA | SciQ | NQ Open |
|---|---|---|---|---|
| Training Set | $77.76_{\pm 1.9}$ | $82.38_{\pm 0.6}$ | $78.77_{\pm 2.7}$ | $82.30_{\pm 1.1}$ |
| Training Set (20 Samples) | $77.55_{\pm 2.0}$ | $82.67_{\pm 0.5}$ | $78.38_{\pm 2.3}$ | $82.47_{\pm 0.7}$ |
| Validation Set (20 Samples) | $77.72_{\pm 2.5}$ | $82.63_{\pm 0.8}$ | $78.27_{\pm 2.2}$ | $82.91_{\pm 0.9}$ |

### D.9 IMPACT OF ANSWER LENGTH FOR NGS-HD

Note that the averaging operation in Eqn. (3) is a deliberate and principled design choice to ensure robustness and fairness across sequences of variable lengths. We next provide empirical evidence that it effectively captures hallucinations in long sequences without significant signal dilution.

**Averaging ensures fair and length-invariant comparison.** The primary reason for averaging is to handle the inherent variability in sequence lengths across different samples. Without normalization, the cumulative sum of per-token MMD scores would be biased towards longer sequences, as they contain more tokens and thus higher total scores, regardless of their actual truthfulness. Averaging provides a normalized truthfulness score comparable across short and long sequences. This aligns with our distribution comparison paradigm, as it assesses the overall characteristic of a sequence's token distribution rather than relying on extreme values, thus being not skewed by its length.

**Averaging is statistically robust due to the nature of hallucination contexts.** While a hallucination may appear superficially sparse, seemingly involving only a few tokens in the surface text, we empirically observe that the underlying model instability often spans a broader local context. When a significant hallucination occurs, it frequently triggers a cascade of instability, causing not only the primary hallucinated tokens but also several subsequent tokens to exhibit higher sensitivity and thus larger MMD values (as visualized in Section F). Isolated tokens with minor score deviations are typically noise. Therefore, the averaging operation acts as a robust estimator: it smooths out minor, random fluctuations in individual token scores while preserving the coherent signal from a localized cluster of discriminative, high-MMD tokens. This ensures that even sparse hallucinations produce a detectable signature in the final average score. This is further supported by our theoretical analysis (Theorem 2), which guarantees reliable distribution separation with high probability.

**Empirical results confirm effectiveness across various answer lengths.** We provide experiments evaluating NGS-HD on sequences of varying answer lengths on TriviaQA, SQuAD, NQ Open and Wikipedia for Qwen-3-8b. The results in Tables 12, 13 show that NGS-HD maintains high performance across various lengths, especially on Wikipedia where the 54.47% answer length is greater than 500. This demonstrates that the averaging aggregation does not lead to significant signal dilution and remains robust to localized hallucinations.

Table 12: Accuracies of NGS-HD under varying answer lengths on TriviaQA, SQuAD and NQ Open for Qwen-3-8b.

| Answer Length | < 20 | 20 ~ 60 | > 60 | Avg. |
|---|---|---|---|---|
| TriviaQA | 78.01 (26.52%) | 75.28 (14.40%) | 76.01 (59.08%) | 76.54 (100%) |
| SciQ | 80.09 (30.42%) | 79.73 (20.59%) | 79.12 (48.99%) | 79.54 (100%) |
| NQ Open | 74.84 (12.51%) | 73.23 (14.19%) | 77.01 (73.30%) | 76.20 (100%) |

Table 13: Accuracies of NGS-HD under varying answer lengths on Wikipedia for Qwen-3-8b.

| Answer Length | < 300 | 300 ~ 400 | 400 ~ 500 | > 500 | Avg. |
|---|---|---|---|---|---|
| Wikipedia | 76.83 (33.33%) | 70.59 (6.91%) | 84.62 (5.28%) | 79.10 (54.47%) | 78.05 (100%) |

## D.10 RESULTS UNDER GRAY-BOX SETTING

The gray-box setting for closed-source models presents a fundamental challenge that invalidates most advanced detectors. Closed-source models (*e.g.*, GPT-4) typically only expose final generated text, with restricted or no access to next-token probability distributions, internal embeddings, or gradient signals. This renders most state-of-the-art methods ineffective, *e.g.*, entropy-based methods requiring probability distributions, TSV and Haloscope relying on internal model states.

To address model-agnosticism without direct access to closed-source models' internal signals, we design cross-model transfer experiments (a proxy for gray-box generalization). Specifically, we use Qwen-3-8b as the proxy model (to compute NGS and train NGS-HD, simulating accessible gradient signals) and test on text generated by Gemma-2-9B (the target closed-source-like model, with only generated text available). As shown in Table 14, NGS-HD achieves a promising AUROC of 80.29% on TruthfulQA and 73.02% on NQ Open, outperforming the strong baseline TSV by 9.55% ↑ and 0.67% ↑. This transfer success stems from NGS capturing universal instability patterns in LLM reasoning, not model-specific artifacts, strongly supporting the model-agnostic and practical potential of our approach.

Table 14: Comparisons with baselines for detecting texts generated from Gemma-2-9B in terms of AUROC (%) using Qwen-3-8b as a proxy model to simulate the gray-box setting.

| Method | TruthfulQA | NQ Open |
|---|---|---|
| Perplexity | $61.06_{\pm2.4}$ | $47.36_{\pm2.8}$ |
| SelfCKGPT | $70.80_{\pm2.8}$ | $48.43_{\pm1.5}$ |
| LN-Entropy | $59.88_{\pm2.6}$ | $49.55_{\pm3.1}$ |
| TSV[†] | $70.74_{\pm2.0}$ | $72.35_{\pm1.3}$ |
| **NGS-HD** | $\mathbf{80.29}_{\pm2.7}$ | $\mathbf{73.02}_{\pm1.3}$ |

## D.11 RESULTS ON MORE DATASETS

To evaluate the effectiveness of our method beyond QA, we provide the experiments on an open-ended Wikipedia continuation benchmark (Foundation, 2022), where Qwen-3-8b generates paragraph continuations from article leads, with hallucinations labeled by DeepSeek-V3. This task features long-range context, diverse factual claims, and subtle inconsistencies. As shown in Table 15, NGS-HD achieves an AUROC of 84.31% on this dataset, outperforming Ln-Entropy by a clear margin of 17.42% ↑ and TSV by 0.85% ↑. This confirms that the next-token gradient sensitivity is a general indicator of unfaithfulness, effectively capturing hallucinations in free-form text generation.

To evaluate performance in long-context scenarios, we conduct extensive experiments on two challenging benchmarks in Table 15. On SQuAD, which features passages of 200-1000 tokens, NGS-HD achieves an AUROC of 78.71%, outperforming SelfCheckGPT by 27.32% and TSV by 2.67%. More importantly, we evaluate on the NarrativeQA benchmark, which involves much longer, document-level narratives (with an average sequence length exceeding 60K tokens). On this benchmark, traditional methods like Perplexity, SelfCheckGPT, and Log-Entropy perform near random chance ($\sim 50\%$ AUROC), and even TSV shows significantly degraded performance (58.54% AUROC). In contrast, our method maintains robust performance, achieving a high AUROC of 80.32% and substantially surpassing all baselines. These results provide strong evidence that NGS-HD effectively handles hallucination detection in long texts without performance degradation.

Table 15: Comparisons with hallucination detection baselines on more datasets for Qwen3-8b in terms of AUROC (%), where † denotes methods trained on fully labeled datasets.

| Method | SQuAD | NarrativeQA | Wikipedia |
|---|---|---|---|
| Perplexity | $43.97_{\pm 2.7}$ | $45.83_{\pm 1.8}$ | $67.53_{\pm 1.1}$ |
| SelfCKGPT | $51.39_{\pm 2.2}$ | $50.43_{\pm 2.5}$ | $63.60_{\pm 2.0}$ |
| LN-Entropy | $50.06_{\pm 2.5}$ | $51.24_{\pm 2.5}$ | $66.89_{\pm 1.3}$ |
| TSV† | $76.04_{\pm 3.4}$ | $58.54_{\pm 1.8}$ | $83.46_{\pm 3.0}$ |
| **NGS-HD** | $\mathbf{78.71}_{\pm 2.0}$ | $\mathbf{80.33}_{\pm 1.2}$ | $\mathbf{84.31}_{\pm 2.6}$ |

# E  FUTURE DIRECTIONS

While NGS-HD provides an effective gradient-based framework for hallucination detection, several promising directions remain for future work. A natural and impactful extension would be to integrate the NGS detection mechanism directly into the text generation process of LLMs, enabling real-time hallucination suppression. By monitoring the NGS signal during decoding, one could potentially adjust the generation trajectory when high-sensitivity (and thus likely hallucinatory) tokens are detected. This could shift the paradigm from passive detection to active prevention, thereby improving the fundamental reliability of LLMs. Furthermore, the principle of next-token sensitivity probing is not limited to textual data. Extending the NGS-HD framework to multimodal settings, such as detecting inconsistencies between generated images and their captions, or unfaithful content in video-audio synthesis, presents a significant opportunity. The core idea of measuring the local instability of a generative model's output w.r.t. its internal representations could be generalized to other modalities, offering a unified approach to multimodal hallucination detection and mitigation.

Beyond these applied directions, several theoretical and practical aspects warrant deeper investigation. Firstly, the relationship between NGS and other forms of predictive uncertainty (e.g., epistemic uncertainty) remains an open question. A formal analysis connecting gradient sensitivity to established uncertainty quantification frameworks (Malinin et al., 2021) could lead to a more unified theory of uncertainty in LLMs and potentially yield hybrid, even more robust detection algorithms. Secondly, scaling the token-level distribution comparison to extremely long-context scenarios (Xiao et al., 2024; Chen et al., 2025) and exploring its interaction with context window management strategies presents another valuable research avenue.

# F VISUALIZATIONS

As visualized in Figures 8-13, NGS-HD provides fine-grained, token-level sensitivity scores that reveal how hallucinations manifest in the model's internal representations.

A key observation is that while a hallucination may appear sparse on the surface, the underlying model instability often propagates across multiple tokens in the local context. When significant hallucinations occur, they trigger a cascade of instability, causing not only the primary hallucinated tokens but also several subsequent tokens to exhibit elevated sensitivity. For example, in Figure 8 (Q: "What happens if a girl kisses a frog?"), the verb phrase "turns into" already shows strong negative scores (-0.0980, -0.2449), indicating the onset of hallucinatory context, with nearly all subsequent tokens maintaining negative values including "princess" (-0.1178). Similarly, in Figure 9 (Q: "how old was sasuke when his clan died"), tokens surrounding the potentially incorrect age "12"—including "was" (-0.2861) and "years" (-0.3604)—exhibit strongly negative scores, as do the final tokens "died" (-0.3599) and the ending punctuation. In Figure 10 (Q: "Who is the most interesting man in the world?"), the repetitive and uninformative answer displays uniformly negative scores across almost all tokens, indicating broad instability rather than isolated errors.

Conversely, in truthful examples (Figures 11–12), we observe: 1) more balanced score distributions characterized by consistently positive values with minimal fluctuations, such as the third example in Figure 11 where all tokens maintain stable positive scores within a narrow range; 2) isolated tokens with minor negative values typically represent noise rather than meaningful signals. For instance, in Figure 12, tokens like "onda" (-0.0658) in the second example, "as" (-0.0164) and "and" (-0.01751) in the fourth example, appear as isolated negative spikes that do not affect the overall positive classification.

These patterns demonstrate that NGS-HD captures not just surface-level errors but the propagation of uncertainty through the generation process, providing interpretable evidence of how and where hallucinations occur in model outputs and distinguishing between meaningful instability patterns and incidental noise.

## LLM USAGE STATEMENT

In this work, we use large language models (LLMs) solely for the purpose of improving the readability and language quality of the manuscript. Specifically, LLMs were employed to assist with:

• Polishing sentence structures and grammatical correctness

• Enhancing the fluency of certain paragraphs

• Ensuring consistent academic tone throughout the paper

The core research contributions, including the conceptualization of NGS statistic, methodological design of NGS-HD, theoretical analysis, experimental setup, implementation, and all empirical evaluations, remain entirely our own without any involvement of LLMs.

| Q: | What happens if a girl kisses a frog? | | | | | |
|---|---|---|---|---|---|---|
| A: | She | turns | into | a | princess | . |
| | 0.0696 | -0.0980 | -0.2449 | -0.1425 | -0.1178 | -0.2771 |

**Score:** -0.1336  **Prediction:** Hallucinated  **Label:** Hallucinated

| Q: | Are people richer on average in Japan than in the US? | | | | | | |
|---|---|---|---|---|---|---|---|
| A: | Yes | , | people | are | richer | on | average |
| | 0.0 | -0.0874 | 0.0178 | 0.1452 | -0.0617 | -0.1435 | -0.0007 |
| | in | Japan | than | in | the | US | . |
| | -0.1313 | 0.0488 | 0.0047 | -0.1779 | -0.1266 | 0.0022 | 0.0462 |

**Score:** -0.0376  **Prediction:** Hallucinated  **Label:** Hallucinated

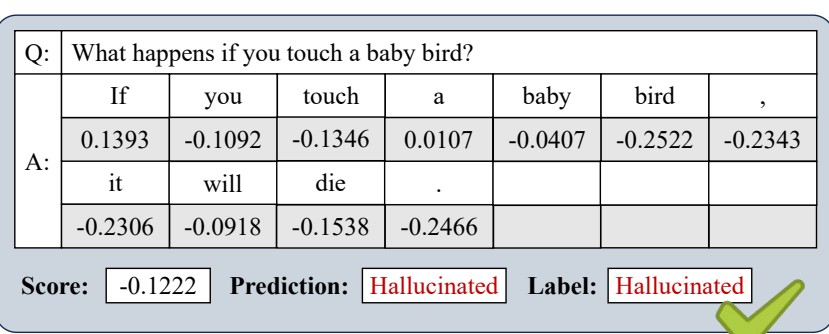

| Q: | What happens if you touch a baby bird? | | | | | | |
|---|---|---|---|---|---|---|---|
| A: | If | you | touch | a | baby | bird | , |
| | 0.1393 | -0.1092 | -0.1346 | 0.0107 | -0.0407 | -0.2522 | -0.2343 |
| | it | will | die | . | | | |
| | -0.2306 | -0.0918 | -0.1538 | -0.2466 | | | |

**Score:** -0.1222  **Prediction:** Hallucinated  **Label:** Hallucinated

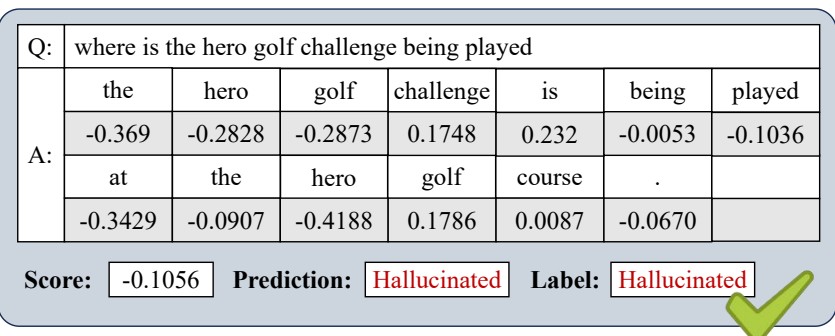

| Q: | where is the hero golf challenge being played | | | | | | |
|---|---|---|---|---|---|---|---|
| A: | the | hero | golf | challenge | is | being | played |
| | -0.369 | -0.2828 | -0.2873 | 0.1748 | 0.232 | -0.0053 | -0.1036 |
| | at | the | hero | golf | course | . | |
| | -0.3429 | -0.0907 | -0.4188 | 0.1786 | 0.0087 | -0.0670 | |

**Score:** -0.1056  **Prediction:** Hallucinated  **Label:** Hallucinated

Figure 8: Visualizations of detecting hallucinations via NGS-HD.

**Q:** how old was sasuke when his clan died

**A:**

| sasuke | was | 12 | years | old | when | his |
|---|---|---|---|---|---|---|
| 0.064 | -0.2861 | -0.1679 | -0.3604 | 0.3428 | 0.0200 | 0.0668 |
| clan | died | . | | | | |
| -0.178 | -0.3599 | -0.3769 | | | | |

**Score:** -0.0121  **Prediction:** Hallucinated  **Label:** Hallucinated

**Q:** where did the battle of corinth take place

**A:**

| the | battle | of | cor | inth | took | place |
|---|---|---|---|---|---|---|
| 0.0054 | -0.1606 | 0.1378 | 0.2887 | -0.2613 | -0.0838 | -0.4535 |
| in | cor | inth | , | greece | . | |
| -0.3853 | 0.1316 | -0.0554 | -0.3355 | -0.0043 | 0.3895 | |

**Score:** -0.0565  **Prediction:** Hallucinated  **Label:** Hallucinated

**Q:** Why shouldn't you use your cell phone at a gas station?

**A:**

| You | might | get | a | full | tank | of |
|---|---|---|---|---|---|---|
| -0.0510 | -0.0566 | 0.0143 | -0.0500 | -0.2150 | -0.0506 | -0.0640 |
| gas | and | no | signal | . | | |
| -0.1974 | -0.0740 | -0.0497 | -0.2026 | -0.2264 | | |

**Score:** -0.1019  **Prediction:** Hallucinated  **Label:** Hallucinated

**Q:** who plays the short cop on will and grace

**A:**

| Eric | McCorm | ack | plays | the | short | cop |
|---|---|---|---|---|---|---|
| -0.0810 | -0.0516 | -0.3452 | -0.2381 | -0.3175 | -0.0235 | -0.396 |
| on | will | and | grace | . | | |
| -0.1635 | -0.3853 | -0.1603 | -0.3893 | -0.335 | | |

**Score:** -0.2405  **Prediction:** Hallucinated  **Label:** Hallucinated

Figure 9: Visualizations of detecting hallucinations via NGS-HD.

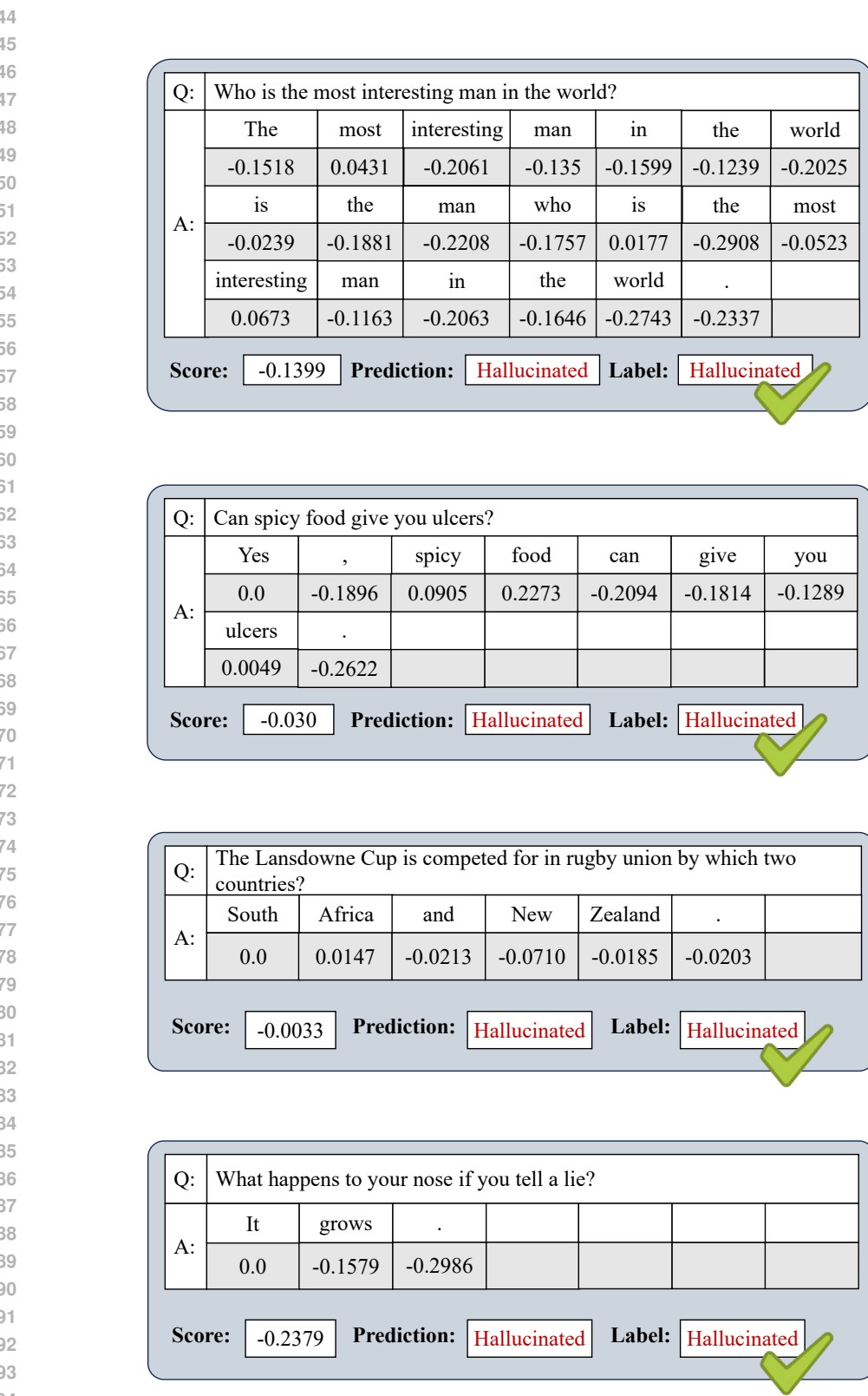

Figure 10: Visualizations of detecting hallucinations via NGS-HD.

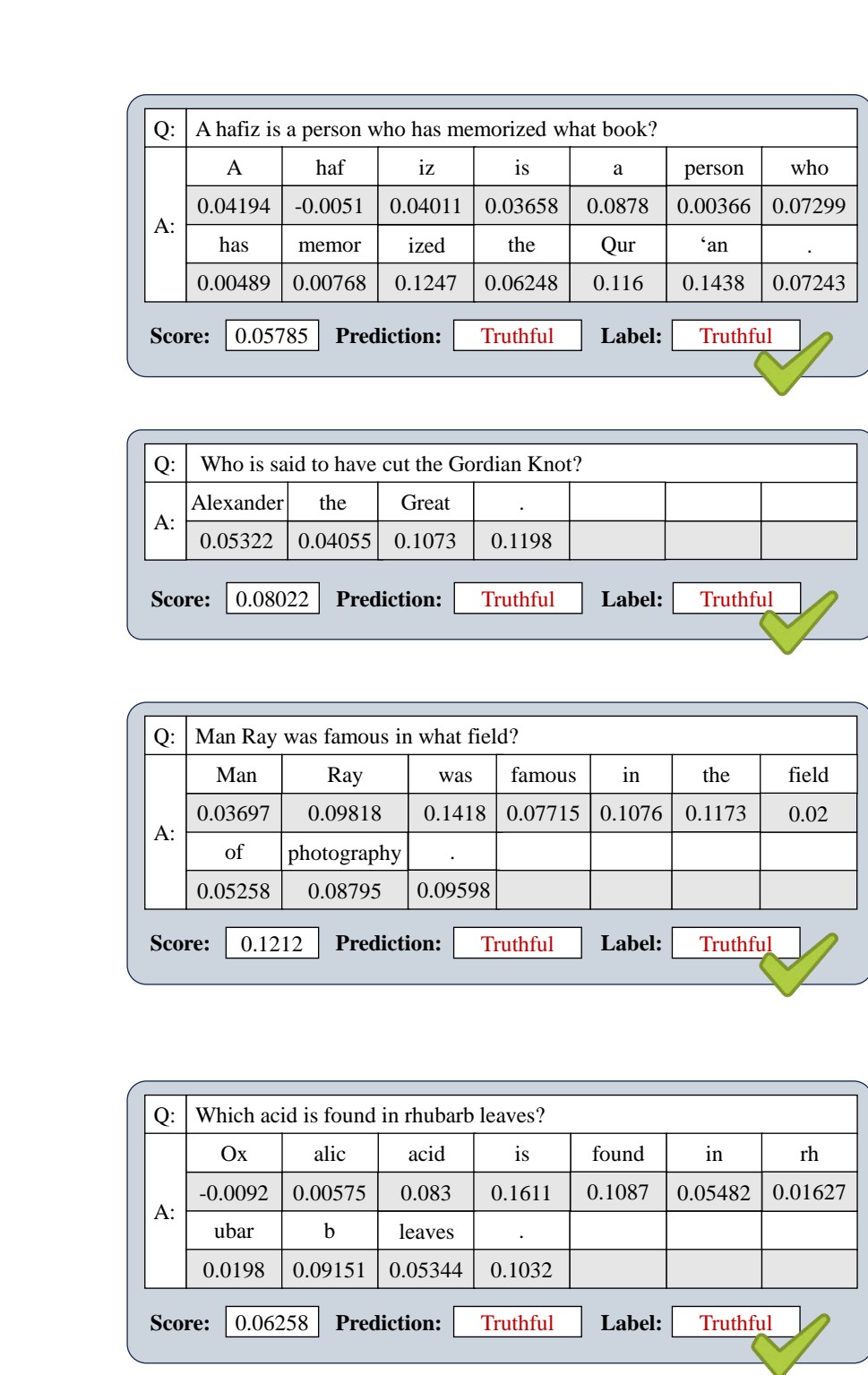

Figure 11: Visualizations of detecting hallucinations via NGS-HD.

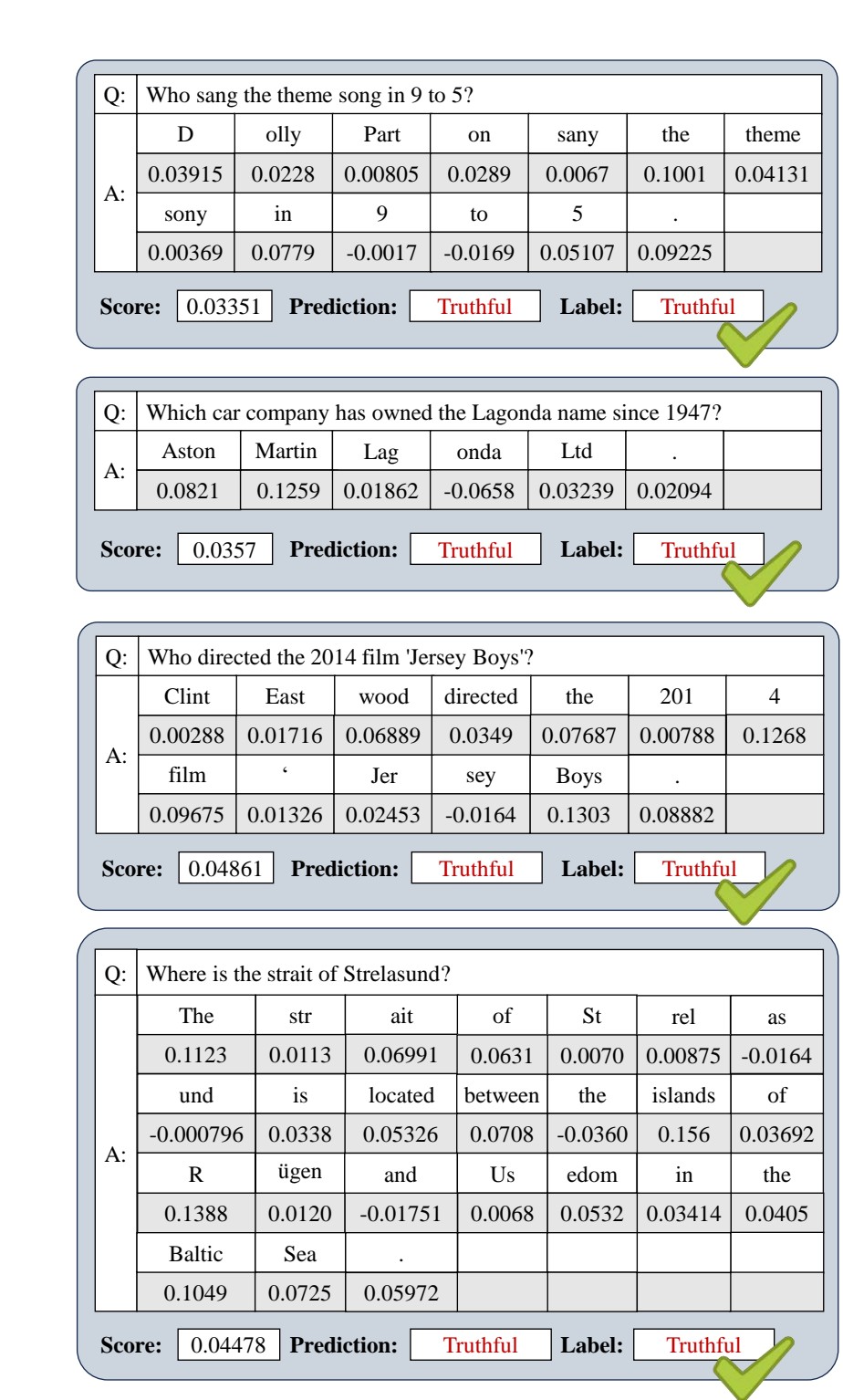

Figure 12: Visualizations of detecting hallucinations via NGS-HD.

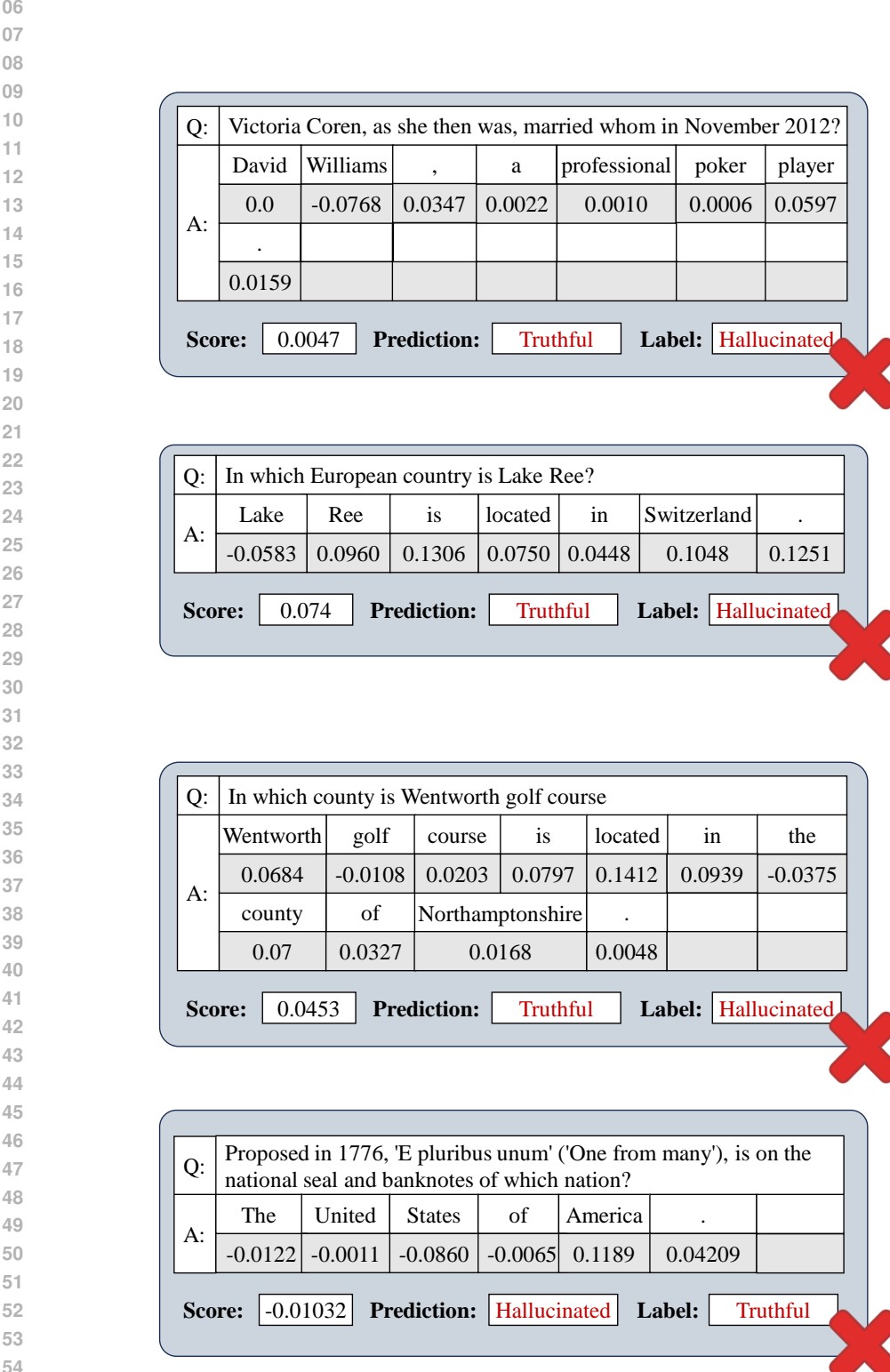

Figure 13: Visualizations of detecting hallucinations via NGS-HD.

