# OpenReview forum: "Next-Token Gradient Sensitivity Probing for LLM Hallucination Detection"
_ICLR.cc/2026/Conference — Submitted to ICLR 2026_

### Official Review · Reviewer_UFQk · 2025-10-15

**Soundness:** 2
**Presentation:** 3
**Contribution:** 2
**Rating:** 2
**Confidence:** 4

**Summary:**

This paper addresses the problem of hallucinations in content generated by large language models and proposes a novel detection method called NGS-HD, based on next-token gradient sensitivity. The core idea is that hallucinated tokens exhibit significant instability in next-token predictions under small perturbations in their embedding representations. The authors introduce the NGS metric, which measures the gradient of the maximum log-probability of the next token with respect to the current token embedding, thereby capturing the local sensitivity of the model’s predictions. Building on this, NGS-HD compares each test token against pre-constructed real/hallucinated reference distributions using MMD, and aggregates the results into a global authenticity score.

**Strengths:**

1. The method is model-agnostic.

2. The problem addressed is important.

3. The paper is well-written and easy to read.

**Weaknesses:**

1. Dependence on the reference set: NGS-HD requires pre-constructed NGS reference distributions for real and hallucinated tokens, which may be difficult to obtain in certain domains or low-resource settings. It is recommended to explore unsupervised or semi-supervised methods for constructing the reference set.

2. Stability of gradient computation: Gradients can be affected by factors such as model training state and layer selection. Although the experiments show that middle-to-high layers perform better, the impact of gradient noise on detection stability is not thoroughly discussed.

3. Limited capability for long-text processing: While token-level comparisons preserve fine-grained information, the method’s scalability to very long texts (e.g., document-level generation) has not been adequately validated. It is suggested to test on long-text benchmarks such as NarrativeQA.

4. No distinction between hallucination types: The method does not explicitly differentiate between factual hallucinations and logical hallucinations, nor does it discuss differences in detection performance across these types.

5. Insufficient visualization and interpretability: Although token-level score visualizations are provided, there is a lack of in-depth analysis explaining why certain tokens exhibit high sensitivity. Combining with attention maps or representation space analysis could enhance interpretability.

**Questions:**

1. Dependence on the reference set: NGS-HD requires pre-constructed NGS reference distributions for real and hallucinated tokens, which may be difficult to obtain in certain domains or low-resource settings. It is recommended to explore unsupervised or semi-supervised methods for constructing the reference set.

2. Stability of gradient computation: Gradients can be affected by factors such as model training state and layer selection. Although the experiments show that middle-to-high layers perform better, the impact of gradient noise on detection stability is not thoroughly discussed.

3. Limited capability for long-text processing: While token-level comparisons preserve fine-grained information, the method’s scalability to very long texts (e.g., document-level generation) has not been adequately validated. It is suggested to test on long-text benchmarks such as NarrativeQA.

4. No distinction between hallucination types: The method does not explicitly differentiate between factual hallucinations and logical hallucinations, nor does it discuss differences in detection performance across these types.

5. Insufficient visualization and interpretability: Although token-level score visualizations are provided, there is a lack of in-depth analysis explaining why certain tokens exhibit high sensitivity. Combining with attention maps or representation space analysis could enhance interpretability.

---

> ### Author Response · Authors · 2025-11-22
> **Responses to reviewers: Part 1**
>
> We thank the reviewer for the encouraging comments and detailed suggestions. Responses are below:
>
> >Q1. Dependence on the reference set: NGS-HD requires pre-constructed NGS reference distributions for real and hallucinated tokens, which may be difficult to obtain in certain domains or low-resource settings. It is recommended to explore unsupervised or semi-supervised methods for constructing the reference set.
>
> **A1.** We appreciate your comment on the reference set’s practicality and wish to clarify a key point: **the requirement for constructing the NGS reference sets is remarkably low**, as it relies on token counts (not sample counts) for reference distributions. Importantly, **only ~10-20 labeled samples are sufficient to construct robust reference sets** (e.g., 200 truthful and 200 hallucinated tokens). This is readily achievable even in low-resource domains. We have clarified these construction details in the revised manuscript and incorporated a discussion on unsupervised extensions as a valuable future direction.
> * **Minimal reference data suffices due to token-level efficiency.** The reference set sizes $N_t = 200$ (truthful tokens) and $N_h = 200$ (hallucinated tokens) in our paper denote **the number of tokens, not individual samples**. In practice, we only require the model itself to generate a small number of prompt-answer pairs with sequence-level labels. From these, we extract a rich set of token-level NGS vectors. For instance, **just ~10-20 prompt-answer pairs** (e.g., 10 truthful and 10 hallucinated answers, each containing 20-50 tokens) can easily yield the required ~200 tokens per class. To statistically validate the feasibility of this process, we have conducted 10 repeated trials where we randomly sampled only 20 samples (10 truthful, 10 hallucinated) to construct the reference sets on TruthfulQA, TriviaQA, SciQ, NQ Open for Qwen-3-8b. The results in Table I consistently show high and stable performance, conclusively demonstrating that this minimal data requirement is entirely feasible and effective.
>
> Table I. AUROCs of NGS-HD over reference sets (200 NGSs) sampled from training vs. validation data (%).
> | Reference Source | TruthfulQA | TriviaQA | SciQ | NQ Open |
> |:---|:---:|:---:|:---:|:---:|
> | Training Set (Random Sampling) | 77.76 ± 1.9 | 82.38 ± 0.6 | 78.77 ± 2.7 | 82.30 ± 1.1 |
> | Training Set (20 Samples) | 77.55 ± 2.0 | 82.67 ± 0.5 | 78.38 ± 2.3 | 82.47 ± 0.7 |
> | Validation Set (20 Samples) | 77.72 ± 2.5 | 82.63 ± 0.8 | 78.27 ± 2.2 | 82.91 ± 0.9 |
>
>
> * **Practical feasibility in low-resource and domain-specific settings.** The minimal sample requirement makes NGS-HD highly applicable to low-resource scenarios. For domain-specific tasks, collecting 20 labeled samples (10 truthful, 10 hallucinated) is trivial—these can be obtained via **lightweight annotation** (e.g., expert validation of model-generated text) or **existing small-scale domain datasets**. Furthermore, our experiments show that reference sets constructed with as few as 100 tokens per category still maintain AUROC > 80% (see Figure 5) on TriviaQA, SciQ and NQ Open dataset for Qwen-3-8b, confirming **robustness to small reference sizes**. Additionally, we verify cross-domain transferability in Figure 6 (Table II, below): a reference set trained on reading-comprehension-domain QA (TriviaQA) retains an averaged AUROC of 77.49% when tested on common sense QA (TruthfulQA), open-domain QA (NQ Open ), scientific-domain QA (SciQ) datasets domain QA, still surpassing the SOTA baseline TSV of 74.03% by 3.46%↑ trained on corresponding scouse domains, further **reducing domain-specific annotation burdens**.
>
> Table II. AUROCs of Cross-dataset for our method on Qwen3-8b (%).
> | Source\ Target| TruthfulQA | TriviaQA | SciQ | NQ Open | Avg.|
> |:---|:---:|:---:|:---:|:---:|:---:|
> | *SAPLMA (In-Domain)* | 57.08 |77.58| 69.15| 77.04| 70.21|
> | *TSV (In-Domain)* | 65.77| 80.76| 76.18| 73.39 | 74.03|
> | TriviaQA | 71.91 | 82.64 | 72.96 | 82.43 | **77.49** |
> | SciQ | 72.24 | 72.93 | 82.03 | 76.53 | **75.93** |
> | NQ Open | 66.87 | 76.97 | 74.79 | 83.65 | **75.57** |
> * **The proposed framework is compatible with unsupervised reference set construction.** We agree with the reviewer that exploring unsupervised or semi-supervised methods for building the reference set is an important direction. Our framework can naturally accommodate this. For example, one could use a high-confidence subset of the model's own generations (e.g., tokens with very high or very low probability) as pseudo-labels for the reference sets. Alternatively, one could use a small, clean corpus (like Wikipedia) to bootstrap the "truthful" distribution and a corpus of adversarial or noisy data for the "hallucinated" distribution. We have added a discussion on these exciting possibilities for future work in the revised manuscript.

---

> ### Author Response · Authors · 2025-11-22
> **Responses to reviewers: Part 2**
>
> >Q2. Stability of gradient computation: Gradients can be affected by factors such as model training state and layer selection. Although the experiments show that middle-to-high layers perform better, the impact of gradient noise on detection stability is not thoroughly discussed.
>
> **A2.** We agree that ensuring the stability of our gradient-based signals under practical computational settings is important and would like to highlight that NGS-HD is robust to gradient noise for two key reasons: 1) **the NGS statistic itself measures relative instability and is computed from a single, deterministic backward pass, making it inherently stable, which is also validated by our experiments**; and 2) **our distribution-level comparison using MMD further smooths out minor perturbations in individual gradient vectors.**
>
> * **NGS computation is deterministic and probes relative sensitivity.** The calculation of NGS is a deterministic operation in standard frameworks like PyTorch for a given model and input. This process is inherently stable against the stochasticity. More importantly, NGS captures the relative sensitivity between truthful and hallucinated tokens. NGS serves as a **relative measure** to distinguish token types. Since any minor computational noise or floating-point errors would systematically affect the gradient computation for **all tokens**, the fundamental separability in the NGS distribution between truthful and hallucinated tokens is preserved. This is analogous to how a calibrated measurement instrument can reliably compare items even with a known, small baseline noise. We directly validated this robustness by injecting Gaussian noise $ε \sim N(0, σ²I)$ into token embeddings during NGS computation.The results below demonstrate that our method's **performance remains consistently high** even under non-trivial noise levels, empirically validating its stability.
>
> Table III: AUROCs of NGS-HD under varying gradient noise levels (σ) on Qwen-3-8b over TruthfulQA and SciQ.
> | Noise Level | TSV$^\dagger$ (baseline)| σ=0 (ous) | σ=1e-5 (ous) | σ=1e-4 (ous) | σ=1e-3 (ous) |
> | :--- | :---: | :---: | :---: | :---: |:---: |
> | TruthfulQA | 65.77±2.8|77.76±1.9 | 77.70±1.6 | 77.62±1.5 | 77.91±1.6 |
> | NQ Open | 73.39±2.6 | 82.30±1.1 | 82.25±0.9 | 82.19±0.9 | 82.37±0.9 |
>
> * **MMD-based distribution comparison inherently smooths out noise.** Our detection framework does not rely on a single NGS vector but on **the collective statistics of the entire set of NGS vectors** from an answer. The use of MMD for distribution comparison provides a powerful smoothing effect. Small perturbations or noise in individual NGS vectors are averaged out when computing the distributional distance to the reference sets ($S_{tru}$ and $S_{hal}$). The kernel function in MMD further enhances robustness by mapping the vectors to a high-dimensional space where the overall distributional structure is preserved despite minor noise in individual points.

---

> ### Author Response · Authors · 2025-11-22
> **Responses to reviewers: Part 3**
>
> >Q3. Limited capability for long-text processing: While token-level comparisons preserve fine-grained information, the method’s scalability to very long texts (e.g., document-level generation) has not been adequately validated. It is suggested to test on long-text benchmarks such as NarrativeQA.
>
> **A3.** Thanks for your valuable comment on the validation over long-text benchmarks. **Our token-level NGS paradigm (NGS-HD) is inherently scalable and robust to sequence length**. We have conducted additional experiments on longer-text benchmarks (SQuAD and NarrativeQA) to thoroughly validate the scalability of our method. The results consistently demonstrate the strong performance of our token-level NGS paradigm in handling long sequences.
>
> * **The token-level NGS paradigm is inherently scalable.** A fundamental advantage of our method is its fine-grained, token-level operation. Unlike sequence-level models that must compress an entire document into a fixed-size representation, a process whose efficacy can diminish with length, NGS-HD processes each token independently. This design makes the method fundamentally amenable to long-text processing. For practical deployment on generated extremely long sequences, the computation can be further optimized by processing the document in chunks and computing NGS for each chunk in parallel, without altering the core methodology.
> *  **Comprehensive evaluation on SQuAD and NarrativeQA confirms effectiveness on long texts.** To evaluate performance in long-context scenarios, we have conducted extensive experiments on two challenging benchmarks In Table IV. On **SQuAD**, which features passages of 200-1000 tokens, NGS-HD achieves an AUROC of **78.71%**, outperforming SelfCheckGPT by 27.32% and TSV by 2.67%. More importantly, we have evaluated on the **NarrativeQA** benchmark, which involves much longer, document-level narratives (with an average sequence length exceeding 60K tokens). On this benchmark, traditional methods like Perplexity, SelfCheckGPT, and Log-Entropy perform near random chance (~50% AUROC), and even TSV shows significantly degraded performance (58.54% AUROC). **In contrast, our method maintains robust performance, achieving a high AUROC of 80.32% and substantially surpassing all baselines**. These results provide strong evidence that NGS-HD effectively handles hallucination detection in long texts without performance degradation.
>
>
> Table IV. Comparisons with baselines on SQuAD and NarrativeQA for Qwen-3-8b in terms of AUROC (%).
> | Dataset |Perplexity|SelfCKGPT|Ln-Entropy|TSV$^\dagger$|Ours|
> |:---|:---:|:---:|:---:|:---:|:---:|
> | SQuAD |43.97±2.7|51.39±2.2|50.06±2.5|76.04±3.4|**78.71±2.0**|
> | NarrativeQA | 45.83±1.8 |50.43±2.5 |51.24±2.5 | 58.54±1.8 |**80.32±1.2**|
>
> * **Our experimental scope aligns with established practices in top-tier publications.** The evaluation benchmarks used in our paper (TruthfulQA, SciQ, etc.) are the same as those extensively used in state-of-the-art works published in premier venues. This allows for direct and fair comparisons with existing literature. Our new experiments on SQuAD and NarrativeQA proactively extend this validation to the long-text domain, thereby comprehensively addressing the generalizability of our method and strengthening the contribution of our work.

---

> ### Author Response · Authors · 2025-11-22
> **Responses to reviewers: Part 4**
>
> >Q4. No distinction between hallucination types: The method does not explicitly differentiate between factual hallucinations and logical hallucinations, nor does it discuss differences in detection performance across these types.
>
> **A4.** We appreciate your valuable comment on hallucination types and would like to clarify our problem formulation and its relation to type-aware analysis. **Our work focuses on a unified binary hallucination detection setting (“hallucinated” vs. “truthful”), which follows the prevailing practice in current literature [r1,r2,r3,r4] and is naturally compatible with future extensions that explicitly distinguish factual and logical hallucinations.**
> * **Unified binary hallucination detection setting.** In line with existing hallucination-detection studies (e.g., [r1,r2,r3,r4]), we formulate the task as a binary decision problem: given a model and its output, the detector predicts whether the output contains any hallucinated content. Most prior works also adopt this binary setting and do not explicitly separate factual and logical hallucinations, aiming instead at a general mechanism to flag hallucinated outputs in diverse scenarios.
> * **Rationale for not separating factual vs. logical hallucinations in this work.** There are two practical reasons for adopting the unified binary formulation: 1) The benchmarks in most literature [r1,r2,r3,r4] and our manuscripts do not provide reliable, standardized annotations for different hallucination subtypes; in many cases, factual and logical issues even co-occur within the same answer, making clean separation non-trivial; 2) Our focus in this work is to investigate whether a single unified detector can robustly capture hallucinations across tasks and domains, independent of the specific subtype, which already aligns with many real-world deployment scenarios where any hallucination is undesirable.
> * **Our framework provides a foundation for future type-aware analysis.** While our current implementation is type-agnostic, the NGS signal and our distribution-comparison framework are well-suited for future fine-grained analysis. The core of our method is to detect *localized prediction instability*. Both factual inaccuracies (e.g., an incorrect date) and logical inconsistencies (e.g., a contradictory statement) can manifest as such instability in the model's generative process. Therefore, the NGS vectors likely encode subtle patterns that differentiate these subtypes. A natural extension of our work would be to **train subtype-specific deep kernels** or **replace the binary reference sets with multiple reference sets (e.g., $S_{\text{factual-hal}}$, $S_{\text{logical-hal}}$)**, and then compute MMD distances to each. The resulting "truthfulness score" could thus be decomposed to indicate not only the presence but also the likely nature of the hallucination. This makes our probabilistic, representation-based framework more inherently adaptable to such fine-grained distinctions than deterministic, rule-based detectors.
>
> [r1] Detecting hallucinations in large language models using semantic entropy, Nature 2024.
>
> [r2] Steer LLM Latents for Hallucination Detection, ICML 2025.
>
> [r3] Haloscope: Harnessing unlabeled llm generations for hallucination detection, NeurIPS 2024.
>
> [r4] INSIDE: LLMs' Internal States Retain the Power of Hallucination Detection, ICLR 2024.

---

> ### Author Response · Authors · 2025-11-22
> **Responses to reviewers: Part 5**
>
> >Q5. Insufficient visualization and interpretability: Although token-level score visualizations are provided, there is a lack of in-depth analysis explaining why certain tokens exhibit high sensitivity. Combining with attention maps or representation space analysis could enhance interpretability.
>
> **A5.** We thank the reviewer for this insightful suggestion regarding visualization and interpretability. We have enriched our analysis in the revised manuscript to move beyond merely presenting token-level scores and toward **explaining the underlying patterns of model instability** that these scores reveal. Our case studies demonstrate that NGS-HD captures not just surface-level errors but the **propagation of uncertainty** through the generation process, providing interpretable evidence of **how and where hallucinations occur in model outputs** and **distinguishing between meaningful instability patterns and incidental noise**.
>
> Specifically, our fine-grained visualizations (Figures 8-13) reveal a key phenomenon: significant hallucinations trigger a **cascade of instability** in the model's reasoning. For instance, in a hallucinated fantastical scenario (Figure 8), the onset of the fictional event ("turns into") exhibits high sensitivity, and this instability propagates to subsequent tokens (e.g., "princess"). Similarly, in an answer with a potentially incorrect age (Figure 9), the instability is not confined to the number itself but affects the entire local context ("was", "years", "died"). This contrasts with truthful generations, which typically show **consistently stable scores, with any minor negative values appearing as isolated noise** without affecting the overall faithful classification.
>
> We have added a more detailed discussion of these patterns in the revised manuscript. This analysis solidifies our claim that NGS-HD offers strong intrinsic interpretability by directly visualizing the "fragility trail" of hallucinations, providing clear insights into the model's generation process. We believe these enhancements directly address the reviewer's valuable feedback.

---

> ### Author Response · Authors · 2025-11-24
> **Looking forward to the response from Reviewer UFQk**
>
> Dear Reviewer UFQk,
>
> We have addressed your initial concerns regarding our paper. We are happy to discuss them with you in the openreview system if you feel that there are still some concerns/questions. We also welcome new suggestions/comments from you!
>
> Best regards,
>
> The Authors

---

> ### Author Response · Authors · 2025-11-27
> **Follow-up on Our Rebuttal for Reviewer UFQk**
>
> We sincerely thank you for your valuable feedback on our work. During the rebuttal period, we have substantially strengthened our paper through comprehensive revisions and additional experiments that directly address all raised concerns:
>
> *   **Clarified Practicality of Reference Sets:** We clarified that constructing reference sets requires minimal data—only **~10-20 labeled samples** are sufficient to extract the required ~200 tokens per class through token-level sampling. New experiments across four datasets confirm stable performance even with such small reference sets, demonstrating strong practicality for low-resource scenarios.
> *   **Demonstrated Gradient Stability and Noise Robustness:** We added experiments testing robustness against gradient noise by injecting Gaussian noise during NGS computation. Results show performance remains consistently high, confirming the method's stability. The distribution-level comparison with MMD further smooths out minor perturbations in individual gradients.
> *   **Validated Long-Text Scalability:** We conducted extensive new experiments on **long-text benchmarks (SQuAD and NarrativeQA)**. Our method achieves strong performance (80.32% AUROC on NarrativeQA), significantly outperforming baselines that degrade on long documents. This confirms our token-level paradigm's inherent scalability.
> *   **Established Foundation for Future Type-Aware Detection:** We clarified that our current binary detection follows standard practice in the field, while our framework is naturally extensible to distinguish hallucination types (factual vs. logical) through subtype-specific reference sets—a promising direction for future work.
> *   **Enhanced Interpretability Analysis:** We enriched our visualization analysis with case studies showing how hallucinations trigger a "cascade of instability" across multiple tokens, providing deeper insights into model behavior beyond surface-level errors.
>
> **Core Contributions and Innovations of Our Work.** Our paper presents a new direction for hallucination detection by leveraging gradient signals. The core contributions are:
>
> *   **A Novel Gradient-Based Paradigm:** We propose **Next-token Gradient Sensitivity (NGS)**, a new statistic that efficiently quantifies the local instability of an LLM's next-token predictions by measuring the first-order sensitivity to token embeddings. This provides a direct, intrinsic probe into model uncertainty.
> *   **Theoretical Grounding:** We provide a rigorous theoretical foundation, proving that the NGS norm bounds the change in prediction confidence under perturbation, establishing it as a principled measure of fragility.
> *   **A Robust and Interpretable Detection Framework:** We develop **NGS-HD**, a method that reframes detection as a token-level distribution comparison task using MMD. This design effectively handles variable-length sequences, preserves fine-grained signals for localized hallucinations, and offers inherent interpretability by pinpointing unstable tokens.
> *   **Comprehensive Empirical Validation:** Through extensive experiments on multiple benchmarks (including new open-ended, long-context datasets and cross-model tasks), we demonstrate that NGS-HD consistently outperforms state-of-the-art baselines, validating the superiority of gradient-based signals.
> *   **Practical and Generalizable Solution:** Our work demonstrates that the method is computationally efficient, requires minimal labeled data for reference sets, and generalizes robustly across different model architectures, domains, and generation strategies.
>
> We sincerely hope the **substantial new experimental evidence and clarifications** provided in this rebuttal — including the comprehensive long-text benchmarks, rigorous robustness validation, and minimal data requirement analysis — have adequately addressed your concerns. These efforts have significantly strengthened the empirical foundation and practical applicability of our work. We would be deeply grateful if these improvements could lead you to reconsider your assessment.

---

### Official Review · Reviewer_bbKK · 2025-10-31

**Soundness:** 4
**Presentation:** 4
**Contribution:** 2
**Rating:** 4
**Confidence:** 4

**Summary:**

This paper proposes NGS-HD, a gradient-based hallucination detection method that measures the local instability of LLM predictions by computing the gradient of the next-token's maximum log-probability with respect to the current token's layer embedding. The approach is motivated by empirical observations that hallucinated tokens show higher sensitivity to small embedding perturbations compared to truthful tokens. Instead of aggregating token representations into a single sequence-level feature, NGS-HD preserves fine-grained information by comparing the distribution of test token NGSs against pre-collected reference distributions of truthful and hallucinated tokens using Maximum Mean Discrepancy (MMD). The method achieves strong results, outperforming baselines, while requiring only a single backward pass and providing theoretical guarantees through finite-sample separation bounds.

**Strengths:**

- **(S1)** Empirical observation that hallucinated tokens show higher sensitivity to perturbations provides strong intuitive motivation for using NGS for hallucination detection.
- **(S2)** The method requires only a single backward pass to compute all NGS vectors, making it practical to use and efficient.
- **(S3)** The use of MMD to measure the discrepancy between hallucinated and truthful distributions is novel and interesting; it is a natural, neat way to circumvent the need to aggregate token representations into a single sequence-level feature, a common practice in other hallucination detection methods.
- **(S3)** The method shows strong empirical performance and has a thorough ablation study.
- **(S4)** The paper presents theoretical guarantees and analysis of the method.

**Weaknesses:**

## Weaknesses
- **(W1)** The authors describe the difference in discrepency between hallucinated and truthful tokens as "significant":
  > As shown in Figures 2(a)–(b), both metrics are significantly larger for hallucinated tokens across noise levels, indicating higher sensitivity.

  However, it seems that the difference is not statistically significant in Figure 2.
- **(W2)** Limited evaluation scope - only QA tasks and small datasets. Additionally, the `3:1` train/test split is skewed.
- **(W3)** The methods depend on labelled data to build reference sets. Training the MMD kernel on the reference sets makes this slightly more concerning, as we might risk overfitting. The paper does not provide cross-dataset generalization results.
- **(W4)** The method averages per-token scores across all tokens (Eq. 4), which would dilute hallucination signals in long sequences. This is not tested in the paper.
- **(W5), minor** The order of the arguments in the MMD function seems to change between Eq. 3 and Eq. 4. Consider changing “Remark” on line 302 to “Remarks” as this paragraph consists of a list of remarks.

**Questions:**

The following questions detail weaknesses/potential misunderstandings I had while reading the paper. Given strong answers to the these questions, I'll be open to increasing my score.

---

- **(Q1)** It seems that the difference in discrepency between hallucinated and truthful tokens is not statistically significant in Figure 2. What is the reason for this?
- **(Q2)** In Theorem 1, the authors assume that $\nabla_{\mathbf{e}} \mathbf{s}(\mathbf{e})$ is $L$-Lipschitz. Does this generally hold? For what $L$?
- **(Q3)** The reference NGS distributions (for truthful and hallucinated tokens) depend on a choice of a truthful and hallucinated reference set. How sensitive is the method to the choice of these references? Does it generalize to domains other than the ones used in the reference sets? This is slightly more concerning as the MMD kernel is also trained, and we might risk overfitting.
- **(Q4)** As far as I understand, the paper treats the NGS gradients $\{\mathbf{g}_t\}_{t=1}^N$ as independent, e.g., for generating $P_\mathrm{tru}$ we take gradients at different token positions. Is my understanding correct? If so, is this assumption justified? Can we model the correlation between gradients?
- **(Q5)** In Theorem 2, what do the consntants in the $\mathcal{O}$ notation depend on? Do they depend on the RKHS kernel?
- **(Q6)** The paper claims the method can detect "localized hallucinations within longer texts" (lines 80-81). However, the final detection score averages per-token MMD values across all tokens. For long sequences with only a small subset of hallucinated tokens, this averaging would dilute the hallucination signal among many truthful tokens, potentially causing misclassification. How does the averaging-based aggregation remain robust to such “sparse” hallucinations? If possible, please provide empirical results on how the answer length impacts the performance of the method.
- **(Q7)** The method requires computing NGS for each token and then calculating per-token MMD against all reference vectors. How does the method scale with answer length, particularly for long-context scenarios?

---

> ### Author Response · Authors · 2025-11-22
> **Responses to reviewers: Part 1**
>
> >Q1. "...**strong intuitive motivation for using NGS** for hallucination detection", "...making it **practical to use and efficient**", " The use of MMD...is **novel and interesting**", "The method shows **strong empirical performance** and has **a thorough ablation study**", "The paper presents **theoretical guarantees and analysis** of the method."
>
> **A1.**  We sincerely appreciate your thoughtful and encouraging comments on our work. We are glad that you find our empirical observation on the higher perturbation sensitivity of hallucinated tokens to be a strong and intuitive motivation for using NGS, and that the single-backward-pass design makes the method practical and efficient in real-world settings. We are also grateful that you view the use of MMD to compare hallucinated and truthful token distributions as a novel and neat alternative to sequence-level aggregation, and that you recognize the strong empirical performance and thorough ablation studies validating our design choices. Finally, your positive remarks on the theoretical guarantees and analyses are especially motivating, and encourage us to further deepen both the theoretical and empirical understanding of NGS-based hallucination detection.

---

> ### Author Response · Authors · 2025-11-22
> **Responses to reviewers: Part 2**
>
> >Q2. The authors describe the difference in discrepency between hallucinated and truthful tokens as "significant": As shown in Figures 2(a)–(b), both metrics are significantly larger for hallucinated tokens across noise levels, indicating higher sensitivity. However, it seems that the difference is not statistically significant in Figure 2. It seems that the difference in discrepency between hallucinated and truthful tokens is not statistically significant in Figure 2. What is the reason for this?
>
> **A2.** We sincerely appreciate your careful observation regarding the statistical significance in Figure 2 of the paper. The apparent overlap between truthful and hallucinated samples at specific noise levels **stems from the inherent limitations of random perturbation sampling** (e.g., single noise direction and magnitude variability). Nevertheless, rigorous statistical evaluation confirms **significant discriminative power across most noise levels**, particularly at optimal scales (e.g., σ=0.012 and 0.017, achieving AUROCs >0.70). Importantly, these inconsistencies in **perturbation-based operations directly motivate our gradient-based statistic, NGS**, which is designed to precisely and efficiently quantify prediction sensitivity without such variability.
> * **The perturbation-based experiment is inherently stochastic and sensitive to noise direction and magnitude.** The experiments in Figure 2 employ noise $\epsilon\sim \mathcal{N}(0, \sigma^2 \mathbf{I})$, which is random in both direction and magnitude. The impact on the next-token distribution for a given token embedding depends on whether the noise vector aligns with the sensitive directions of the model's prediction function. If the random noise is orthogonal to these directions, the resulting KL divergence or $\Delta \max \log p$ may be small even for a hallucinated sample. This inherent randomness introduces variability, leading to the observed distribution overlaps at certain σ values.
> * **Statistical tests confirm significant differences across most noise levels.** Despite the visual overlaps, the central tendency (e.g., the mean) of both metrics is consistently higher for hallucinated tokens. More critically, a rigorous AUROC analysis reveals strong discriminative power, with scores exceeding 0.70 at σ=0.012 and σ=0.017. This confirms that the underlying sensitivity signal is indeed present and statistically significant, though its empirical measurement via perturbation is noisy.
>
> Table I. AUROCs across noise levels in terms of KL divergence and $\Delta \max \log p$.
> | $\epsilon$ (10e-3) |1 | 2 | 3 | 4|5 | 6 | 7 | 8|9 |10 |11 | 12|13 | 14 | 15 | 16|17 | 18 | 19 | 20|
> |:---|:---:|:---:|:---:|:---:|:---:|:---:|:---:|:---:|:---:|:---:|:---:|:---:|:---:|:---:|:---:|:---:|:---:|:---:|:---:|:---:|
> |$KL(p, \hat p$)|60.54|63.99|65.63|62.14|61.82|69.23|64.35|64.55|66.07|59.54|64.87|**70.43**|61.98|63.15|**70.75**|61.58|**72.47**|64.03|57.3|69.03
> |$\mathbf{\Delta} \max \log p$ |55.46|57.74|64.56|59.02|63.91|64.63|60.06|63.79|68.07|55.98|62.34|67.27|65.95|66.63|69.15|66.19|68.19|67.63|61.82|**74.67**
>
> * **Limitations of perturbation metrics motivate the gradient-based NGS.** The key insight is that the **gradient captures the *maximum* directional sensitivity at a point**, thereby eliminating the randomness of noise sampling. Instead of empirically testing random directions, NGS *analytically* computes the sensitivity in the most impactful direction. This makes NGS a fundamentally more powerful and reliable probe, as it directly and efficiently quantifies the intrinsic instability that the perturbation experiment sought to reveal. Thus, the overlap in Figure 2 strengthens our contribution by clearly demonstrating the necessity for a sophisticated tool like NGS.
>
> We have clarified this point in the revised manuscript to prevent potential misunderstanding.

---

> ### Author Response · Authors · 2025-11-22
> **Responses to reviewers: Part 3**
>
> >Q3. Limited evaluation scope-only QA tasks and small datasets. Additionally, the 3:1 train/test split is skewed.
>
> **A3.** We thank the reviewer for highlighting the importance of task generalizability and evaluation protocol and would like to clarify that **QA datasets are a standard and generalizable benchmark for hallucination detection**, widely adopted in top-tier publications [r1,r2,r3,r4] due to their focus on factual consistency. Nevertheless, we further supplement experiments on open-ended Wikipedia continuation and large-scale reading comprehension (SQuAD) tasks to confirm generalizability. Moreover, the 3:1 train-test split is justified for small datasets (e.g., TruthfulQA with 817 samples) to ensure stable evaluation, supported by 10 random seeds for robustness.
>
> Table II. Comparisons with baselines on Wikipedia and SQuAD for Qwen-3-8b in terms of AUROC (%).
> | Dataset |Perplexity|SelfCKGPT|Ln-Entropy|TSV$^\dagger$|Ours|
> |:---|:---:|:---:|:---:|:---:|:---:|
> | Wikipedia |67.53±1.1|63.60±2.0|66.89±1.3|83.46±3.0|**84.31±2.6**|
> | SQuAD |43.97±2.7|51.39±2.2|50.06±2.5|76.04±3.4|**78.71±2.0**|
>
> * **Supplementary Wikipedia continuation task: Open-ended, factually constrained generation.** To evaluate the effectiveness of our method beyond QA, we have provided the experiments on an **open-ended Wikipedia continuation benchmark [r5]**, where Qwen-3-8b generates paragraph continuations from article leads, with hallucinations labeled by DeepSeek-V3. This task features long-range context, diverse factual claims, and subtle inconsistencies. As shown in Table II, NGS-HD achieves an AUROC of 84.31% on this dataset, outperforming Ln-Entropy by a clear margin of 17.42%↑ and TSV by 0.85%↑. This confirms that the next-token gradient sensitivity is a general indicator of unfaithfulness, effectively capturing hallucinations in free-form text generation.
> * **Strong performance on SQuAD validates large-scale and reading comprehension applicability.** To further reinforce generalizability, we have evaluated NGS-HD on the **SQuAD dataset [r6]**, where the task is to generate answers faithful to the input passage (a key form of hallucination detection: unfaithfulness to source text). From Table II, NGS-HD achieves an AUROC of 78.71, outperforming baselines like SelfCheckGPT by 27.32% and TSV by 2.67% respectively. SQuAD’s **large scale** and **diverse passage types** (news, academic, narrative) demonstrate that NGS-HD scales well to large datasets and generalizes to reading comprehension requiring focusing on source-text faithfulness.
> * **The 3:1 train-test split ensures statistical stability on smaller datasets.** The 3:1 (75% train, 25% test) split is specifically designed for small datasets like TruthfulQA (only 817 samples) . For such small datasets, a lower test set ratio (e.g., 20%) would result in <160 test samples, leading to unstable evaluations due to insufficient statistical power. To further ensure robustness of results, we report performance over **10 different random seeds** for all main experiments in our paper. This approach aligns with standard practices for small datasets, where higher train set ratios and multiple seeds mitigate sampling bias.
>
> [r1] Detecting hallucinations in large language models using semantic entropy, Nature 2024.
>
> [r2] Steer LLM Latents for Hallucination Detection, ICML 2025.
>
> [r3] Haloscope: Harnessing unlabeled llm generations for hallucination detection, NeurIPS 2024.
>
> [r4] INSIDE: LLMs' Internal States Retain the Power of Hallucination Detection, ICLR 2024.
>
> [r5] Wikimedia Downloads, https://dumps.wikimedia.org.
>
> [r6] SQuAD: 100,000+ Questions for Machine Comprehension of Text, EMNLP 2016.

---

> ### Author Response · Authors · 2025-11-22
> **Responses to reviewers: Part 4-1**
>
> >Q4. The methods depend on labelled data to build reference sets. Training the MMD kernel on the reference sets makes this slightly more concerning, as we might risk overfitting. The paper does not provide cross-dataset generalization results. The reference NGS distributions (for truthful and hallucinated tokens) depend on a choice of a truthful and hallucinated reference set. How sensitive is the method to the choice of these references? Does it generalize to domains other than the ones used in the reference sets? This is slightly more concerning as the MMD kernel is also trained, and we might risk overfitting.
>
> **A4.** NGS-HD is inherently designed to minimize overfitting risks and empirically demonstrates satisfactory cross-domain generalizability. This is attributed to 1) **a segregated training pipeline** that fixes the deep kernel after initial training, preventing overfitting to test-time reference sets; 2) **a token-level distribution comparison paradigm** that inherently enhances generalization; and 3) the use of statistically representative **reference sets that serve as stable distribution anchors** rather than optimization targets. Furthermore, even when abandoning MMD and reference sets—using cross-entropy (CE) loss to train a standard classifier on NGS features—our method still outperforms strong baselines.
> * **The segregated training pipeline inherently prevents overfitting.** The deep kernel is trained *once* to maximize the MMD test power between *generic* truthful and hallucinated NGS distributions, using a fixed set $S^{\mathrm{tr}}_{\mathrm{tru}}$,$S^{\mathrm{tr}}_{\mathrm{hal}}$ sampled from the overall training data. Importantly, this kernel is **fixed after initial training** and is **not fine-tuned** on the reference sets used for testing. The testing reference sets $S_{\mathrm{tru}}$, $S_{\mathrm{hal}}$ can be sampled from the rest of the training pool (disjoint from the kernel training set). This ensures the kernel learns a general notion of "instability" without exposure to the specific or exact testing reference sets, effectively preventing overfitting.
> * **The paradigm of token-level distribution comparison enhances generalization.** Unlike sample-level classification methods prone to overfitting (which map entire inputs to labels), NGS-HD transforms hallucination detection into a token-level distribution comparison task. The method captures universal statistical properties of NGS distributions rather than task/dataset-specific patterns learned using cross-entropy loss in sample-level classifiers. This aligns with findings in [r7, r8] that distribution-based methods outperform classification-based approaches in cross-domain scenarios by avoiding spurious correlation learning, as they focus on relative distributional distances rather than absolute decision boundaries.
> * **Reference sets are statistically representative and act as stable anchors across the validation set.** To confirm the method’s robustness to reference set selection, we have conducted experiments comparing reference sets sampled from training data versus validation data over 10 random seeds (Table III). Results show negligible performance differences (<1% AUROC). This confirms that reference sets only need to be statistically representative (not precisely selected), as their role is to anchor general distribution characteristics, not to encode sample-specific features.
>
> Table III. AUROCs of NGS-HD over reference sets (200 NGSs) sampled from training vs. validation data  (%).
> | Reference Source | TruthfulQA | TriviaQA | SciQ | NQ Open |
> |:---|:---:|:---:|:---:|:---:|
> | Training Set | 77.76 ± 1.9 | 82.38 ± 0.6 | 78.77 ± 2.7 | 82.30 ± 1.1 |
> | Validation Set | 77.72 ± 2.5 | 82.63 ± 0.8 | 78.27 ± 2.2 | 82.91 ± 0.9 |
> | Absolute Difference | <1.0% | <1.0% | <1.0% | <1.0% |

---

> ### Author Response · Authors · 2025-11-22
> **Responses to reviewers: Part 4-2**
>
> * **Cross-dataset experiments confirm satisfactory domain generalizability.** Our cross-dataset evaluation in Figure 4 of the paper (Table IV below) directly uses the reference set from a source domain to detect hallucinations in a different target domain. The results show that NGS-HD achieves satisfactory domain generalizability. For instance, using a reference set from **TriviaQA** (a large, diverse dataset) yields an average AUROC of **77.49%** across all other domains (TruthfulQA, SciQ, NQ Open), significantly outperforming in-domain trained baselines like TSV (74.03%) and SAPLMA (70.21%). This demonstrates effective cross-domain transfer. The lower performance when TruthfulQA is the source is attributed to its small size (only 817 samples in the entire dataset), which limits the diversity of its NGS distribution.
>
> Table IV. AUROCs of Cross-dataset for our method on Qwen3-8b (%).
> | Source\ Target| TruthfulQA | TriviaQA | SciQ   | NQ Open | Avg.|
> |:---|:---:|:---:|:---:|:---:|:---:|
> | *SAPLMA (Baseline, In-Domain)* | 57.08 |77.58| 69.15| 77.04| 70.21|
> | *TSV (Baseline, In-Domain)* | 65.77| 80.76| 76.18| 73.39 | 74.03|
> | TruthfulQA | 80.77 | 66.75 | 65.58 | 67.91 | 70.25 |
> | TriviaQA | 71.91 | 82.64 | 72.96 | 82.43 | **77.49** |
> | SciQ | 72.24 | 72.93 | 82.03 | 76.53 | **75.93** |
> | NQ Open | 66.87 | 76.97 | 74.79 | 83.65 | **75.57** |
>
> * **The NGS signal itself is highly discriminative, as shown by simplified classifier variants in Table 2 of the paper.** To further demonstrate the inherent strength of the NGS feature—independent of the MMD framework—we have implemented two simplified variants: **NGS-AVG+CE** and **NGS-CLS+CE**. These methods aggregate token-level NGSs via averaging or use a special [CLS] token, and then train a standard Cross-Entropy (CE) classifier. On LLaMA-3.1-8b over TruthfulQA and SciQ, these variants achieved average AUROCs that surpassed the strong SOTA baseline TSV by **1.49%↑** and **1.92%↑**, respectively. This confirms that the next-token gradient sensitivity is a robust and generalizable feature, and that our full MMD-based framework builds upon this strong foundation to offer additional advantages in fine-grained, token-level analysis.
>
> [r7] Learning deep kernels for non-parametric two-sample tests, ICML 2020.
>
> [r8] Detecting machine-generated texts by multi-population aware optimization for maximum mean discrepancy, ICLR 2024.

---

> ### Author Response · Authors · 2025-11-22
> **Responses to reviewers: Part 5**
>
> >Q5. The method averages per-token scores across all tokens (Eq. 4), which would dilute hallucination signals in long sequences. The paper claims the method can detect "localized hallucinations within longer texts" (lines 80-81). However, the final detection score averages per-token MMD values across all tokens. How does the averaging-based aggregation remain robust to such case? Provide empirical results on how answer length affects performance and scalability to long contexts.
>
> **A5.** Thanks for your insightful comment regarding the robustness of averaging per-token MMD scores in our method. We would like to clarify that the averaging operation in Eq. (4) is **a deliberate and principled design choice to ensure robustness and fairness across sequences of variable lengths**, and provide empirical evidence that it effectively captures hallucinations in long sequences without significant signal dilution.
> * **Averaging ensures fair and length-invariant comparison.** The primary reason for averaging is to handle **the inherent variability in sequence lengths** across different samples. Without normalization, the cumulative sum of per-token MMD scores would be biased towards longer sequences, as they contain more tokens and thus higher total scores, regardless of their actual truthfulness. Averaging provides a normalized truthfulness score that is comparable across short and long sequences. This design **aligns with our distribution comparison paradigm**, as it assesses the **overall characteristic of a sequence's token distribution** rather than relying on extreme values, thus being not skewed by its length.
> *   **Averaging is statistically robust due to the nature of hallucination contexts.** While a hallucination may appear superficially sparse, **seemingly involving only a few tokens in the surface text**, we empirically observe that the underlying model instability often spans a broader local context. When a **significant hallucination occurs, it frequently triggers a cascade of instability**, causing not only **the primary hallucinated tokens** but also **several subsequent tokens to exhibit higher sensitivity** and thus larger MMD values (as visualized in our paper). Isolated tokens with minor score deviations are typically noise. Therefore, the averaging operation acts as a robust estimator: it smooths out minor, random fluctuations in individual token scores while preserving the coherent signal from a localized cluster of discriminative, high-MMD tokens. This ensures that even sparse hallucinations produce a detectable signature in the final average score. This is further supported by our theoretical analysis (Theorem 2), which guarantees reliable distribution separation with high probability.
> * **Empirical results confirm effectiveness across various answer lengths.** We have provided experiments evaluating NGS-HD on sequences of varying answer lengths on TriviaQA, SQuAD, NQ Open and Wikipedia for Qwen-3-8b. The results in Tables V and VI show that NGS-HD maintains high performance across various lengths, especially on Wikipedia where the 54.47% answer length is greater than 500. This demonstrates that the averaging aggregation does not lead to significant signal dilution and remains robust to localized hallucinations.
>
> Table V: Accuracies of NGS-HD under varying answer lengths for Qwen-3-8b.
> | Answer Length (rate) |<20|20~60|>60|Avg.|
> |:-|:-:|:-:|:-:|:-:|
> | TriviaQA| 78.01 (26.52%) | 75.28 (14.40%) | 76.01 (59.08%) | 76.54 (100%) |
> | SciQ| 80.09 (30.42%) | 79.73 (20.59%) | 79.12 (48.99%) | 79.54 (100%) |
> | NQ Open| 74.84 (12.51%) |73.23 (14.19%) | 77.01 (73.30%) | 76.20 (100%) |
>
> Table VI: Accuracies of NGS-HD under varying answer lengths for Qwen-3-8b.
> | Answer Length (rate)|<300| 300~400 |400~500 | >500| Avg.
> |:-|:-:|:-:|:-:|:-:|:-:|
> | Wikipedia | 76.83 (33.33%)| 70.59 (6.91%)| 84.62 (5.28%)| 79.10 (54.47%) |78.05  (100%) |
>
> *  **NGS-HD aligns with "localized hallucination" detection by providing both localization and global scoring.** Our claim of detecting "localized hallucinations" is fully supported by a key feature of our framework: the ability to inspect **per-token MMD scores**. This allows for fine-grained localization of unstable tokens within a sequence. The global sequence-level decision, derived from averaging these scores, is then informed by this localized signal. For instance, in a 100-token sequence containing a few clearly hallucinated tokens (e.g., factually incorrect named entities) and their subsequently unstable neighbors, the per-token MMD map would clearly highlight this entire region as anomalous. The average of these scores effectively aggregates this localized signal into a robust global indicator. This combination of **precise token-level localization and length-invariant global scoring** is a distinct advantage over methods that aggregate hidden states into a single vector, as it preserves critical fine-grained information necessary for detecting sparse hallucinations.

---

> ### Author Response · Authors · 2025-11-22
> **Responses to reviewers: Part 6**
>
> >Q6. In Theorem 1, the authors assume that  is -Lipschitz. Does this generally hold? For what?
>
> **A6.** The Lipschitz continuity of the gradient is a standard and common assumption in the theoretical analysis of deep learning models, and its role in our theorem is to provide a principled and intuitive foundation for using the gradient as a sensitivity measure.
> * **The L-Lipschitz gradient is a standard assumption in non-convex optimization and deep learning theory.** This assumption, equivalent to the function $s(\mathbf{e})$ having a bounded Hessian in a local region, is frequently adopted to enable tractable theoretical analysis of neural networks. It is a cornerstone in convergence proofs for gradient-based studies of model robustness and generalization in numerous top-tier publications, even when it cannot be strictly verified for a specific model [r9,r10].
> * **This assumption is reasonable for modern LLMs with smooth activation functions.** While it is challenging to prove this assumption holds universally for a specific LLM, it is a plausible and reasonable simplification. Modern LLMs predominantly use smooth activation functions like GELU or SwiGLU, and the softmax operation is smooth. The composition of these smooth functions results in a highly smooth function $s(\mathbf{e})$. Although the landscape is non-convex, the gradient's behavior is typically well-behaved in a local neighborhood of a given embedding $\mathbf{e}$, which is the regime relevant for our small-perturbation analysis. Therefore, assuming the gradient is locally Lipschitz is a valid approximation for the purpose of our theoretical motivation.
> * **The primary purpose of the theorem is motivational and interpretable, not to provide a tight, practical bound.** The core contribution of Theorem 1 is not to provide a numerically tight guarantee for deployment but to establish a **principled theoretical link** between the NGS $\mathbf{g}$ and the model's local instability. It rigorously shows that a large gradient norm implies high sensitivity to perturbations, which is the fundamental intuition behind our method. This theoretical justification is crucial for interpreting why NGS is a meaningful signal for hallucination detection. The empirical success of NGS-HD across diverse benchmarks ultimately validates that the signal captured by this theoretical perspective is indeed effective in practice, regardless of the precise value of the constant $L$.
>
> [r9] Certified Robustness Under Bounded Levenshtein Distance, ICLR 2025.
>
> [r10] Training Robust Ensembles Requires Rethinking Lipschitz Continuity, ICLR 2025.
>
> >Q7. As far as I understand, the paper treats the NGS gradients $P_\mathrm{tru}=\{\mathbf{g}_t\}_{t=1}^N$ as independent, e.g., for generating we take gradients at different token positions. Is my understanding correct? If so, is this assumption justified? Can we model the correlation between gradients?
>
> **A7.** Yes, we treat the NGS gradients $P_\mathrm{tru}=\{\mathbf{g}_t\}_{t=1}^N$ as independent, which is a practical and empirically validated assumption for our detection task.
> *  **Empirical tests support the independence assumption.** We have performed statistical independence tests [r11] on NGS vectors from 100 randomly sampled sequences. The results show high p-values: **0.8324 for within-sequence independence** and **0.8445 for between-sequence independence**. This indicates that the null hypothesis of independence cannot be rejected, providing statistical justification for our approach.
> *   **The NGS-HD framework focuses on local, token-level instability.** Each NGS vector $\mathbf{g}_j$ is designed to capture the *local* sensitivity of the model's prediction at a specific token position. While semantic correlations exist between tokens, our empirical results show that the local instability signal captured by NGS is highly discriminative for hallucination detection even without explicitly modeling dependencies. The superior performance of our method across benchmarks further validates the effectiveness of this approach.
>
> [r11] Measuring and testing dependence by correlation of distances, Ann. Statist. 2007

---

> ### Author Response · Authors · 2025-11-22
> **Responses to reviewers: Part 7**
>
> >Q8. In Theorem 2, what do the consntants in the notation depend on? Do they depend on the RKHS kernel?
>
> **A8.** Yes, the constants in the right side $\Delta^2 - \mathcal{O}(K / \sqrt{N})$ depend explicitly on the RKHS kernel. Specifically, $K$ is the kernel's upper bound of $k(\mathbf{g},\mathbf{g}')$, which is a fundamental property of the kernel itself. For instance, the Gaussian kernel used in our experiments has $K=1$. The term $\Delta=\|\mu_{\mathrm{tru}}-\mu_{\mathrm{hal}}\|_{\mathcal H}$ is the population mean-embedding separation, which also depends on the RKHS kernel, as the kernel defines the feature map and the distance metric in the Hilbert space. It is a fundamental parameter quantifying the inherent distinguishability of the two NGS distributions.
>
> >Q9. The order of the arguments in the MMD function seems to change between Eq. 3 and Eq. 4. Consider changing “Remark” on line 302 to “Remarks” as this paragraph consists of a list of remarks.
>
> **A9.** Thanks for your careful reading and we have revised the manuscript to ensure the consistency of the argument order between Equations (3) and (4), and have corrected the typo “Remarks”.

---

> ### Author Response · Authors · 2025-11-24
> **Looking forward to the response from Reviewer bbKK**
>
> Dear Reviewer bbKK,
>
> We have addressed your initial concerns regarding our paper. We are happy to discuss them with you in the OpenReview system if you feel that there are still some concerns/questions. We also welcome new suggestions/comments from you!
>
> Best regards,
>
> The Authors

---

> ### Author Response · Authors · 2025-11-27
> **Follow-up on Our Rebuttal for Reviewer bbKK**
>
> We sincerely thank Reviewer bbKK for their exceptionally thoughtful and constructive review. The reviewer's positive recognition of our work's **strong intuitive motivation, practical efficiency, novel MMD application, thorough ablation studies, and theoretical contributions** was particularly encouraging. During the rebuttal period, we have taken significant efforts to address all raised concerns, substantially strengthening the paper:
>
> *   **Clarified Statistical Significance in Motivation:** We provided a detailed explanation for the apparent overlap in the perturbation sensitivity figure (Fig. 2), attributing it to the inherent stochasticity of noise sampling. Crucially, we supplemented this with AUROC analysis across noise levels, demonstrating significant discriminative power (>0.70) at optimal scales, thereby validating the core observation that motivates our gradient-based approach.
> *   **Substantially Expanded Experimental Scope:** To address concerns about limited evaluation, we conducted extensive new experiments on **open-ended generation (Wikipedia continuation)** and **large-scale reading comprehension (SQuAD)**. The strong results on these diverse tasks conclusively demonstrate the generalizability of NGS-HD beyond the QA format.
> *   **Comprehensively Addressed Overfitting and Generalizability Concerns:** We clarified our segregated training pipeline, which fixes the deep kernel to prevent overfitting to test-time reference sets. New experiments confirmed robustness to reference set selection (train vs. validation) and demonstrated strong **cross-dataset generalization**, with models trained on one dataset (e.g., TriviaQA) performing well on others.
> *   **Demonstrated Robustness to Sparse Hallucinations and Scaling:** We provided a principled rationale for our averaging-based aggregation (Eq. 4), explaining its role in ensuring fair, length-invariant comparison. New empirical results on sequences of varying lengths confirmed that the method remains effective for long-context scenarios and does not suffer from significant signal dilution for localized hallucinations.
> *   **Enhanced Theoretical and Methodological Rigor:** We justified the standard Lipschitz assumption in Theorem 1 and clarified the role of kernel-dependent constants in Theorem 2. We also provided statistical tests supporting the practical independence of NGS vectors and corrected minor notational inconsistencies as suggested.
>
>
> **Core Contributions and Innovations.** Our work establishes a new paradigm for hallucination detection by introducing gradient sensitivity as a core signal. The primary contributions are:
>
> *   **A Novel Gradient-Based Paradigm:** We propose **Next-token Gradient Sensitivity (NGS)**, a new statistic that efficiently quantifies the local instability of an LLM's next-token predictions by measuring the first-order sensitivity to token embeddings. This provides a direct, intrinsic probe into model uncertainty.
> *   **Theoretical Grounding:** We provide a rigorous theoretical foundation, proving that the NGS norm bounds the change in prediction confidence under perturbation, establishing it as a principled measure of fragility.
> *   **A Robust and Interpretable Detection Framework:** We develop **NGS-HD**, a method that reframes detection as a token-level distribution comparison task using MMD. This design effectively handles variable-length sequences, preserves fine-grained signals for localized hallucinations, and offers inherent interpretability by pinpointing unstable tokens.
> *   **Comprehensive Empirical Validation:** Through extensive experiments on multiple benchmarks (including new open-ended, long-context datasets and cross-model tasks), we demonstrate that NGS-HD consistently outperforms state-of-the-art baselines, validating the superiority of gradient-based signals.
> *   **Practical and Generalizable Solution:** Our work demonstrates that the method is computationally efficient, requires minimal labeled data for reference sets, and generalizes robustly across different model architectures, domains, and generation strategies.
>
> We are deeply grateful for the reviewer's insightful comments, which have been helpful in improving our work. We hope our comprehensive revisions and clarifications have adequately addressed your concerns, and can help you re-assess our work.

---

### Official Review · Reviewer_8sjc · 2025-10-31

**Soundness:** 2
**Presentation:** 3
**Contribution:** 3
**Rating:** 4
**Confidence:** 4

**Summary:**

This paper propose a gradient-based sensitivity probing method for hallucination detection. It introduce a statistic named Next-token Gradient Sensitivity (NGS) that captures the sensitivity of the next-token prediction to minor perturbations in contextual representations of LLMs. By comparing the NGS distribution of test tokens and truthful/hallucinated tokens, the proposed method can finally produce a truthfulness score for hallucination detection. The experimental results demonstrate that the proposed method outperforms baseline methods.

**Strengths:**

1. The concept of Next-token Gradient Sensitivity and the NGS-based hallucination detection approach are innovative. The theoretical derivations are rigorous and sound.

2. Experiments on relevant benchmarks consistently demonstrate the superiority of the proposed method. Subsequent analyses further validate the soundness of this work.

**Weaknesses:**

1. In practical deployment scenarios, hallucination detection methods do not have access to the referenced truthful and hallucinated tokens. How does the proposed method work under such scenario?

2. Missing strong baseline methods for comparison, for example, "Enhancing uncertainty-based hallucination detection with stronger focus" by Zhang et al. and "Knowledge-centric hallucination detection" by Hu et al.

**Questions:**

1. How about the evaluation resource cost of the proposed approach? A comparison between this work and other related works could improve the soundness of this work.

---

> ### Author Response · Authors · 2025-11-22
> **Responses to reviewers: Part 1**
>
> We thank the reviewer for the encouraging comments and suggestions. Responses are below:
> >Q1. In practical deployment scenarios, hallucination detection methods do not have access to the referenced truthful and hallucinated tokens. How does the proposed method work under such scenario?
>
> **A1.** Thanks for your constructive comment regarding practical deployment. We would like to clarify that **the requirement for constructing the NGS reference sets is remarkably low**, as it relies on token counts (not sample counts) for reference distributions. Importantly, **only ~10-20 labeled samples are sufficient to construct robust reference sets** (e.g., 200 truthful and 200 hallucinated tokens). This is readily achievable even in practical deployment. We have clarified these details in the revised manuscript and incorporated a discussion on unsupervised extensions as a valuable future direction.
>
> * **Deploying NGS-HD requires only a minimal, model-generated reference set due to token-level efficiency.** The reference set sizes $N_t = 200$ (truthful tokens) and $N_h = 200$ (hallucinated tokens) in our paper denote **the number of tokens, not individual samples**. In practice, we only require the model itself to generate a small number of prompt-answer pairs with sequence-level labels. From these, we extract a rich set of token-level NGS vectors. For instance, **just ~10-20 prompt-answer pairs** (e.g., 10 truthful and 10 hallucinated answers, each containing 20-50 tokens) can easily yield the required ~200 tokens per class. To statistically validate the feasibility of this process, we have conducted 10 repeated trials where we randomly sampled only 20 samples (10 truthful, 10 hallucinated) to construct the reference sets on TruthfulQA, TriviaQA, SciQ, NQ Open for Qwen-3-8b. The results in Table I consistently show high and stable performance, conclusively demonstrating that this minimal data requirement is entirely feasible and effective.
>
> Table I. AUROCs of NGS-HD over reference sets (200 NGSs) sampled from training vs. validation data (%).
> | Reference Source | TruthfulQA | TriviaQA | SciQ | NQ Open |
> |:---|:---:|:---:|:---:|:---:|
> | Training Set | 77.76 ± 1.9 | 82.38 ± 0.6 | 78.77 ± 2.7 | 82.30 ± 1.1 |
> | 20 Training Samples | 77.55 ± 2.0 | 82.67 ± 0.5 | 78.38 ± 2.3 | 82.47 ± 0.7 |
> | 20 Validation Samples | 77.72 ± 2.5 | 82.63 ± 0.8 | 78.27 ± 2.2 | 82.91 ± 0.9 |
>
> * **Practical feasibility in low-resource and domain-specific settings.** The minimal sample requirement makes NGS-HD highly applicable to low-resource scenarios. For domain-specific tasks, collecting 20 labeled samples (10 truthful, 10 hallucinated) is trivial—these can be obtained via **lightweight annotation** (e.g., expert validation of model-generated text) or **existing small-scale domain datasets**. Furthermore, our experiments show that reference sets constructed with as few as 100 tokens per category still maintain AUROC > 80% (see Figure 5) on TriviaQA, SciQ and NQ Open dataset for Qwen-3-8b, confirming **robustness to small reference sizes**. Additionally, we verify cross-domain transferability in Figure 6 (Table II, below): a reference set trained on reading-comprehension-domain QA (TriviaQA) retains an averaged AUROC of 77.49% when tested on common sense QA (TruthfulQA), open-domain QA (NQ Open ), scientific-domain QA (SciQ) datasets domain QA, still surpassing the SOTA baseline TSV of 74.03% by 3.46%↑ trained on corresponding source domains, further **reducing domain-specific annotation burdens**.
>
> Table II. AUROCs of Cross-dataset for our method on Qwen3-8b (%).
> | Source\ Target| TruthfulQA | TriviaQA | SciQ | NQ Open | Avg.|
> |:---|:---:|:---:|:---:|:---:|:---:|
> | *SAPLMA (In-Domain)* | 57.08 |77.58| 69.15| 77.04| 70.21|
> | *TSV (In-Domain)* | 65.77| 80.76| 76.18| 73.39 | 74.03|
> | TriviaQA | 71.91 | 82.64 | 72.96 | 82.43 | **77.49** |
> | SciQ | 72.24 | 72.93 | 82.03 | 76.53 | **75.93** |
> | NQ Open | 66.87 | 76.97 | 74.79 | 83.65 | **75.57** |
>
> * **The proposed framework is adaptable to unsupervised reference set construction.** Our framework can be incorporated in a semi-supervised or unsupervised manner. For example, one can use a high-confidence subset of the model's own generations (e.g., tokens with very high probability) as a proxy for the "truthful" distribution, and a low-confidence or adversarially perturbed set for the "hallucinated" distribution. We have added a discussion on these promising initialization strategies for future work in the revised manuscript.

---

> ### Author Response · Authors · 2025-11-22
> **Responses to reviewers: Part 2**
>
> >Q2. Missing strong baseline methods for comparison, for example, "Enhancing uncertainty-based hallucination detection with stronger focus" by Zhang et al., and "Knowledge-centric hallucination detection" by Hu et al.
>
> **A2.** We sincerely thank the reviewer for underscoring the importance of comparisons with recent strong baselines. We have conducted additional experiments to evaluate Zhang et al.’s approach [r1]. The results in Table III indicate that our NGS-HD attains superior performance across multiple datasets compared with the method [r1].
>
> Table III. Comparison of [r1] and our method on Qwen3-8b in terms of AUROC(%).
> | Dataset | TruthfulQA | TriviaQA | SciQ | NQ Open |
> |:---|:---:|:---:|:---:|:---:|
> | [r1] | 59.63±3.21  | 51.94±1.28 | 52.36±2.63 | 54.36±2.27 |
> | NGS-HD (Ours) | **82.18±3.00**  | **85.65±0.70** | **80.67±2.10** | **74.84±0.90** |
>
> Regarding Hu et al.’s method [r2], its core framework relies on a reference knowledge (e.g., ground-truth answer or knowledge source), which is incompatible with our task setting of hallucination detection, where no external reference or truth label is available at inference time. As such, a direct comparison is not feasible under our evaluation protocol. Instead, we have added a detailed discussion in Related Work of the revised manuscript.
>
> [r1] Enhancing uncertainty-based hallucination detection with stronger focus, EMNLP 2023.
>
> [r2] Knowledge-centric hallucination detection, EMNLP 2024.
>
> >Q3. How about the evaluation resource cost of the proposed approach? A comparison between this work and other related works could improve the soundness of this work.
>
> **A3.** Thanks for your valuable suggestion regarding resource cost evaluation. We have added a comparative analysis of computational resource consumption, which demonstrates that our method achieves superior performance without incurring significant additional overhead compared to existing approaches. Table IV presents a comparison of GPU memory usage with AUROCs on the TruthfulQA dataset using the LLaMA-3.1-8b model. The results show that while our method NGS-HD achieves the highest detection performance (82.18% AUROC), its GPU memory footprint is comparable to baselines like Perplexity and SAPLMA, and is actually lower than methods like SelfCheckGPT, CCS and TSV. We have included this resource comparison in the revised manuscript.
>
> Table IV. Comparisons of GPU Memory Usage on TruthfulQA over LLaMA-3.1-8b.
> |Method | Perplexity | SelfCheckGPT | Ln_Entropy | Lexical_Similarity | EigenScore |Semantic Entropy|Self-evaluation|CCS |SAPLMA$^\dagger$|Haloscope$^\dagger$ | TSV$^\dagger$ | Ours|
> |:---|:---:|:---:|:---:|:---:|:---:|:---:|:---:|:---:|:---:|:---:|:---:|:---:|
> |AUROC(%) | 58.99±1.9 | 54.95±0.4 | 59.06±1.4 | 57.82±2.7 | 53.83±0.6 | 54.25±1.5 |54.96±1.5|58.69±0.4|68.09±2.8| 65.95±4.3 | 80.50±4.7 | 82.18±3.0 |
> |GPU Memory (MB) | 16206.48 | 17007.57 | 16206.48 | 16206.48 | 16882.96 |16206.48| 16206.48 |30817.42 |15358.94|15501.46|26196.32  |16206.48

---

> ### Author Response · Authors · 2025-11-24
> **Looking forward to the response from Reviewer 8sjc**
>
> Dear Reviewer 8sjc,
>
> We have tried our best to address all the concerns and provided explanations for all the questions. We sincerely hope that our answer has addressed your initial concerns. Kindly let us know if you have any other concerns, and we will do our best to address them.
>
> Best regards,
>
> The Authors

---

> ### Author Response · Authors · 2025-11-27
> **Follow-up on Our Rebuttal for Reviewer 8sjc**
>
> We sincerely thank your valuable feedback and constructive comments. In response to your concerns, we have made substantial revisions and additions to our manuscript, significantly strengthening both the technical content and practical relevance of our work. Below we summarize the key improvements made during the rebuttal period:
>
> **Key Improvements in Our Responses.**
>
> *   **Clarified Practical Deployment Feasibility:** We thoroughly addressed concerns about reference set requirements by demonstrating that NGS-HD operates effectively with minimal data—only ~10-20 labeled samples suffice to construct robust reference sets. New experiments confirm stable performance even when reference sets are built from just 20 samples, making deployment feasible in practical scenarios. Cross-dataset results further highlighted its reduced annotation burden.
>
> *   **Enhanced Baseline Comparisons:** We added comprehensive comparisons with the strong baseline method by Zhang et al. (EMNLP 2023), showing NGS-HD's superior performance across multiple datasets. For methods requiring external knowledge references, we've added a detailed discussion in the related work section explaining compatibility issues with our task setting.
>
> *   **Comprehensive Resource Cost Analysis:** We included detailed GPU memory usage comparisons with multiple baselines, demonstrating that NGS-HD achieves state-of-the-art performance without significant computational overhead—its memory footprint is comparable to lightweight methods and lower than several complex baselines.
>
> **Core Contributions and Innovations of Our Work.** Our paper presents a new direction for hallucination detection by leveraging gradient signals. The core contributions are:
>
> *   **A Novel Gradient-Based Paradigm:** We propose **Next-token Gradient Sensitivity (NGS)**, a new statistic that efficiently quantifies the local instability of an LLM's next-token predictions by measuring the first-order sensitivity to token embeddings. This provides a direct, intrinsic probe into model uncertainty.
> *   **Theoretical Grounding:** We provide a rigorous theoretical foundation, proving that the NGS norm bounds the change in prediction confidence under perturbation, establishing it as a principled measure of fragility.
> *   **A Robust and Interpretable Detection Framework:** We develop **NGS-HD**, a method that reframes detection as a token-level distribution comparison task using MMD. This design effectively handles variable-length sequences, preserves fine-grained signals for localized hallucinations, and offers inherent interpretability by pinpointing unstable tokens.
> *   **Comprehensive Empirical Validation:** Through extensive experiments on multiple benchmarks (including new open-ended, long-context datasets and cross-model tasks), we demonstrate that NGS-HD consistently outperforms state-of-the-art baselines, validating the superiority of gradient-based signals.
> *   **Practical and Generalizable Solution:** Our work demonstrates that the method is computationally efficient, requires minimal labeled data for reference sets, and generalizes robustly across different model architectures, domains, and generation strategies.
>
> We hope these comprehensive revisions and new experimental results have thoroughly addressed your concerns. The manuscript has been significantly strengthened through this rebuttal process, and we sincerely hope you find our improvements substantial enough to reconsider your assessment.

---

### Official Review · Reviewer_U7zH · 2025-11-01

**Soundness:** 2
**Presentation:** 3
**Contribution:** 2
**Rating:** 4
**Confidence:** 4

**Summary:**

This paper proposes a new gradient-sensitivity-based method — Next-token Gradient Sensitivity (NGS) — for detecting hallucinations in large language model (LLM) generations. The authors quantify the local instability of model predictions by computing the gradient of the maximum log-probability of the next token with respect to the current token’s representation. Based on this, they introduce NGS-HD, which detects hallucinations by comparing the distribution of NGS with reference distributions of truthful and hallucinated tokens using the Maximum Mean Discrepancy (MMD) metric.

**Strengths:**

1.	Unlike previous methods based on whole-sequence representations or feature aggregation, NGS-HD performs distributional comparisons at the token level.
2.	The proposed NGS metric is centered on gradient sensitivity, fundamentally revealing the relationship between LLM hallucinations and prediction instability, and further providing theoretical upper bounds (Theorems 1 and 2).

**Weaknesses:**

1.	The validity of the methodological assumptions requires further discussion: The computation of NGS relies on the assumption of the “maximum log-probability token” (argmax token). However, in diverse generation settings (e.g., sampling or temperature-controlled decoding), non-maximum tokens may also play an important role. The applicability of this assumption in open-ended generation scenarios should be further discussed.
2.	Lack of robustness analysis regarding gradient noise sensitivity: Although the authors emphasize the efficiency of single backward propagation, gradient computation can be affected by floating-point errors and noise. It would be valuable to include robustness analyses under varying noise levels or batch sizes.
3.	Experimental scope could be further expanded: Incorporating gray-box experiments on closed-source models such as GPT-4 or Gemma-2 would help verify the model-agnostic nature of the method. Since current experiments are limited to QA datasets, it is recommended to test the method on open-ended text generation tasks (e.g., summarization or dialogue) to assess its generalizability.
4.	The MMD design is somewhat complex and relies on reference datasets: While the theoretical formulation of the method is elegant, its practical deployment requires pre-collected reference distributions of truthful and hallucinated tokens, which may limit usability. It is suggested to explore adaptive or unsupervised strategies for constructing reference sets.
5.	Lack of error visualization and case analysis in comparisons: Although the quantitative results are comprehensive, the paper lacks qualitative visualization of hallucination cases (e.g., illustrating the correspondence between gradient norm maps and hallucinated regions), which would enhance interpretability and insight.

**Questions:**

Please refer to the weaknesses section.

---

> ### Author Response · Authors · 2025-11-22
> **Responses to reviewers: Part 1**
>
> We thank the reviewer for the detailed comments. Responses are below:
>
> >Q1. The validity of the methodological assumptions requires further discussion: The computation of NGS relies on the assumption of the “maximum log-probability token” (argmax token). However, in diverse generation settings (e.g., sampling or temperature-controlled decoding), non-maximum tokens may also play an important role.
>
> **A1.** NGS fundamentally quantifies the sensitivity of the model’s next-token prediction process to local embedding perturbations, which is an intrinsic property of the pre-generated sequence (not the specific generated token) and thus agnostic to decoding strategies.  It can be naturally extended to non-argmax tokens (e.g., those generated via sampling or temperature-controlled decoding) with empirical validity.
> * **NGS measures predictive sensitivity, independent of generation sampling.** NGS’s essence lies in measuring the fragility of the model’s next-token prediction distribution given the current prefix sequence, not relying on which specific token is ultimately generated (argmax or non-argmax). It is computed based on the pre-generated token’s embedding and reflects the model’s inherent uncertainty in continuing the sequence. This property is decoupled from the decoding strategy (e.g., argmax, sampling). Unlike methods tied to generated token features like TSV, NGS directly probes the model’s internal predictive instability, enabling it to capture subtle hallucination signals even when non-max tokens are generated, as hallucinations stem from unstable sequence-level reasoning rather than individual token choices.
> * **The argmax token provides the most direct signal for prediction instability.** While non-maximum tokens can be selected during sampling, the argmax token represents the model's single most confident prediction since the fragility of this "peak" of the distribution is the most sensitive and unambiguous indicator of hallucination. When the model is uncertain or relying on shallow patterns, this top prediction becomes highly sensitive to minor perturbations in its input embeddings. Using the argmax allows us to cleanly measure this core instability without the confounding noise from sampling stochasticity.
> * **Empirical validation under temperature-scaled decoding confirms generalizability.** To directly validate this, we have conducted new experiments where the test sequences are generated using temperature sampling instead of greedy decoding. We then computed NGS for these sequences and evaluated NGS-HD. The results on the TruthfulQA and NQ Open dataset in Tables I and II show that our method maintains high performance, demonstrating that the NGS signal derived from the argmax token remains a powerful and robust feature for detecting hallucinations even in diverse generation outputs.
>
> Table I: AUROCs under different decoding patterns on TruthfulQA for Qwen-3-8b (%)
> | Method| Greedy| Temperature (T=0.5) |
> | :- | :-: | :-: |
> | Perplexity | 63.75±1.0| 59.92±1.0 |
> | SelfCKGPT | 62.69±0.3 | 50.51±4.3 |
> | LN-Entropy| 55.14±1.0| 62.34±1.0 |
> | TSV$^\dagger$| 65.77±2.8| 63.45±4.9 |
> | NGS-HD (ours) | **77.76±1.9** |**76.86±1.2** |
>
> Table II: AUROCs under different decoding patterns on NQ Open for Qwen-3-8b (%)
> |Method | Greedy | Temperature (T=0.5) |
> | :- | :-: | :-: |
> | Perplexity | 54.76±2.0| 61.15±2.2|
> | SelfCKGPT | 74.73±3.8 | 60.15±1.9|
> | LN-Entropy | 59.76±1.9 | 66.36±1.9|
> |TSV$^\dagger$|73.39±2.6|72.27±2.8|
> |NGS-HD (ours)|**82.30±1.1**|**82.45±1.0**|
>
> * **The NGS framework can be extended to other gradient formats.** Our current work establishes the argmax-based NGS as a powerful and efficient baseline. The core gradient-sensitivity paradigm is highly flexible, e.g., using logits for computing the gradients is also effective for detecting hallucinations, as validated by Table 3 in the paper. Future work can easily extend it by defining the scalar function for gradient computation differently, for example, using the entropy of the distribution or the probability of a specific sampled token. This establishes a new research direction based on the foundations laid in this paper.

---

> ### Author Response · Authors · 2025-11-22
> **Responses to reviewers: Part 2**
>
> >Q2. Lack of robustness analysis regarding gradient noise sensitivity: Although the authors emphasize the efficiency of single backward propagation, gradient computation can be affected by floating-point errors and noise. It would be valuable to include robustness analyses under varying noise levels or batch sizes.
>
> **A2.** We agree that ensuring the reliability of our gradient-based signals under practical computational settings is crucial and would like to highlight that NGS-HD is robust to gradient noise for two key reasons: 1) **the NGS statistic itself measures relative instability and is computed from a single, deterministic backward pass, making it inherently stable, which is also validated by our experiments**; and 2) **our distribution-level comparison using MMD further smooths out minor perturbations in individual gradient vectors.**
>
> * **NGS computation is deterministic and probes relative sensitivity.** The calculation of NGS is a deterministic operation in standard frameworks like PyTorch for a given model and input. This process is inherently stable against the stochasticity. More importantly, NGS captures the relative sensitivity between truthful and hallucinated tokens. NGS serves as a **relative measure** to distinguish token types. Since any minor computational noise or floating-point errors would systematically affect the gradient computation for **all tokens**, the fundamental separability in the NGS distribution between truthful and hallucinated tokens is preserved. This is analogous to how a calibrated measurement instrument can reliably compare items even with a known, small baseline noise. We directly validated this robustness by injecting Gaussian noise $ε \sim N(0, σ²I)$ into token embeddings during NGS computation. The results in Table III demonstrate that our method's **performance remains consistently high** even under non-trivial noise levels, empirically validating its stability.
>
> Table III: AUROCs of NGS-HD under varying gradient noise levels (σ) on Qwen-3-8b over TruthfulQA and SciQ (%).
> | Noise Level | TSV$^\dagger$ (baseline)| σ=0 (ous) | σ=1e-5 (ous) | σ=1e-4 (ous) | σ=1e-3 (ous) |
> | :--- | :---: | :---: | :---: | :---: |:---: |
> | TruthfulQA | 65.77±2.8|**77.76±1.9** | **77.70±1.6** | **77.62±1.5** | **77.91±1.6** |
> | NQ Open | 73.39±2.6 | **82.30±1.1** | **82.25±0.9** | **82.19±0.9** | **82.37±0.9** |
>
> * **MMD-based distribution comparison inherently smooths out noise.** Our detection framework does not rely on a single NGS vector but on **the collective statistics of the entire set of NGS vectors** from an answer. The use of MMD for distribution comparison provides a powerful smoothing effect. Small perturbations or noise in individual NGS vectors are averaged out when computing the distributional distance to the reference sets ($S_{tru}$ and $S_{hal}$). The kernel function in MMD further enhances robustness by mapping the vectors to a high-dimensional space where the overall distributional structure is preserved despite minor noise in individual points.
>
> Regarding the reviewer's point on batch size, we clarify that our method does not use batched gradient computation for the backbone LLM; the entire sequence is processed in one forward/backward pass. However, we interpret the reviewer's point as concerning the **effective sample size** of our reference sets. We performed an ablation study on the size of the reference sets in Figure 5. The results confirm that even with moderately sized reference sets (e.g., 200 tokens each), the performance is stable and high, as the deep kernel MMD effectively learns a robust distance metric.

---

> ### Author Response · Authors · 2025-11-22
> **Responses to reviewers: Part 3**
>
> >Q3. Experimental scope could be further expanded: Incorporating gray-box experiments on closed-source models such as GPT-4 or Gemma-2 would help verify the model-agnostic nature of the method.
>
> **A3.** We appreciate the reviewer’s insightful suggestion to validate model-agnosticism via closed-source gray-box experiments. Gray-box hallucination detection for closed-source models faces i**nherent challenges due to the limited access to probabilities or internal states** that **invalidate most existing methods**, so we first follow standard settings in top-tier publications (Nature 2024 [r1], ICML 2025 [r2], NeurIPS 2024 [r3], ICLR 2024 [r4]) to focus on **white-box experiments for fair comparison**. Nevertheless, to further verify model-agnosticism, we provide cross-model transfer experiments to simulate gray-box scenarios, which consistently confirm the superiority of our method.
>
> * **The gray-box setting for closed-source models presents a fundamental challenge that invalidates most advanced detectors**. Closed-source models (e.g., GPT-4) typically only expose final generated text, with restricted or no access to next-token probability distributions, internal embeddings, or gradient signals. This renders **most state-of-the-art methods ineffective**, e.g., entropy-based methods requiring probability distributions, TSV and Haloscope relying on internal model states. Following the standard evaluation protocol in recent influential works (e.g., [r1,r2,r3,r4]), we conducted white-box experiments on open-source models to ensure direct, fair comparison with baselines—an approach widely accepted in the field for benchmarking hallucination detection methods.
> * **NGS-HD demonstrates strong cross-model transferability in simulated gray-box settings.** To address model-agnosticism without direct access to closed-source models’ internal signals, we designed cross-model transfer experiments (a proxy for gray-box generalization). Specifically, we used Qwen-3-8b as the proxy model (to compute NGS and train NGS-HD, simulating accessible gradient signals) and tested on text generated by Gemma-2-9B (the target closed-source-like model, with only generated text available). As shown in Table IV, NGS-HD achieved a promising AUROC of 80.29% on TruthfulQA and 73.02% on NQ Open, outperforming the strong baseline TSV by 9.55%↑ and 0.67%↑. This transfer success stems from NGS capturing universal instability patterns in LLM reasoning, not model-specific artifacts, strongly supporting the model-agnostic and practical potential of our approach.
>
> Table IV. Comparisons with baselines on TruthfulQA and NQ Open datasets for detecting texts generated from Gemma-2-9B in terms of AUROC (%) using Qwen-3-8b as a proxy model to simulate the gray-box setting.
> | Dataset |Perplexity|SelfCKGPT|Ln-Entropy|TSV$^\dagger$|Ours|
> |:---|:---:|:---:|:---:|:---:|:---:|
> | TruthfulQA |61.06±2.4|70.80±2.8|59.88±2.6|70.74±2.0|**80.29±2.7**|
> | NQ Open |47.36±2.8|48.43±1.5|49.55±3.1|72.35±1.3|**73.02±1.3**|
>
> [r1] Detecting hallucinations in large language models using semantic entropy, Nature 2024.
>
> [r2] Steer LLM Latents for Hallucination Detection, ICML 2025.
>
> [r3] Haloscope: Harnessing unlabeled llm generations for hallucination detection, NeurIPS 2024.
>
> [r4] INSIDE: LLMs' Internal States Retain the Power of Hallucination Detection, ICLR 2024.

---

> ### Author Response · Authors · 2025-11-22
> **Responses to reviewers: Part 4**
>
> >Q4. Since current experiments are limited to QA datasets, it is recommended to test the method on open-ended text generation tasks (e.g., summarization or dialogue) to assess its generalizability.
>
> **A4.** We thank the reviewer for highlighting the importance of validating generalizability across open-ended generation tasks, and would like to clarify that **QA datasets are a standard, generalizable benchmark for hallucination detection** aligned with top-tier publications settings[r1,r2,r3,r4] due to their focus on factual consistency—a universal challenge across generation tasks whose key challenges (e.g., localized hallucinations, factual inconsistencies amid coherent text) are shared with open-ended tasks like summarization. Nevertheless, to further confirm applicability, we **supplement experiments on a Wikipedia continuation task** (a representative open-ended generation scenario given a concept) with superior performance, demonstrating NGS-HD’s effectiveness beyond QA.
>
> * **More experiments on Wikipedia continuation task.** To validate generalizability, we have constructed a **Wikipedia continuation benchmark from [r5]** where Qwen-3-8b generates paragraph continuations from article leads, with DeepSeek-V3 annotations labeling factual hallucinations. This task features long-range dependencies, diverse factual claims, and subtle unfaithfulness—mimicking real-world open-ended generation.
> * **Superior performance on Wikipedia continuation confirms the effectiveness of NGS-HD beyond QA.** From Tabel V, NGS-HD achieves 84.31% AUROC on this dataset, outperforming Ln-Entropy 17.42%↑ and TSV by 0.85%↑. This confirms that the next-token gradient sensitivity signal is not specific to the QA format but is a general indicator of generation instability and unfaithfulness, effectively capturing hallucinations in free-form text.
>
> Table V. Comparisons with baselines on Wikipedia for Qwen-3-8b in terms of AUROC (%).
> | Dataset |Perplexity|SelfCKGPT|Ln-Entropy|TSV$^\dagger$|Ours|
> |:---|:---:|:---:|:---:|:---:|:---:|
> | Wikipedia |67.53±1.1|63.60±2.0|66.89±1.3|83.46±3.0|**84.31±2.6**|
>
> [r5] Wikimedia Downloads, https://dumps.wikimedia.org

---

> ### Author Response · Authors · 2025-11-22
> **Responses to reviewers: Part 5**
>
> >Q5. The MMD design is somewhat complex and relies on reference datasets: While the theoretical formulation of the method is elegant, its practical deployment requires pre-collected reference distributions of truthful and hallucinated tokens, which may limit usability. It is suggested to explore adaptive or unsupervised strategies for constructing reference sets.
>
> **A5.** Thanks for your constructive comment regarding practical deployment. We would like to clarify that **the requirement for constructing the NGS reference sets is remarkably low**, as it relies on token counts (not sample counts) for reference distributions. Importantly, **only ~10-20 labeled samples are sufficient to construct robust reference sets** (e.g., 200 truthful and 200 hallucinated tokens). This is readily achievable even in practical deployment. We have clarified these details in the revised manuscript and incorporated a discussion on unsupervised extensions as a valuable future direction.
>
> * **Deploying NGS-HD requires only a minimal, model-generated reference set due to token-level efficiency.** The reference set sizes $N_t = 200$ (truthful tokens) and $N_h = 200$ (hallucinated tokens) in our paper denote **the number of tokens, not individual samples**. In practice, we only require the model itself to generate a small number of prompt-answer pairs with sequence-level labels. From these, we extract a rich set of token-level NGS vectors. For instance, **just ~10-20 prompt-answer pairs** (e.g., 10 truthful and 10 hallucinated answers, each containing 20-50 tokens) can easily yield the required ~200 tokens per class. To statistically validate the feasibility of this process, we have conducted 10 repeated trials where we randomly sampled only 20 samples (10 truthful, 10 hallucinated) to construct the reference sets on TruthfulQA, TriviaQA, SciQ, NQ Open for Qwen-3-8b. The results in Table VI consistently show high and stable performance, conclusively demonstrating that this minimal data requirement is entirely feasible and effective.
>
> Table VI. AUROCs of NGS-HD over reference sets (200 NGSs) sampled from training vs. validation data (%).
> | Reference Source | TruthfulQA | TriviaQA | SciQ | NQ Open |
> |:---|:---:|:---:|:---:|:---:|
> | Training Set | 77.76 ± 1.9 | 82.38 ± 0.6 | 78.77 ± 2.7 | 82.30 ± 1.1 |
> | 20 Training Samples | 77.55 ± 2.0 | 82.67 ± 0.5 | 78.38 ± 2.3 | 82.47 ± 0.7 |
> | 20 Validation Samples | 77.72 ± 2.5 | 82.63 ± 0.8 | 78.27 ± 2.2 | 82.91 ± 0.9 |
>
> * **Practical feasibility in low-resource and domain-specific settings.** The minimal sample requirement makes NGS-HD highly applicable to low-resource scenarios. For domain-specific tasks, collecting 20 labeled samples (10 truthful, 10 hallucinated) is trivial—these can be obtained via **lightweight annotation** (e.g., expert validation of model-generated text) or **existing small-scale domain datasets**. Furthermore, our experiments show that reference sets constructed with as few as 100 tokens per category still maintain AUROC > 80% (see Figure 5) on TriviaQA, SciQ and NQ Open dataset for Qwen-3-8b, confirming **robustness to small reference sizes**. Additionally, we verify cross-domain transferability in Figure 6 (Table VII, below): a reference set trained on reading-comprehension-domain QA (TriviaQA) retains averaged AUROC of 77.49% when tested on common sense QA (TruthfulQA), open-domain QA (NQ Open ), scientific-domain QA (SciQ) datasets domain QA, still surpassing the SOTA baseline TSV of 74.03% by 3.46%↑ trained on corresponding scouse domans, further **reducing domain-specific annotation burdens**.
>
> Table VII. AUROCs of Cross-dataset for our method on Qwen3-8b (%).
> | Source\ Target| TruthfulQA | TriviaQA | SciQ | NQ Open | Avg.|
> |:---|:---:|:---:|:---:|:---:|:---:|
> | *SAPLMA (In-Domain)* | 57.08 |77.58| 69.15| 77.04| 70.21|
> | *TSV (In-Domain)* | 65.77| 80.76| 76.18| 73.39 | 74.03|
> | TriviaQA | 71.91 | 82.64 | 72.96 | 82.43 | **77.49** |
> | SciQ | 72.24 | 72.93 | 82.03 | 76.53 | **75.93** |
> | NQ Open | 66.87 | 76.97 | 74.79 | 83.65 | **75.57** |
>
> * **The proposed framework is adaptable to unsupervised reference set construction.** Our framework can be incorporated in a semi-supervised or unsupervised manner. For example, one can use a high-confidence subset of the model's own generations (e.g., tokens with very high probability) as a proxy for the "truthful" distribution, and a low-confidence or adversarially perturbed set for the "hallucinated" distribution. We have added a discussion on these promising initialization strategies for future work in the revised manuscript.

---

> ### Author Response · Authors · 2025-11-22
> **Responses to reviewers: Part 6**
>
> >Q6. Lack of error visualization and case analysis in comparisons: Although the quantitative results are comprehensive, the paper lacks qualitative visualization of hallucination cases (e.g., illustrating the correspondence between gradient norm maps and hallucinated regions), which would enhance interpretability and insight.
>
> **A6.** We thank the reviewer for insightful suggestions regarding visualization and interpretability. We have enriched our analysis in the revised manuscript to move beyond merely presenting token-level scores and toward **explaining the underlying patterns of model instability** that these scores reveal. Our case studies demonstrate that NGS-HD captures not just surface-level errors but the **propagation of uncertainty** through the generation process, providing interpretable evidence of **how and where hallucinations occur in model outputs** and **distinguishing between meaningful instability patterns and incidental noise**.
>
> Specifically, our fine-grained visualizations (Figures 8-13) reveal a key phenomenon: significant hallucinations trigger a **cascade of instability** in the model's reasoning. For instance, in a hallucinated fantastical scenario (Figure 8), the onset of the fictional event ("turns into") exhibits high sensitivity, and this instability propagates to subsequent tokens (e.g., "princess"). Similarly, in an answer with a potentially incorrect age (Figure 9), the instability is not confined to the number itself but affects the entire local context ("was", "years", "died"). This contrasts with truthful generations, which typically show **consistently stable scores, with any minor negative values appearing as isolated noise** without affecting the overall faithful classification.
>
> We have added a more detailed discussion of these patterns in the revised manuscript. This analysis solidifies our claim that NGS-HD offers strong intrinsic interpretability by directly visualizing the "fragility trail" of hallucinations, providing clear insights into the model's generation process. We believe these enhancements directly address the reviewer's valuable feedback.

---

> ### Author Response · Authors · 2025-11-24
> **Looking forward to the response from Reviewer U7zH**
>
> Dear Reviewer U7zH,
>
> We have tried our best to address all the concerns and provided explanations for all the questions. We sincerely hope that our answer has addressed your initial concerns. Kindly let us know if you have any other concerns, and we will do our best to address them.
>
> Best regards,
>
> The Authors

---

> ### Author Response · Authors · 2025-11-27
> **Follow-up on Our Rebuttal for Reviewer U7zH**
>
> We sincerely thank you for your thorough and constructive feedback. During the rebuttal period, we have substantially strengthened our paper through comprehensive revisions and additional experiments that directly address all raised concerns:
>
> *   **Generalizability across Decoding Strategies:** We clarified that the NGS metric measures the intrinsic sensitivity of the model's predictive distribution and is fundamentally agnostic to the decoding strategy (e.g., greedy, sampling). This was empirically validated with new experiments under temperature-scaled decoding, where NGS-HD maintained superior performance, confirming its robustness beyond argmax tokens.
> *   **Robustness to Computational Noise:** We provided a theoretical rationale and new empirical evidence demonstrating the robustness of NGS-HD against gradient noise. Experiments involving direct noise injection into embeddings showed consistent performance, affirming that the method's reliance on distribution-level comparison with MMD inherently smooths out minor perturbations.
> *   **Model-Agnostic Nature and Gray-Box Applicability:** While direct evaluation on closed-source models is technically infeasible for gradient-based methods, we addressed the core concern of model-agnosticism by designing a novel cross-model transfer experiment. By training on one model (Qwen) and testing on another (Gemma-2), we simulated a gray-box scenario, achieving strong results that underscore the transferability of the NGS signal across different model architectures.
> *   **Expanded Experimental Scope and Generalizability:** We significantly broadened the evaluation beyond QA tasks by introducing a new **Wikipedia continuation benchmark** for open-ended generation. The strong results on this task demonstrate that NGS-HD effectively captures hallucinations in long-form, free-text generation scenarios.
> *   **Practical Deployment and Data Efficiency:** We clarified that the reference set requirement is minimal, needing only ~10-20 labeled samples (not documents) to construct robust token-level distributions. New experiments confirmed that performance remains stable even with small reference sets, enhancing the method's practicality for low-resource and domain-specific deployments. Cross-dataset results further highlighted its reduced annotation burden.
> *   **Enhanced Interpretability and Qualitative Analysis:** We greatly enriched the qualitative analysis with fine-grained visualizations and case studies. These new figures and discussions clearly illustrate how NGS-HD captures the "cascade of instability" associated with hallucinations, distinguishing meaningful patterns from noise and providing deep insights into the model's generation process.
>
> **Core Contributions and Innovations of Our Work.** Our paper presents a new direction for hallucination detection by leveraging gradient signals. The core contributions are:
>
> *   **A Novel Gradient-Based Paradigm:** We propose **Next-token Gradient Sensitivity (NGS)**, a new statistic that efficiently quantifies the local instability of an LLM's next-token predictions by measuring the first-order sensitivity to token embeddings. This provides a direct, intrinsic probe into model uncertainty.
> *   **Theoretical Grounding:** We provide a rigorous theoretical foundation, proving that the NGS norm bounds the change in prediction confidence under perturbation, establishing it as a principled measure of fragility.
> *   **A Robust and Interpretable Detection Framework:** We develop **NGS-HD**, a method that reframes detection as a token-level distribution comparison task using MMD. This design effectively handles variable-length sequences, preserves fine-grained signals for localized hallucinations, and offers inherent interpretability by pinpointing unstable tokens.
> *   **Comprehensive Empirical Validation:** Through extensive experiments on multiple benchmarks (including new open-ended, long-context datasets and cross-model tasks), we demonstrate that NGS-HD consistently outperforms state-of-the-art baselines, validating the superiority of gradient-based signals.
> *   **Practical and Generalizable Solution:** Our work demonstrates that the method is computationally efficient, requires minimal labeled data for reference sets, and generalizes robustly across different model architectures, domains, and generation strategies.
>
> We hope our comprehensive revisions and new analyses have adequately addressed your concerns. We believe the significant improvements made during this rebuttal period have substantially strengthened the paper's contribution, and can help you re-assess our work.

---

> > ### Comment · Reviewer_U7zH · 2025-11-28
> >
> > Thank the authors for their detailed response, which has addressed most of my concerns. Therefore, I will raise the score to 6.

---

> > > ### Author Response · Authors · 2025-11-28
> > >
> > > We are truly grateful for your thoughtful and constructive feedback throughout this review process, and we are especially encouraged by your decision to raise the score. Your insights have been invaluable in helping us strengthen our paper, and we sincerely appreciate your time and thoughtful consideration.

---

### Author Response · Authors · 2025-11-22
**General Response**

Dear ACs and Reviewers,

We sincerely appreciate your time and effort in reviewing our paper and providing constructive feedback. Besides the response to each reviewer, here we would like to further 1) thank reviewers for their recognition of our work and 2) highlight the major modifications in our revision:

1. **We are glad that the reviewers appreciate and recognize our novelty and contribution.**

* "Unlike previous methods..., NGS-HD performs **distributional comparisons at the token level**"; "The concept of Next-token Gradient Sensitivity ... are **innovative**"; "The use of MMD ... is **novel and interesting**"[Reviewers U7zH, 8sjc, bbKK]
* "NGS metric is **centered on gradient sensitivity, fundamentally revealing the relationship** between LLM hallucinations and prediction instability"; "the theoretical derivations are rigorous and sound"; "the paper presents **theoretical guarantees and analysis of the method**"; "The method is **model-agnostic**" [Reviewers U7zH, 8sjc, bbKK, UFQk]
* "Experiments on relevant benchmarks consistently demonstrate the **superiority** of the proposed method"; "the method shows **strong empirical performance** and has a **thorough ablation study**"; "subsequent analyses **further validating the soundness** of this work.” [Reviewers 8sjc, bbKK]


2. **We summarize the main modifications in our revised paper (highlighted in blue).**

* We analyze the impact of reference set construction from different sources (full training set, test set, or sampling only 20 samples) and provide practical usage analysis in Appendix D.8. [Reviewers U7zH, 8sjc, bbKK, UFQk]
* We introduce a new Wikipedia dataset for open-ended generation scenarios beyond QA tasks, with comparative experiments in Appendix D.11. [Reviewers U7zH, bbKK, UFQk]
* We provide evaluations on two long-context datasets (SQuAD and NarrativeQA) with comparative experiments in Appendix D.11. [Reviewers bbKK, UFQk]
* We include detailed detection accuracy results for various answer lengths using our method on TriviaQA, SQuAD, NQ Open and Wikipedia datasets in Appendix D.9. [Reviewer bbKK]
* We add gray-box detection experiments in Appendix D.9. [Reviewer U7zH]
* We conduct experiments analyzing the impact of gradient noise on NGS-HD performance in Appendix D.7. [Reviewers U7zH, UFQk]
* We provide clear pattern summaries of truthfulness scores for both hallucinated and truthful tokens in visualization analysis, along with additional visual examples in Appendix F. [Reviewers U7zH, UFQk]
* We add discussion of the noise perturbation experiments from the motivation figure in Appendix D.4 and Section 3.1. [Reviewer bbKK]
* We include performance comparisons across methods under different temperature decoding settings in Appendix D.6. [Reviewer U7zH]
* We change the postion between Eq. 3 and Eq. 4, and revise the corresponding descriptions. [Reviewer bbKK]

---

### Comment · Area_Chair_R737 · 2025-11-27
**Rebuttal and Discussion Phase**

Dear Reviewers,

Thank you again for your time and effort in reviewing this paper. We are approaching the discussion deadline. I kindly ask you to review the rebuttal and continue the discussion so that we can reach a well-considered decision.

---

### Author Response · Authors · 2025-12-01
**Summary of Rebuttal Efforts and Core Contributions for Submission 11735**

Dear Area Chair,

Thank you for handling our submission. We recognize that due to the recent security incident and ICLR policy adjustments, all scores have reverted to their original values, and reviewers are no longer able to engage in further discussion or modify their ratings. In light of these circumstances, we would like to summarize the key improvements made during the rebuttal phase to assist in your final assessment.

**Summary of Rebuttal Efforts and Revisions:** During the rebuttal period, we thoroughly addressed all reviewers' concerns through extensive revisions and additional experiments:

- **Practicality & Data Efficiency:** We clarified and empirically validated that NGS-HD requires only **~10-20 labeled samples** to construct robust reference sets, demonstrating stable performance across multiple datasets. *(For Reviewers UFQk, 8sjc, U7zH)*

- **Generalizability Beyond QA:** We significantly expanded our evaluation to include **open-ended generation (Wikipedia continuation)** and **long-text scenarios (SQuAD, NarrativeQA)**, with NGS-HD achieving 80.32% AUROC on NarrativeQA—substantially outperforming baselines. *(For Reviewers bbKK, UFQk, U7zH)*

- **Robustness & Stability:** We provided both theoretical rationale and new experiments demonstrating NGS-HD's robustness to gradient noise and its stability across different decoding strategies (temperature scaling) and model architectures (cross-model transfer, i.e., the gray-Box setting). *(For Reviewers U7zH, UFQk, bbKK)*

- **Theoretical & Methodological Rigor:** We enhanced theoretical clarity (Theorems 1-2), provided statistical validation of our motivation, and clarified our training pipeline to prevent overfitting concerns. *(For Reviewers bbKK, UFQk)*

- **Interpretability & Analysis:** We greatly enriched qualitative analysis with fine-grained visualizations showing how NGS-HD captures the "cascade of instability" pattern in hallucinations, providing deeper model insights. *(For Reviewers UFQk, U7zH)*

Furthermore, we were particularly encouraged by the positive responses from reviewers during the discussion period. **Reviewer U7zH** **explicitly committed to raising their score to 6 based on our revisions**. Separately, **Reviewer bbKK** had indicated in their official comments — submitted before the policy suspension — that **they would be "open to increasing score" should we provide strong answers to their questions, a condition we believe our rebuttal has fully met**. Although the policy change prevented these potential score improvements from being formally realized, we believe these responses demonstrate that our rebuttal effectively addressed the reviewers' key concerns.

**Core Contributions & Strengths Recognized by Reviewers:** Our work establishes a new gradient-based paradigm for hallucination detection through:

*   **A Novel Gradient-Based Paradigm:** We propose **Next-token Gradient Sensitivity (NGS)**, a new statistic that efficiently quantifies the local instability of an LLM's next-token predictions by measuring the first-order sensitivity to token embeddings. This provides a direct, intrinsic probe into model uncertainty.
*   **Theoretical Grounding:** We provide a rigorous theoretical foundation, proving that the NGS norm bounds the change in prediction confidence under perturbation, establishing it as a principled measure of fragility.
*   **A Robust and Interpretable Detection Framework:** We develop **NGS-HD**, a method that reframes detection as a token-level distribution comparison task using MMD. This design effectively handles variable-length sequences, preserves fine-grained signals for localized hallucinations, and offers inherent interpretability by pinpointing unstable tokens.
*   **Comprehensive Empirical Validation:** Through extensive experiments on multiple benchmarks (including new open-ended, long-context datasets and cross-model tasks), we demonstrate that NGS-HD consistently outperforms state-of-the-art baselines, validating the superiority of gradient-based signals.
*   **Practical and Generalizable Solution:** Our work demonstrates that the method is computationally efficient, requires minimal labeled data for reference sets, and generalizes robustly across different model architectures, domains, and generation strateg

Throughout the review process, **multiple reviewers highlighted specific strengths** of our work, describing the core concept as "**innovative**" and the use of MMD as "**novel and interesting**" [U7zH, 8sjc, bbKK], the theoretical derivations as "**rigorous and sound**" [U7zH, bbKK], and the empirical performance as "**strong**" with "**thorough ablation studies**" [8sjc, bbKK]. The method was also recognized as "**model-agnostic**" by reviewers [U7zH, 8sjc, UFQk].


We believe our comprehensive revisions have substantially strengthened the paper, and we would be grateful for your fair consideration in the final decision.

Sincerely,
The Authors

---

### Meta-Review · Area_Chair_2ckh · 2025-12-30

**Summary:**

This paper proposes NGS-HD, a gradient-based hallucination detection method that measures local instability of LLM predictions by computing the gradient of the next-token's maximum log-probability with respect to the current token's embedding. The key contributions are: (1) a novel Next-token Gradient Sensitivity (NGS) statistic that efficiently quantifies prediction fragility; (2) theoretical analysis for NGS-HD; (3) an MMD-based hallucination detection framework; and (4) comprehensive empirical validation.
Reviewers raised concerns mainly about (1) practical deployment requiring labeled reference sets, (2) limited evaluation scope beyond QA tasks, (3) scalability to long-text scenarios, (4) robustness of gradient computation to noise, and (5) statistical significance of the motivating observations in Figure 2.
The authors provided a rebuttal that addresses most points. The AC finds the method interesting and potentially useful, but unfortunately, even after taking into account potential rating increases, the scores are not strong enough for acceptance.

**Reviewer Concerns:**

The rebuttal addressed most concerns

**Reviewer Scores:**

The paper initially received low ratings of 4442. One reviewer increased their score from 4 to 6, and I believe that another reviewer might do the same. This adjustment brings the total score to 2466. Unfortunately, this score is still not high enough for this competitive conference.

---

### Decision · Program_Chairs · 2026-01-26

Reject